# Kinematics and extent of the Piemont-Liguria Basin – implications for subduction processes in the Alps

Eline Le Breton[1], Sascha Brune[2,3], Kamil Ustaszewski[4], Sabin Zahirovic[5], Maria Seton[5], R. Dietmar Müller[5]

[1]Department of Earth Sciences, Freie Universität Berlin, Germany
[2]Geodynamic Modelling Section, German Research Centre for Geosciences, GFZ Potsdam, Germany
[3]Institute of Geosciences, University of Potsdam, Potsdam, Germany
[4]Institute for Geological Sciences, Friedrich-Schiller-Universität Jena, Germany
[5]EarthByte Group, School of Geosciences, The University of Sydney, NSW 2006, Australia

*Correspondence to*: Eline Le Breton (eline.lebreton@fu-berlin.de)

**Abstract.** Assessing the size of a former ocean, of which only remnants are found in mountain belts, is challenging but crucial to understand subduction and exhumation processes. Here we present new constraints on the opening and width of the Piemont-Liguria (PL) Ocean, known as the Alpine Tethys together with the Valais Basin. We use a regional tectonic reconstruction of the Western Mediterranean-Alpine area, implemented into a global plate motion model with lithospheric deformation, and 2D thermo-mechanical modelling of the rifting phase to test our kinematic reconstructions for geodynamic consistency. Our model fits well with independent datasets (i.e. ages of syn-rift sediments, rift-related fault activity and mafic rocks) and shows that, between Europe and northern Adria, the PL Basin opened in four stages: (1) Rifting of the proximal continental margin in Early Jurassic (200-180 Ma), (2) Hyper-extension of the distal margin in Early-Middle Jurassic (180-165 Ma), (3) Ocean-Continent Transition (OCT) formation with mantle exhumation and MORB-type magmatism in Middle-Late Jurassic (165-154 Ma), (4) Break-up and mature oceanic spreading mostly in Late Jurassic (154-145 Ma). Spreading was slow to ultra-slow (max. 22 mm/yr, full rate) and decreased to ~ 5 mm/yr after 145 Ma while completely ceasing at about 130 Ma due to motion of Iberia relative to Europe during the opening of the North Atlantic. The final width of the PL mature ("true") oceanic crust reached a maximum of 250 km along a NW-SE transect between Europe and northwestern Adria. Plate convergence along that same transect reached 680 km since 84 Ma (420 km between 84-35 Ma, 260 km between 35-0 Ma), which exceeds largely the width of the ocean. We suggest that at least 63 % of the subducted and accreted material was highly thinned continental lithosphere and most of the Alpine Tethys units exhumed today derived from OCT zones. Our work highlights the significant proportion of distal rifted continental margins involved in subduction and exhumation processes and provides quantitative estimates for future geodynamic modelling and a better understanding of the Alpine Orogeny.

**1 Introduction**

Over the last decades, new concepts on rifting processes, subduction initiation, depth and exhumation of high-pressure rocks during subduction emerged. The Alps are a natural laboratory to test ideas as it preserved a detailed record of continental margin and ophiolite evolution from the Alpine Tethys that were accreted, or subducted to ultra-high pressure and later exhumed at the surface (Froitzheim and Eberli, 1990; Froitzheim and Manatschal, 1996; Bernoulli and Jenkyns, 2009; Mohn et al., 2010; Beltrando et al., 2014; Masini et al., 2014). Studies of hyper-extended magma-poor type of continental margins such as in the Alps have shed light on the tectonic complexities of distal and ocean-continental transition zones, along which various types of rocks such as granitoid basement within continental allochthons, serpentinized mantle rocks, gabbros and basalts are put in contact by major detachment faults (e.g. Florineth and Froitzheim, 1994; Müntener and Hermann, 2001; Ferrando et al., 2004; Manatschal, 2004; Manatschal and Müntener, 2009; Epin et al., 2019). These zones of inherited weakness and thermal anomalies may represent ideal candidates for localization of subduction initiation when plate motion becomes convergent (Beltrando et al., 2010; Tugend et al., 2014; Stern and Gerya, 2018; Kiss et al., 2020; Zhou et al., 2020).

Moreover, the Alpine chain is enigmatic due to its very arcuate plate boundaries and switch of subduction polarity along strike (Figure 1), lack of a well-developed magmatic arc (e.g. McCarthy et al., 2018) and episodes of slab break-off (e.g. Wortel and Spakman, 2000; Handy et al., 2015). Seismic tomography models beneath the Alps are thus difficult to interpret (see Kästle et al., 2020 for a recent review). Thermo-mechanical modelling is a key tool to gain a better understanding of orogenic processes in the Alps (e.g. Gerya et al., 2002; Yamato et al., 2007; Duretz et al., 2011; Ruh et al., 2015; Reuber et al., 2016; Spakman et al., 2018; Dal Zilio et al., 2020) and potentially a better interpretation of seismic tomography models. These geodynamic models require, however, quantitative input such as estimates of the former extent and thickness of the continental margins and oceanic domains involved in subduction, paleo-location of plate boundaries through time, as well as direction and rate of plate convergence. Kinematic reconstructions are thus crucial, but challenging in such a tectonically complex area. First, regional geological reconstructions need to be globally connected and brought into a plate-mantle reference frame in order to assess mantle-plate-surface interactions through time (e.g. Müller et al., 2019). Second, there is growing debate that maximal burial depth of metamorphic rocks in subduction zones, as estimated from mineral phase equilibria and assuming lithostatic pressure, may be overestimated due to local overpressure and change of tectonic regime (Ford et al., 2006; Schmalholz and Podladchikov, 2013; Pleuger and Podladchikov, 2014; Reuber et al., 2016; Yamato and Brun, 2017; Moulas et al., 2019). This has significant implications for regional tectonic reconstructions that use an inferred amount of subduction based on peak depths of (ultra) high-pressure metamorphism (e.g. Schmid et al., 1996; Handy et al., 2010, 2015) and for conceptual models of exhumation of high-pressure rocks (e.g. Brun and Faccenna, 2008).

The past extent and size of the Piemont-Liguria (PL) Ocean and its rifted margins, prior to the formation of the Alps, remain poorly constrained. Studies based on the age range of mafic rocks derived from the PL Ocean in the Alps and inferring ultra-

slow spreading rates, propose a final extent ranging between 300 km (Li et al., 2013; Manzotti et al., 2014) and 500 km (Froitzheim and Manatschal, 1996; their Figure 3) for the oceanic domain, but lack kinematic constraints. Some kinematic models provide quantitative constraints on the amount of plate divergence, for example 450 km between Iberia-Adria to 675 km between Europe-Adria in Vissers et al. (2013; their Figure 7), and from c. 500 km between Europe-Adria to c. 900 km between Sardinia-Adria in Handy et al. (2010; their Figure 8b). However, those estimates do not distinguish between extended continental and oceanic domains. Thus, the aim of this paper is to provide robust kinematic constraints and address the question of how wide the PL Ocean and its rifted margin were, which is crucial to understand slab pull forces, rollback and exhumation of high pressure rock in the Alps. We furthermore discuss the style of rifting and extent of hyper-extended rifted margins versus mature oceanic domain (i.e. "true" oceanic, not transitional crust) that went into subduction during the formation of the Western-Central Alps. For this, we present one possible kinematic scenario for this area, based on a compilation of previously published models and an updated model for the past motion of Corsica-Sardinia. This regional reconstruction goes back to Triassic time and is implemented within the recent global plate model of Müller et al. (2019). Furthermore, we test this kinematic scenario for geodynamic consistency with thermo-mechanical modelling of the rifting phase and compare our results with existing geological records from the Alps.

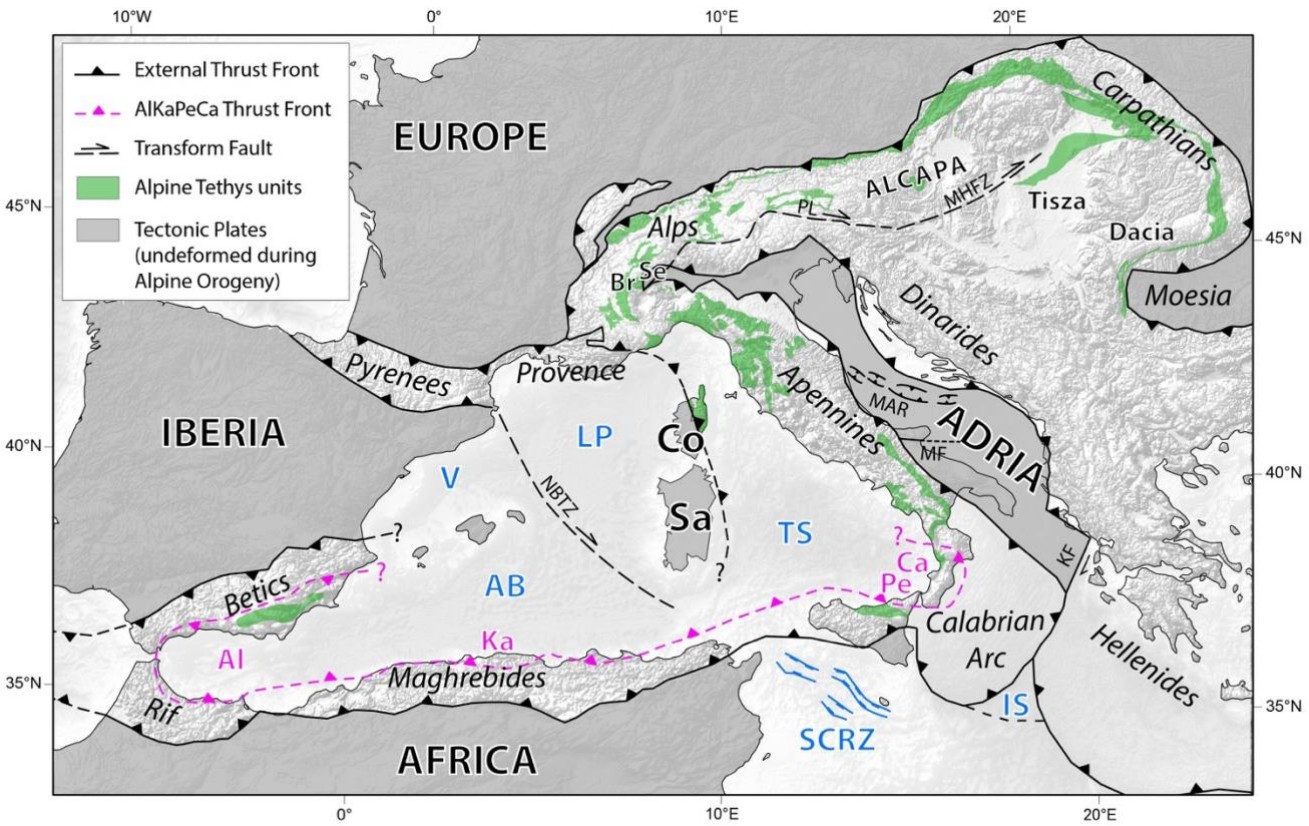

80

**Figure 1. Simplified tectonic map of the Western-Central Mediterranean (modified after Le Breton et al., 2017; Michard et al., 2006; Guerrera et al., 2019; Schmid et al., 2020). Location of Mattinata Fault (MF) from (Argnani et al., 2009). Background topographic-bathymetric map from ETOPO1 model (Amante and Eakins, 2009). Abbr.: AB: Algerian Basin, Al - Alboran, AlCaPa - Alpine-Carpathians-Pannonian Mega-Unit, Br - Briançonnais, Ca - Calabria, Co - Corsica, V – Gulf of Valencia, Ka - Kabylides, IS - Ionian**

85 **Basin, KF: Kefalonia Fault; LP - Liguro-Provençal Basin, MAR Mid-Adriatic Ridge, MHFZ: Mid-Hungarian Fault Zone, MT: Mattinata Fault; NBTZ - North Balearic Transform Zone, Pe - Peloritani, PL - Periadriatic Line, Sa - Sardinia, SCRZ - Sicily Channel Rift Zone, Se - Sesia, TS - Tyrrhenian Sea.**

## 2 Geological Setting

The Alpine-Mediterranean belt results from a complex tectonic evolution that involved the closure of two oceans (Alpine

90 Tethys and Neo-Tethys) and subduction-collision of several continental domains (Briançonnais, Sesia, AlCaPa, Tisza, Dacia, Figure 1) between two major plates Eurasia and Africa that have been converging since Cretaceous time (e.g. Dewey et al., 1989). The Adriatic plate (Adria) is a key player due to its central position between Eurasia and Africa. Today, Adria is surrounded by orogens; the Alps to the north where Adria is the upper indenting plate, the Dinarides to the east and the Apennines to the west where Adria is the subducting lower plate (Figure 1). It is bounded to the south by the accretionary

95 prism of the Calabrian Arc and the Kefalonia Fault (KF on Figure 1).

The Alps contain the remains of two Jurassic- to Cretaceous-age basins (Piemont-Liguria and Valais) together referred to as the Alpine Tethys (e.g., Stampfli, 1998; Figure 1), or, more recently, also as "Alpine Atlantic" to emphasize its kinematic link with the Atlantic Ocean (Gawlick and Missoni, 2019). Here, however, we adopt the well-established term Alpine Tethys. In Cretaceous and Cenozoic times, the former Adriatic continental margin was accreted to the upper plate of the Alpine Orogen, represented today by the Austroalpine (part of "ALCAPA", Figure 1) and Southern Alpine units (south of the Periadriatic Line, Figure 1). This upper plate also contains relics of an older ocean, the Neo-Tethys (also known as Meliata-Maliac-Vardar ocean; Channell and Kozur, 1997; Schmid et al., 2008, 2020). The closure of Neo-Tethys, whose remnants are abundantly found in the Dinarides (Vardar Ophiolites), initiated in the Middle to Late Jurassic (age of metamorphic soles; Maffione and van Hinsbergen, 2018), coevally with rifting and spreading of the PL Ocean (Stampfli et al., 1998; Schmid et al., 2008). In the following sections we will focus on the evolution of the Alpine Tethys and the Western-Central Alps. We note that remnants of the PL Ocean are also preserved in Miocene-to-Recent nappes of the Apennines (including Calabria), the Betic Cordillera and Rif (southern Spain and northern Morocco), and the Maghrebides (northern Africa), as shown in Figure 1 (AlKaPeCa units, Bouillin et al., 1986). Those nappes were emplaced during fast rollback of the subduction zone in the Cenozoic (e.g. Faccenna et al., 2001; Michard et al., 2002; Rosenbaum and Lister, 2004), triggering upper-plate extension and the opening of the Western Mediterranean basins (Algerian Basin, Gulf of Valencia, Liguro-Provencal Basin and Tyrrhenian Sea; Figure 1).

## 2.1 Opening of the Alpine Tethys

### 2.1.1 The Piemont-Liguria Basin

Tectono-stratigraphic, petrological, geochemical and geochronological constraints from the Alps indicate that the PL margin represents a fossil example of an hyper-extended magma-poor continental margin, such as the Iberian margin today (e.g. Manatschal and Bernoulli, 1999; Manatschal, 2004; Decarlis et al., 2015). A first phase of rifting of the proximal part of the continental margin, with horst-and-graben structures bounded by high-angle faults, breccia deposits and platform accumulations, took place in Early Jurassic (c. 200-180 Ma; e.g. Lemoine et al., 1986; Froitzheim, 1988; Froitzheim and Eberli, 1990; Conti et al., 1994; Froitzheim and Manatschal, 1996; Masini et al., 2013; Ribes et al., 2019). This was followed by necking of the continental lithosphere, with the development of crustal-scale extensional detachment faults (Mohn et al., 2012; Ribes et al., 2019), and hyper-extension of the distal part of the margin between c. 180 and 165 Ma, characterized by both low and high angle normal faults, detachment of continental allochthons and exhumation of mantle rocks (e.g. Froitzheim and Eberli, 1990; Froitzheim and Manatschal, 1996; Masini et al., 2013; Ribes et al., 2019). Transects across the proximal to distal parts of the former Adriatic margin are well preserved in the Southern Alps and Austroalpine units of the Central-Eastern Alps (Mohn et al., 2012, their Figure 2; Ribes et al., 2019). For example, in the Central Alps, record of the proximal Adriatic margin is observed within the Ortler and Ela units, the necking domain within the Campo-Grosina and Languard units, the hyper-extended domain within the Bernina and Err units, and the exhumed mantle domain within the upper and lower Plata units (Froitzheim and Manatschal, 1996; Mohn et al., 2010; Ribes et al., 2019).

Radiolarite sedimentation started at around 166 ± 1 Ma (Bill et al., 2001) covering both oceanic rocks and adjacent distal margins, which suggests post-rift sedimentation and used to be interpreted as continental breakup around that time. However, (Ribes et al., 2019) showed a diachroneity in the post-rift sedimentation, younging from the proximal to the distal exhumed mantle domain of the passive margin system. Indeed, extension related to final rifting in the most distal part of magma-poor margin along the ocean–continent transition (OCT) often leads to juxtaposition of extremely diverse types of rock such as

slices of continental crust in contact with exhumed serpentinized mantle, pelagic sediments as well as gabbros and basalts (e.g. (Florineth and Froitzheim, 1994; Froitzheim and Manatschal, 1996; Hermann and Müntener, 1996; Marroni et al., 1998; Müntener and Hermann, 2001; Ferrando et al., 2004; Manatschal et al., 2006; Manatschal and Müntener, 2009; Epin et al., 2019; Ribes et al., 2020). Some Alpine Tethys Ophiolites are interpreted as representing former OCT rather than the mature PL Ocean, due to the presence of exhumed mantle and detachment faults associated with tectono-sedimentary breccias

(Manatschal and Müntener, 2009). U-Pb ages on zircons from MORB-type gabbros and plagiogranites from Corsica, the Northern Apennines and the Alps give a range from c. 165 to 140 Ma (Bortolotti and Principi, 2005; Manatschal and Müntener, 2009; their Figure 6 and references therein), dating both the OCT formation and mature oceanic spreading. The juxtaposition of pillow-lavas directly on mantle rocks and the lack of sheeted dykes in the Alpine Tethys Ophiolites (e.g. Barrett and Spooner, 1977; Lagabrielle and Cannat, 1990; Lagabrielle and Lemoine, 1997; Decrausaz et al. 2021) indicate an ultra-slow type of

spreading for the PL Ocean.

Parts of the continental European margin are preserved in the Alps within the Helvetic and Penninic nappes. Another extended continental unit, the Briançonnais, was that detached from Europe during the opening of another basin: the Valais Basin.

### 2.1.2 The Valais Basin

The opening of the Valais Basin separated the Briançonnais continental unit from the European Plate (e.g. Stampfli, 1993). This is supported by the Early Cretaceous tectono-stratigraphic record between the Briançonnais and Valais domains (Florineth and Froitzheim, 1994) and the so-called "Valaisan trilogy" (Trümpy, 1951, 1954; Loprieno et al., 2011) whose oldest stratigraphic unit (Aroley Fm) is dated to Barremian-Aptian (130-110 Ma).

Based on U/Pb zircon dating on metabasic rocks of the Valais domain, Liati et al. (2005) and Liati and Froitzheim (2006) proposed that the Valais Basin was floored by two generations of oceanic crust and resulted from "re-rifting" of oceanic crust that first formed during the opening of the PL Ocean at about 165-155 Ma and secondly during the opening of the Valais Ocean at about 93 Ma. The existence of a Valais "Ocean" was, however, disputed by Masson et al. (2008) who dated the ophiolites of this domain to Early Carboniferous (U/Pb age). Beltrando et al. (2007) showed also evidence for Permian U/Pb

zircon ages with only a few Cretaceous ages (110-100 Ma) in the rims of some zircons, interpreted as a thermal/fluid event related to extension rather than spreading. Characteristics of hyper-extended margin and OCT zone, with mantle exhumation

and limited magmatic activity within the Valais Basin was later described by Beltrando et al. (2012). Moreover, the presence of significant sedimentation pre-dating the onset of the Valaisan trilogy (Loprieno et al., 2011; Beltrando et al., 2012) confirmed the hypothesis of a multi-stage opening, as proposed by Liati et al. (2005).

Following these observations, we conclude that the Valais Basin was not a mature oceanic basin but a re-rifted, hyper-extended continental margin with exhumed mantle and a few magmatic rocks (OCT zone), that rifted away from Europe first during the opening of the PL Basin in Jurassic but mainly during the Cretaceous. Late Aptian-Albian extensional basins, with turbiditic and basinal series, including olistoliths and breccias, associated with sinistral fault zones are also recorded in Provence (e.g. Joseph et al., 1987; Montenat et al., 2004) and in the Pyrenees (e.g. Debroas, 1990; Choukroune, 1992). The potential kinematic

link between rifting in the Pyrenees-Provence area and the Valais Basin in the Cretaceous is discussed in Section 3.

## 2.2 Closure of the Alpine Tethys

Following tectono-stratigraphic and geochronological studies on syn-orogenic sediments, high-pressure metamorphic rocks and thrust sheets in the Alps (e.g. Froitzheim et al., 1996; Stampfli et al., 1998; Schmid et al., 2008; Handy et al., 2010, 2015), the closure of the Alpine Tethys can be summarized in three stages:

(1) Nappe stacking of continental units and high-pressure metamorphism was first recorded in the Eastern Alps, indicating an intracontinental subduction zone within Adria (Austroalpine unit, part of "AlCaPa"; Stüwe and Schuster, 2010), which developed possibly along late Jurassic strike-slip faults connected to the western termination of the Neo-Tethys Ocean (Schuster and Frank, 1999; Frank and Schlager, 2006). This phase corresponds to the "Eo-Alpine" Orogeny and lasted between c. 130- 84 Ma (Faupl and Wagreich, 1999), as indicated by both synorogenic clastics (Rossfeld Formation; Faupl and

Tollmann, 1979) and geochronological data on high-pressure metamorphic rocks within the Austroalpine units of the Eastern Alps (e.g. Thöni, 2006; Manzotti et al., 2014, their Figure 5 and references therein). Regional scale sinistral strike-slip faults offsetting Austroalpine units were also active during Cretaceous time and were potentially related to the opening of the North Atlantic and subsequent motion of Iberia relative to Europe (Neubauer et al., 1995; Sieberer & Ortner, 2020).

(2) South/Southeast-directed "Alpine" subduction of the PL Ocean, the Valais Basin and the distal European continental

margin followed from c. 84 to 35 Ma. The subduction initiated at the Adriatic margin (Sesia, c. 85-65 Ma) and progressed from SE to NW across the PL ocean (c. 50-45 Ma), the Briançonnais (c. 45-42 Ma) and Valais Basin (c. 42-35 Ma) as indicated by the tectono-metamorphic evolution of those units (e.g. Rubatto et al., 1998; Manzotti et al., 2014, 2018; Handy et al. 2010). The subduction phase is also dated by syn-orogenic trench fill (flysch) sediments, which began at c. 94-86 Ma in the PL Ocean, c. 70 Ma in the Valais Basin and ended at c. 32-35 Ma (Matter et al., 1980). Relics of this "Alpine" subduction and high-

metamorphic rocks with ages ranging from 84 Ma (Lahondère and Guerrot, 1997) to 35 Ma (Martin et al., 2011) are also found in NE Corsica ("Alpine" Corsica, Figure 1; Molli, 2008). North of Corsica, East-West trending thrusts and folds, associated with syn-tectonic foreland sedimentation also started in the uppermost Santonian (c. 83 Ma) in Provence (e.g. Espurt et al.,

2012) and continue till the Eocene (e.g. Lacombe and Jolivet, 2005; Andreani et al., 2010). This shortening phase in Provence is related to the formation of the Pyrenees to the west, which initiated at about 83 Ma (e.g. Mouthereau et al., 2014).

(3) Collision in the Alps and "Apenninic" rollback subduction of the remaining PL Ocean and Ionian Basin initiated at about 35 Ma. The exact timing of onset of collision in the Alps differs depending on the criteria used. For instance, continental units, such as the above mentioned Briançonnais, entered the trench and were subducted prior to 35 Ma. However, here we distinguish continental subduction, in which rifted and thinned continental lithosphere behaves similarly to oceanic lithosphere, from continental collision, where slab pull is out-weighted by the positive buoyancy of the (less rifted) continental lithosphere

following slab break-off and detachment of the subducted lithosphere. Timing of slab break-off in the Alps is inferred from timing of magmatism along the Periadriatic Line, mainly between 34-28 Ma (Rosenberg, 2004). Indeed, the geochemistry of these magmatic rocks indicates melting of lithospheric mantle, best explained by a slab break-off event (von Blanckenburg and Davies, 1995). Moreover, this time period (35-30 Ma) is also marked by a change in sedimentation from "Flysch" to "Molasse" (Rupelian Lower Marine Molasse, 33.9-28.1 Ma) in the Alpine Foreland Basin (Matter et al., 1980; Sinclair, 1997),

and onset of medium temperature – medium pressure Barrovian-type metamorphism within the orogen (Lepontine dome, Tauern Dome; e.g. Bousquet et al., 2008) attributed to thickening of the European crust (Venediger Duplex formation in the Tauern Window; Scharf et al., 2013b). Thus, 35 Ma appears to be a reasonable time for onset of collision in the Alps, as defined above.

In Provence, Oligocene extensional tectonics started at about 35 Ma in response to rollback of the "Apenninic" subduction

(e.g. Séranne, 1999). In the Carpathians, the eastward continuation of the Alps, eastward retreat of the subduction began in early Miocene time (e.g. Horváth et al., 2006) contemporaneously to collision of the Adriatic Plate with Europe and lateral extrusion in the Eastern Alps (e.g. Ratschbacher et al., 1991; Scharf et al., 2013a). Rollback subduction was compensated by upper-plate extension (Pannonian Basin) and arcuation (Ustaszewski et al., 2008), and involved little convergence between Europe and Adria (Royden and Burchfiel, 1989).


## 3 Kinematic reconstructions of the Alpine-Mediterranean area

### 3.1 Approach

Reconstructing the past motion of the Alpine-Mediterranean area is challenging due to the broad deformation, especially around the Adriatic Plate (Figure 1). Our approach is thus to reconstruct the tectonic deformation that affected the plates during

the Alpine Orogeny, with a focus on Adria and Sardinia-Corsica, to quantify the amount of divergence and convergence between stable parts of each plates, i.e. those not deformed during the Alpine Orogeny (in grey on Figure 1). To do so, we compile existing kinematic reconstructions and geological-geophysical data from the surrounding orogens (Alps, Provence, Apennines, Dinarides) and basins (Western Mediterranean Basins, Ionian Sea; Figure 2; see more details on the method in Le Breton et al., 2017 for Neogene time). The motion of Europe, Iberia and Africa is obtained from previously published

reconstructions that fit conjugate magnetic anomalies in the Atlantic and restore the rifting history of the continental margins. This step is crucial to provide a global frame for the regional reconstructions of the Alpine area. Major tectonic events are dated by syn-tectonic sediments, magmatic and metamorphic rocks, and magnetic anomalies in the Atlantic Ocean (magneto-chronostratigraphy), which allow us to draw tectonic maps at key times of the evolution of the Alpine Tethys (rifting and spreading) and Alpine Orogeny (subduction and collision). We then use GIS software (e.g. ArcGIS) and GPlates

(https://www.gplates.org, Müller et al., 2018) to digitize the tectonic maps, calculate rotation poles when needed and obtain a kinematic model back to 200 Ma. Moreover, GPlates (version 2.2) has the functionality to build plate motion models with continuously evolving topologies for rigid and deforming plates, and to include deformation both within the plates and along their boundaries (Gurnis et al., 2012, 2018; Müller et al., 2019). We are thus able to model the evolution of the deforming Adriatic Plate and the strong arcuation of its boundaries especially during the fast slab rollback and trench retreat in Cenozoic

times (Faccenna et al., 2001; Rosenbaum and Lister, 2004).

The workflow between the geological reconstructions (tectonic maps) and GPlates is based on an iterative approach (Figure 2). The tectonic reconstructions give us a range of possible amount of convergence/divergence between the plates and the kinematic reconstructions with GPlates allow us to visualize misfits in our tectonic reconstructions, i.e. overlaps between stable

parts of the plates, and thus indicate "problematic" areas in our tectonic maps that should be further studied. This approach is a never-ending endeavor as new geological and geophysical data from the Alpine-Mediterranean and the Atlantic are constantly being produced and/or are the subject of debated interpretations. Thus, after a brief description of the existing debates and potential scenarios, we present our current "best-fit" model for the Western Mediterranean-Alpine area and the main assumptions on which it lies. Our regional model for the Western-Central Mediterranean is part of the global plate model

of Müller et al. (2019) and all reconstructions and rotation files are available at https://www.earthbyte.org/webdav/ftp/Data_Collections/Muller_etal_2019_Tectonics/.

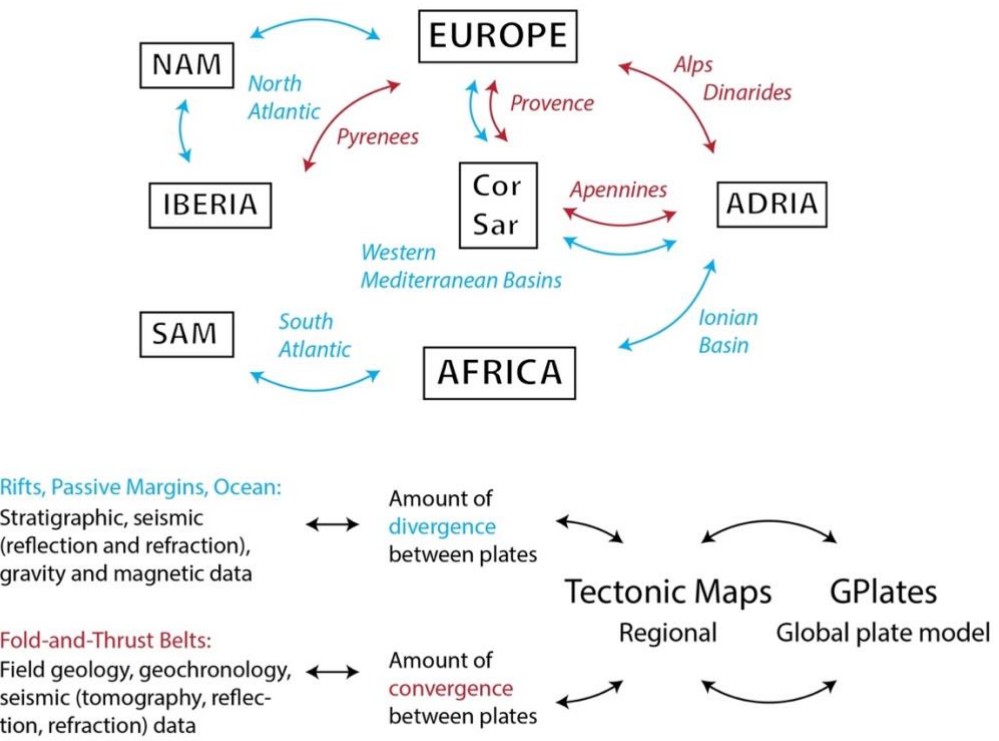

Figure 2. Approach and data used in this study to estimate the amount of deformation between plates in the Western-Central Mediterranean area and reconstruct their past plate motions within a global frame using GPlates. Abbr.: Cor – Corsica, NAM - North America, Sar – Sardinia, SAM - South America.

## 3.2 Existing kinematic scenarios and debates

Due to the complexity of the Alpine-Mediterranean area and based on different datasets and assumptions, several kinematic scenarios back to Mesozoic time have been proposed (e.g. Le Pichon et al., 1988; Dewey et al., 1989; Srivastava et al., 1990; Rosenbaum et al., 2002; Stampfli and Borel, 2002; Handy et al., 2010; Schettino and Turco, 2011; Van Hinsbergen et al., 2020). A key player in the reconstruction of the Western Mediterranean Area is Iberia. Its exact position in Mesozoic time – prior to spreading and clear magnetic anomalies such as the Chron C34 (83.5 Ma; Macchiavelli et al., 2017) – is highly debated depending on the interpretation of paleomagnetic studies on land, magnetic anomalies in the Atlantic and geological-geophysical data from the Pyrenees. Two end-member kinematic scenarios for the Mesozoic motion of Iberia have been proposed: (1) transtensional eastward motion of Iberia versus Europe (e.g. Le Pichon et al., 1988; Olivet, 1996; Stampfli and Borel, 2002; Jammes et al., 2009; Handy et al., 2010; Schettino and Turco, 2011; Barnett-Moore et al., 2018) and (2) scissor-style opening of the Bay of Biscay and rotation of Iberia in Lower Cretaceous time (Sibuet, 2004; Vissers and Meijer, 2012; Vissers et al., 2016a; van Hinsbergen et al., 2020). The main difference is that (1) implies transtension (or strike-slip followed

by orthogonal extension; Jammes et al. 2009), in the Pyrenees in Early Cretaceous, whereas (2) involves subduction and slab break-off in the Pyrenees during that same time period (130-110 Ma).

Another discussed topic is the potential kinematic link between the opening of the Bay of Biscay, the motion of Iberia and the opening of the Valais Basin along a major transtensional zone, which would follow the abovementioned end-member (1) for
Iberia (e.g. Frisch, 1979; Stampfli, 1993; Stampfli and Borel, 2002; Handy et al., 2010). This kinematic link is mostly based on the synchronous time of opening of the Bay of Biscay (Tugend et al., 2014), sinistral transtensional deformation in the Pyrenees (Peybernés and Souquet, 1984; Choukroune, 1992; Oliva-Urcia et al., 2011; Canérot, 2017), in Provence (Bestani et al., 2015 and references therein), and the main opening phase of the Valais Basin (section 2.1.2) in Early Cretaceous time. Handy et al. (2010) furthermore proposed that this sinistral motion was kinematically linked further to the East to the Eo-
Alpine Orogeny (c. 130-84 Ma; section 2.2). This view is however refuted by a model based on paleo-magnetic data from Sardinia, which indicates an independent rotation of Sardinia compared to Iberia (Advokaat et al., 2014). This led to propose an alternative kinematic scenario, in which Sardinia-Corsica remains close to its present-day location with respect to the European Plate, and the opening of the Valais is kinematically independent from the motion of Iberia (Van Hinsbergen et al., 2020).


Complex late-stage rift processes along the magma-poor Iberian margin led to a wide continent-ocean transition zone and to debate on the nature of the M0 magnetic anomaly or "J"-anomaly in the North Atlantic (Bronner et al., 2011; Tucholke and Sibuet, 2012; Nirrengarten et al., 2017) used to constrain the motion of Iberia in a scissor-style way in Early Cretaceous (end-member model (2) for Iberia; Sibuet, 2004; Vissers and Meijer, 2012; van Hinsbergen et al., 2020). Additionally, the
paleomagnetic dataset in Iberia that indicates a counter-clockwise rotation by 35º of Iberia in Early Cretaceous (Vissers and Meijer, 2012; van Hinsbergen et al., 2020) is subject of an active debate (Neres et al., 2012, 2013; see also discussion of Barnett-Moore et al., 2016, 2017 and van Hinsbergen et al., 2017). Most importantly, scissor-style motion of Iberia and the alternative model for Sardinia mentioned above (Advokaat et al., 2014; Van Hinsbergen et al., 2020) imply subduction in the Pyrenees but also of significant part of the PL Ocean between Iberia and Sardinia-Adria between 130-110 Ma (Figure 3 – 1.A).
However, geological records in the Pyrenees, i.e. thick (up to 5 km) Albo-Cenomanian sedimentary fault-related basins (turbidites and breccias) later inverted during the Pyrenean Orogeny, as well as exhumation of (ultra)mafic rocks and Cretaceous alkali magmatism, all lead towards the interpretation of a rifting phase rather than subduction in Early Cretaceous time (e.g. Jammes et al., 2009; Clerc and Lagabrielle, 2014; Masini et al., 2014; Tugend et al., 2014). The transition from divergence to convergence is dated later by inversion of rift faults and syn-inversion growth strata of Late Santonian age (at
around 84 Ma; e.g. McClay et al., 2004; Mouthereau et al., 2014), which fits with onset of Alpine subduction more to the east (section 2.2) and the Late Cretaceous age of syn-orogenic flysch sediments in the Alps (Matter et al., 1980) and Northern Apennines (Marroni et al., 1992).

Figure 3 summarizes what we name this "Iberia-Sardinia Problem" (Figure 3.1) and following the abovementioned geological record of transtension/extension in the Pyrenees and the absence of geological evidence for convergence between Iberia and Sardinia in Early Cretaceous, we favor the kinematic model involving sinistral transtensional motion between Europe and Iberia, linked to the opening of the Valais Basin and Eo-Alpine Orogeny to the east (Figure 3 – 1.B). We emphasize here that this motion is post-145 Ma (Tugend et al., 2014; Barnett-Moore et al., 2018) and thus does not influence the opening of the PL Basin, which occurs earlier in the Jurassic (section 2.1.1). The implications of a possible earlier phase of Jurassic rifting in the Pyrenees, as suggested from Jurassic ages of pyroxenites (Henry et al., 1998) and shear zones at Cap de Creus (Vissers et al., 2016b), is discussed in section 5.2.

**Figure 3 (next page). 1.** Contrasting paleo-reconstructions of Iberia and Sardinia-Corsica at 145 Ma (blue), 130 Ma (green), 110 Ma (orange) and 83 Ma (purple) relative to Eurasia: **1.A.** model in accordance with paleo-magnetic data (from van Hinsbergen et al. 2020) but implying more than 500 km convergence between Iberia and Sardinia in Lower Cretaceous (130 -110 Ma), of which there is no geological record; and **1.B.** alternative model considering Sardinia-Corsica as part of Iberian plate and implying up to 700 km strike-slip (transtensional) motion along the North Pyrenean Fault. In this model, the convergence relative to Eurasia is accommodated more to the East, along the Eo-Alpine Orogeny, as proposed *e.g.* by Handy et al. (2010). **2.** Contrasting paleo-reconstructions of Adria following model 1.B., at 200 Ma, prior to the opening of the Piemont-Liguria Basin. The left model (2.A.) shows the "overlap" problem between northern and Sardinia-Corsica when reconstructing Adria as a single plate. This is solved in the right model (2.B.) by subdividing Adria in two plates along the Mattinata Fault (MF, Figure 1) as proposed by Schettino and Turco (2011). See text for further explanations. Abbr.: Al - Alboran, Ca - Calabria, Co - Corsica, Ka - Kabylides, Pe - Peloritani, Sa – Sardinia.

## 1 - The Iberia-Sardinia Problem

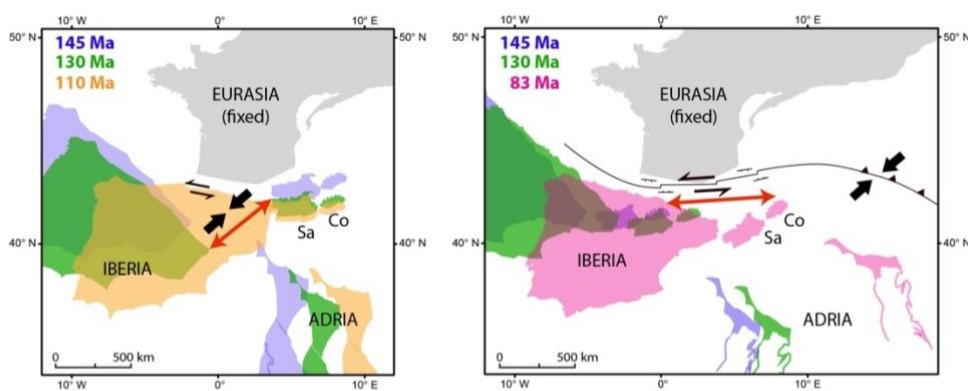

**1.A. Sardinia-Corsica as part of European plate:**
Model for Iberia and Sardinia in accordance with
paleo-magnetic data (from van Hinsbergen et al. 2020)
=> implies convergence between Iberia and Sardinia
(500-700 km depending on kinematic model used for
Iberia) mostly in Lower Cretaceous (130-100 Ma)

**1.B. Sardinia-Corsica as part of Iberian plate:**
Alternative model where motion of Iberia-Sardinia-Corsica
accommodated by sinistral motion (up to 700 km between
145 and 84 Ma) and kinematically linked with opening of the
Valais Basin and intra-continental Eo-Alpine orogeny within
Adria to the East (as proposed *e.g.* by Handy et al., 2010)

## 2 - The Adria Problem

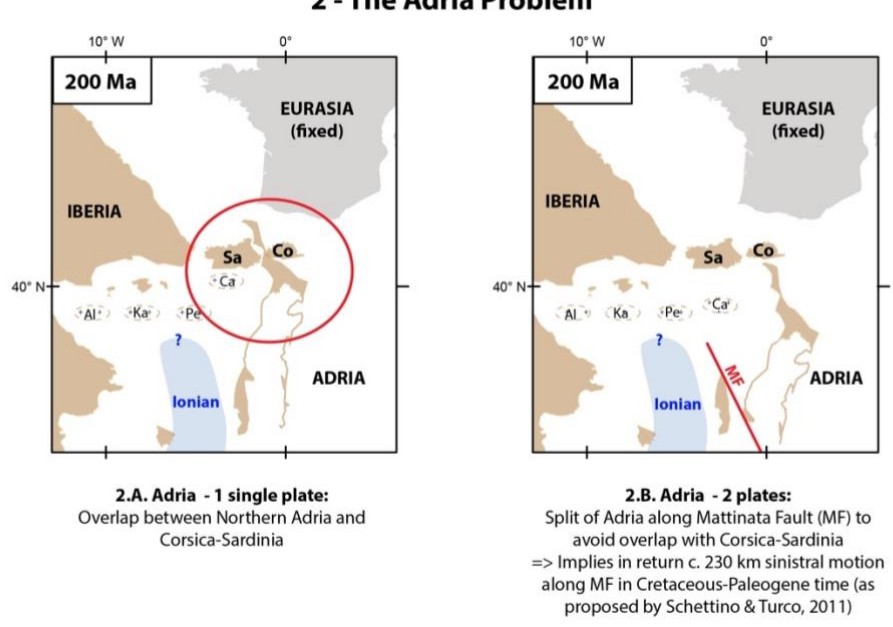

**2.A. Adria - 1 single plate:**
Overlap between Northern Adria and
Corsica-Sardinia

**2.B. Adria - 2 plates:**
Split of Adria along Mattinata Fault (MF) to
avoid overlap with Corsica-Sardinia
=> Implies in return c. 230 km sinistral motion
along MF in Cretaceous-Paleogene time (as
proposed by Schettino & Turco, 2011)

### 3.3 Reconstructions of the Atlantic (Europe, Africa, Iberia)

The motion of Europe, Africa and Iberia is well constrained by the fit of conjugate magnetic anomalies that formed during
325 sea-floor spreading along the mid-Atlantic ridge (Müller et al., 1997, 1999 for the fit of South and North Americas vs Africa;

Gaina et al., 2002 and Schettino and Scotese, 2005 for North America vs. Europe and Iberia, respectively). For the pre-breakup phase, the global plate model of Müller et al. (2019) is built on the reconstructions of Kneller et al. (2012) for the Central Atlantic, Heine et al. (2013) for the South Atlantic and Barnett-Moore et al. (2018) for the North Atlantic. Those models are based on structural restoration of the conjugate Atlantic margins using the rift basins architecture and present-day crustal thickness to approximate the amount and direction of extension, and also of intracontinental rift basins in Africa and South America (Heine et al., 2013). We note that the model used for Iberia (Barnett-Moore et al., 2018) follows the favored model of transtensional motion rather than subduction between Iberia and Europe in Early Cretaceous (see section 2.3).

**3.4 Reconstructions of Adria, Sardinia-Corsica**

Another key player in the geodynamic evolution of the Alpine-Mediterranean area is the past motion of the Adriatic Plate (Adria). In most models, Adria is considered as a "promontory" of Africa (Channell et al., 1979) and thus is restored as moving together with Africa, since the end of opening of the Ionian Basin. The kinematics and opening of this basin are difficult to constrain, as most of it was lost to subduction and the small remaining portion is covered by the Calabrian accretionary prism today (Figure 1). Seismic refraction and reflection data indicate that the Ionian basin is floored by oceanic crust (Dannowski et al., 2019) and magnetic anomalies indicate an early Mesozoic age of spreading (220-230 Ma; Speranza et al., 2012). Handy et al. (2010), using tectonic reconstructions of the Alps, proposed an independent motion of the Adriatic Plate since 84 Ma, which in return would imply significant deformation in the Ionian Basin. However, there is no geological evidence for such deformation within this basin. Only the Sicily Channel Rift Zone (SCRZ, Figure 1) records an extensional event and accommodates some divergence between Africa and Adria, but only in Neogene time and of limited amount (30 km minimum along the Pantelleria Rift, CROP M25; Civile et al., 2010; Le Breton et al., 2017). Moreover, this extensional phase might not be directly related to plate kinematics but rather to dynamics of the subducted African slab (Argnani, 2009).

The Neogene tectonic evolution of Adria and Sardinia-Corsica is constrained by reconstructing the amount of convergence along fold-and-thrust belts (Apennines, Alps, Dinarides for Adria; Provence for Sardinia-Corsica) and coeval divergence along extensional basins (Liguro-Provencal and Tyrrhenian basins, Sicily Channel Rift Zone) (Le Breton et al., 2017). Those reconstructions show that Adria had a slightly independent motion from Africa and rotated counter-clockwise of about 5° relative to Europe since 20 Ma. As mentioned above, the Ionian Basin between Africa and Adria is oceanic (seismic velocities; Dannowski et al., 2019) and opened in the Triassic (220-230 Ma based on magnetic anomalies; Speranza et al., 2012). Thus, our main assumption is to avoid any significant convergence or divergence between Adria and Africa between the end of spreading (220 Ma) and the Neogene (20 Ma). We note that Tugend et al. (2019) recently suggested a Jurassic phase of opening for the Ionian Basin, which will be discussed in section 5.2.

The motion of the Corsica-Sardinia Block (CSB) in Oligo-Miocene times is associated with the opening of the Liguro-Provencal Basin during the fast retreat of the Apenninic subduction zone (e.g. Faccenna et al., 2001). Syn-rift and post-rift

sediments along the Gulf of Lion date the onset of rifting at about 35 Ma and end of rifting at about 21 Ma (Séranne, 1999;
Jolivet et al., 2015). Paleomagnetic data indicates a clear rotation of Sardinia between 21 and 16 Ma (Speranza et al., 2002;
Gattacceca et al., 2007) which is interpreted as spreading along the Liguro-Provencal Basin. We reconstructed the motion of
Corsica relative to France back to 35 Ma using the amount of extension along the Liguro-Provencal Basin estimated to c. 115
km (c. 60 km rifting and 55 km spreading; Le Breton et al. 2017). This fits very well with the rotation pole and amount of
rotation proposed by Speranza et al. (2002, Table 1) for Sardinia based on paleomagnetic data.


The pre-Oligocene evolution of the CSB and its position with respect to Iberia is debated, as mentioned in section 3.2. Here
we follow the kinematic model where the CSB was part of Iberia and moved to its present-day location during a sinistral,
transtensional motion of Iberia and opening of the Valais Basin in Early Cretaceous time (Figure 3 - 1.B; e.g. Stampfli and
Borel, 2002; Handy et al., 2010). Indeed, the Oligo-Miocene rotation of the CSB during the opening of the Liguro-Provencal
Basin implies the presence of a transform fault between Iberia and Sardinia, the North Balearic Transform Zone (NBTZ; e.g.
van Hinsbergen et al., 2014) but there is no evidence for any shortening between Eastern Spain and Sardinia. Thus, our
approach is to avoid significant divergence-convergence between Iberia and the CSB and therefore to move them together
relative to Europe prior to 35 Ma. We implement a strike-slip motion between Iberia and the CSB along the NBTZ to
accommodate extension during the opening of the Pyrenean Rift System - Valais Basin in Mesozoic time (Table 1) and
convergence in the Pyrenees-Provence belt (c. 155 km between Sardinia and Provence; Bestani et al., 2016). This independent
motion may explain, at least in part, the different rotation of the CSB compared to Iberia indicated by paleomagnetic studies
(Advokaat et al., 2014) without having a significant paleogeographic separation between the two domains.

Moving the CSB with Iberia back in Mesozoic time is also in agreement with Carboniferous-Permian paleo-reconstructions of
Tuscany, Calabria and Corsica that indicate a paleogeographic continuity between these three domains (Molli et al., 2020) and
of the Western Alps that indicate a paleogeographic separation between the External Crystalline Massifs (Helvetic Zone,
Europe) and the Briançonnais Zone (different pre-Triassic basement; Ballèvre et al., 2018). This implies in return significant
strike-slip motion (max. 700 km; Figure 3 – 1.B) between Europe and Iberia-CSB-Briançonnais along the North Pyrenean
Fault and the East Variscan Shear Zone, a Variscan transcurrent shear zone that was reactivated in Mesozoic time to
accommodate the motion of Iberia relative to Europe during the opening of the North Atlantic, as proposed by previous authors
(Choukroune, 1992; Matte, 2001; Stampfli and Borel, 2002; Sibuet, 2004; Guillot and Ménot, 2009; Guillot et al., 2009;
Ballèvre et al., 2018). It also implies significant strike-slip motion (c. 230 km) within the Adriatic Plate to avoid an overlap
between Corsica and Northern Adria back to Triassic time (200 Ma, Figure 3-2). This overlap or "Adria Problem" was already
mentioned by Wortmann et al. (2001), Stampfli and Borel (2002), Schettino and Turco (2011), and implemented in
Hosseinpour et al. (2016)'s kinematic model (based on rotation poles from Schettino and Turco, 2011; Table 1), in which they
subdivided the Adriatic Plate in two plates, Northern Adria and Southern Adria (or Apulia), along the Mattinata Fault (MF;
Figure 1 and Figure 3 – 2B) across the Gargano Promontory. Multichannel seismic profiles along the eastward offshore

continuation of this fault shows evidence for strike-slip activity during the Late Cretaceous to Paleogene (Argnani et al., 2009). The amount of displacement (c. 230 km) is however most likely over-estimated in our model. More tectonic deformation are

observed within the Adriatic Sea to the north, along the so-called Mid-Adriatic Ridge (Scisciani and Calamita, 2009; previously named Central Adriatic Deformation Belt by Argnani and Frugoni, 1997). This ridge formed by transpressional reactivation of pre-existing rift-related Mesozoic structures, mostly in Plio-Pleistocene time as shown along seismic profile CROP M15 and present-day GPS data (D'Agostino et al., 2008). An earlier inversion phase is observed by lateral thickness variations and unconformities above the Albian-Aptian reflector (see Figure 6 of Scisciani and Calamita, 2009), especially during the

Paleogene-Miocene succession (Figures 3 and 8 of Scisciani and Calamita, 2009). Moreover, strike-slip deformation and reactivation of Jurassic rift-related structures in Cretaceous-Eocene time have been reported from the Southern Alps (Castellarin et al., 2006) to the Central Apennines (Cipriani and Bottini, 2019a, 2019b). It is thus very likely that the entire Adriatic Plate actually deformed in a more diffuse way along a series of strike-slip faults in Late Cretaceous-Paleogene time. However, it is extremely challenging to constrain the exact magnitude and timing of motion along those faults and would

require further investigations in the future. For the sake of simplicity, we follow the model of Schettino and Turco (2011) and Hosseinpour et al. (2016) and divide the Adriatic Plate into two plates along a single major strike-slip fault. Our model can thus be viewed as an end-member model that implies a maximum amount of sinistral strike-slip displacement along the Mattinata Fault and between Europe and Iberia-CSB, but fits best the paleo-geography of Adria-Sardinia-Calabria (Molli et al., 2020) and subduction record of the PL Basin (section 2). Sinistral motion along the Mattinata Fault (c. 230 km) to "restore"

Adria to its present-day configuration is accommodated progressively during the counter-clockwise motion of Africa relative to Europe, between c. 100-40 Ma. Thus, it does not affect the opening kinematics of the PL Basin occurring earlier in the Jurassic (section 2), which is the focus of this paper. The implication of this strike-slip motion within Adria is discussed in section 5.3 in terms of plate convergence during the Alpine Orogeny.

**Table 1. Total reconstruction rotations used in this study for Sardinia-Corsica and Adria.**

| Age (Ma) | Latitude | Longitude | Angle | Moving vs Relative Plate | Reference |
|---|---|---|---|---|---|
| **Corsica-Sardinia** | | | | | |
| 0.0 | 0.0 | 0.0 | 0.0 | COR-SAR vs Europe | |
| 16.0 | 0.0 | 0.0 | 0.0 | COR-SAR vs Europe | Speranza et al. (2002) |
| 21.0 | 43.5 | 9.0 | -23.0 | COR-SAR vs Europe | Speranza et al. (2002) |
| 35.0 | 43.5 | 9.0 | -53.2 | COR-SAR vs Europe | This study (rifting Liguro-Provençal Basin) |
| 83.0 | 42.7458 | 8.0933 | -56.8041 | COR-SAR vs Europe | This study (shortening Provence) |
| 83.0 | -44.7583 | -168.868 | 47.0239 | COR-SAR vs Iberia | Crossover (change in relative plate) |
| 93.0 | -44.4487 | -168.6366 | 47.77 | COR-SAR vs Iberia | This study (end opening Valais Basin) |
| 144.7 | -45.5809 | -166.3455 | 42.7547 | COR-SAR vs Iberia | This study (opening Valais Basin) |
| 250.0 | -45.5809 | -166.3455 | 42.7547 | COR-SAR vs Iberia | |
| **Northern Adriatic Plate (Adria)** | | | | | |
| 0.0 | 0.0 | 0.0 | 0.0 | Adria vs Apulia | |
| 40.1 | 0.0 | 0.0 | 0.0 | Adria vs Apulia | Schettino and Turco (2011) |
| 47.0 | -26.5 | -166.58 | 1.0 | Adria vs Apulia | Schettino and Turco (2011) |

| Age (Ma) | Latitude | Longitude | Angle | Moving vs Relative Plate | Reference |
|---|---|---|---|---|---|
| 60.0 | -26.5 | -166.58 | 1.0 | Adria vs Apulia | Schettino and Turco (2011) |
| 70.0 | -26.5 | -166.58 | 2.0 | Adria vs Apulia | Schettino and Turco (2011) |
| 100.0 | -26.5 | -166.58 | 7.73 | Adria vs Apulia | Schettino and Turco (2011) |
| 250.0 | -26.5 | -166.58 | 7.73 | Adria vs Apulia | Schettino and Turco (2011) |
| **Southern Adriatic Plate (Apulia)** | | | | | |
| 0.0 | 0.0 | 0.0 | 0.0 | Apulia vs Europe | |
| 20.0 | 38.2028 | -3.1628 | -5.3474 | Apulia vs Europe | Le Breton et al. (2017) |
| 20.0 | 50.5636 | 21.1076 | -2.9022 | Apulia vs Africa | Crossover (change in relative plate) |
| 220.0 | 50.5636 | 21.1076 | -2.9022 | Apulia vs Africa | Africa moving with Africa |
| 220.0 | 49.4645 | -0.0993 | -68.068 | Apulia vs Europe | Crossover (change in relative plate) |
| 230.0 | 50.4113 | -1.4579 | -68.534 | Apulia vs Europe | This study (opening Ionian Basin, timing from |
| 250.0 | 50.4113 | -1.4579 | -68.534 | Apulia vs Europe | Speranza et al., 2012) |

## 3.5 Reconstruction of the AlKaPeCa units

West of Adria, remnants of the western Mediterranean subduction zone are found today in southern Spain (**Al**boran), northern Africa (**Ka**bylia), Sicily (**Pe**loritani) and southern Italy (**Ca**labria; Figure 1), together referred to as the AlKaPeCa units (Bouillin et al., 1986; Guerrera et al., 1993; Michard et al., 2002, 2006). These units are continentally derived, far-travelled nappes that acquire their present-day position during the fast rollback of the Gibraltar and Calabrian subduction zones in Oligo-Miocene time. For this time period, we used the tectonic reconstructions of Van Hinsbergen et al. (2014) which is based on a compilation of shortening and extension estimates from the Atlas mountains, the Betic Cordillera, the Alboran domain, the Algerian Basin, the Gulf of Valencia and the Tyrrhenian Sea (Figure 1).

The similarity of the Triassic-Liassic stratigraphy and the Variscan basement between the AlKaPeCa units and coeval Tuscan and Sesia domains on the Adriatic Plate suggest that the AlKaPeCa units were originally derived from a microplate that rifted away from the Iberian-CSB-Africa-Adria continental margins during the opening of the PL Basin in Jurassic time (Michard et al., 2002, 2006; Molli, 2008, 2020). Similar "Adria-like" features are found in Corsica within the Nebbio units (Molli, 2008) which also suggests a possible continuity of this continental microplate to the NE, between Corsica and Adria towards Sesia (Michard et al., 2002, 2006) and thus supports our reconstructions of the CSB (section 3.3.). This microplate has various names in the literature (e.g. "Alcapecia" in Handy et al. 2010; "Mesomediterranean terrane" in Guerrera et al., 2019). To simplify, we refer only to "AlKaPeCa" here and want to point out that the exact extent of this microplate in Mesozoic times is very uncertain (hence the question marks on our tectonic maps, Figure 4). This microplate is important for the geodynamic evolution of this region as it separated a northern and a southern branch of the PL Ocean, supported by the presence of Tethys-derived ophiolite units in the Betic Cordillera (southern Spain) and in the Maghrebide Belt (northern Africa-Sicily), respectively (Michard et al., 2002, 2014; Guerrera et al., 2019). The polarity of subduction zone(s) in this area is debated (e.g. Jolivet et al., 1998; Argnani, 2012; Guerrera et al., 2019). In our tectonic reconstructions, we follow a "two-subduction" model that implies

first a southeast-dipping ("Alpine") subduction of the northern branch of the PL ocean in Late Cretaceous-Eocene time followed by a switch of subduction polarity to west-northwest ("Apenninic") subduction of the southern branch of the PL when the microplate entered the subduction zone at ~ 35 Ma (e.g. Michard et al., 2006; Molli, 2008; Molli and Malavieille, 2011).

### 3.6 Reconstruction of the AlCaPa-Tisza-Dacia units

Other tectonically complex and far-travelled nappes are found east of Adria in the Eastern Alps, Carpathians and beneath the Pannonian Basin: the AlCaPa-Tisza-Dacia units. The AlCaPa ("**Al**ps-**Ca**rpathians-**Pa**nnonian Basin") unit is derived from Adria and corresponds to the Austroalpine nappes of the Eastern Alps and the Inner West Carpathians (Schmid et al., 2008; Figure 1). The Tisza and Dacia units, presently beneath the Pannonian Basin, consists of tectonic units with a mixed European and Adriatic affinities (Schmid et al., 2008; Handy et al., 2015). These three tectonic units comprise a long and complex tectonic evolution from the opening and closure of several oceans (Neo-Tethys and part of Alpine Tethys; e.g. Schmid et al., 2008, 2020). During the Neogene, they underwent significant extension and rotation during the fast rollback of the Carpathian subduction zone and upper-plate extension with the opening of the Pannonian Basin (e.g. Ustaszewski et al., 2008 and references therein). Thus, it is extremely challenging to reconstruct their past position and former extent. For the Jurassic, we follow Schmid et al. (2008) and Vissers et al. (2013) who proposed that Tisza-Dacia separated from Europe during the opening of the eastern part of the PL Ocean. This in return implies a strike-slip motion along the Dobrogea Fault Zone NE of Moesia, which remains poorly constrained (question marks on Figure 4). After the opening of the PL Ocean, convergence between Africa and Europe was accommodated within the AlCaPa unit along an intracontinental subduction zone (Eo-Alpine orogen; Stüwe and Schuster, 2010). The accretion of the Austroalpine nappes was complete by c. 84 Ma and the "Alpine" south-dipping subduction of Alpine Tethys initiated (Handy et al. 2010). For the Late Cretaceous-Cenozoic, our model is based on the tectonic reconstructions of Handy et al. (2015) and Ustaszewski et al. (2008). This area remains extremely simplified in our tectonic maps (Figure 4) as this is out of scope of our study; a more detailed review and reconstruction of this area can be found in Schmid et al. (2020) and van Hinsbergen et al. (2020).

## 4 Results

### 4.1 Tectonic maps back to 200 Ma

Following the geological records and kinematic constraints summarized in the past two sections, we constructed a series of tectonic maps of the Alpine-Mediterranean area (Figure 4) for key times covering the opening of the Piemont-Liguria Ocean (200-130 Ma) and the Alpine Orogeny (83-0 Ma). Those maps are a deliberately simplified representation and aim at showing the main plates and plate boundaries through time. They do not show intraplate deformation, nor rifting along the continental margins. Those regional reconstructions are incorporated into the global plate model of Müller et al. (2019). Animations

showing the evolution of the Alpine-Mediterranean back to 250 Ma and age-grids of the PL Ocean are available at:

https://www.earthbyte.org/webdav/ftp/Data_Collections/Muller_etal_2019_Tectonics/.

The divergent motion between the two major plates, Europe vs. Africa (Adria), in the Jurassic (blue vectors on Figure 4, obtained with the "flowline" tool on GPlates) shows different stages of opening of the PL Basin. First, a NNW-SSE directed (relative to Europe fixed) divergence between 200 and 164.7 Ma (Early-Middle Jurassic) followed by an oblique E-W directed

motion until 154 Ma (Middle-Late Jurassic). Then the main divergent phase occurs between 154 and 145 Ma (Late Jurassic) in a NW-SE direction. At 145 Ma (Early Cretaceous), Iberia and CSB start moving relative to Europe due to the opening of the North Atlantic. This oblique motion significantly decreases the divergence in the PL Basin, which totally ends at about 130 Ma. The different stages of opening of the PL Basin (200, 164.7, 154, 145 and 130 Ma) are derived from the stages of opening of the Central Atlantic (Kneller et al., 2012) and the North Atlantic (Barnett-Moore et al., 2018).


Based on the review of geological events given in sections 2 and 3, we can summarize the main stages of convergence (right panel of Figure 4) as follows. From 130 Ma, the motion between Europe, Iberia and Africa (Adria) is mostly strike-slip. Subduction of parts of the PL Ocean to the east starts in response to the eastward motion of Iberia-CSB and is kinematically linked with the Eo-Alpine Orogeny within the AlCaPa unit. At 83 Ma, the oblique motion of Iberia and Africa relative to

Europe becomes more orthogonal. The "Alpine" south-directed subduction starts, with Adria and the accreted AlCaPa unit as part of the upper plate. The subduction progresses from SE to NW, with the progressive subduction of Sesia, the PL ocean, the Briançonnais, the Valais Basin and finally the European margin. At 35 Ma, collision starts in the Alps while subduction continues more to the East along the Carpathians and switches to the "Apenninic" northwest-directed subduction in the Western Mediterranean. Finally, since 35-20 Ma, fast rollback of the Gibraltar and Calabria subduction zones in the western

Mediterranean, of the Hellenic subduction in the eastern Mediterranean and of the Carpathians is accompanied by indentation of the Adriatic Plate into Europe, lateral extrusion of the Eastern Alps towards the east and upper-plate extension and opening of the Pannonian Basin, the Western Mediterranean basins, and Aegean Sea, shaping the complex arcuate and broadly deformed plate boundaries that we can observe today.

**Figure 4 (next page): Simplified tectonic maps of the Alpine-Mediterranean area from 200 to 130 Ma (Divergent phase – Left panel) and from 83 to 0 Ma (Convergent phase – Right panel), relative to Eurasia fixed, showing the main plate boundaries and divergence between Europe-Corsica and Adria (blue vectors). Amount of divergence along a 2D transect between Corsica and Adria is indicated in red. Note these maps do not show intraplate deformation and rifting along the continental margins but the divergence between plates when considered rigid. This study focuses on the area between Corsica-Europe and Adria, as the exact paleo-location of the**

**AlKaPeCa units to the west and AlCaPa-Tisza-Dacia units to the east is poorly constrained. Present-day coastlines are represented for orientation. Abbr.: AB: Algerian Basin, Al - Alboran, AlCaPa - Alpine-Carpathians-Pannonian Unit, Br - Briançonnais, Ca - Calabria, Co - Corsica, GF – Giudicarie Fault, Ka - Kabylides, LP - Liguro-Provençal Basin, NBTZ - North Balearic Transform Zone, Pe - Peloritani, PF - Periadriatic Fault, PL – Piemont-Liguria Basin and Ocean, Sa - Sardinia, SCRZ - Sicily Channel Rift Zone, Se - Sesia, Tu – Tuscany, T - Tyrrhenian Sea, V – Gulf of Valencia, Va – Valais Basin.**


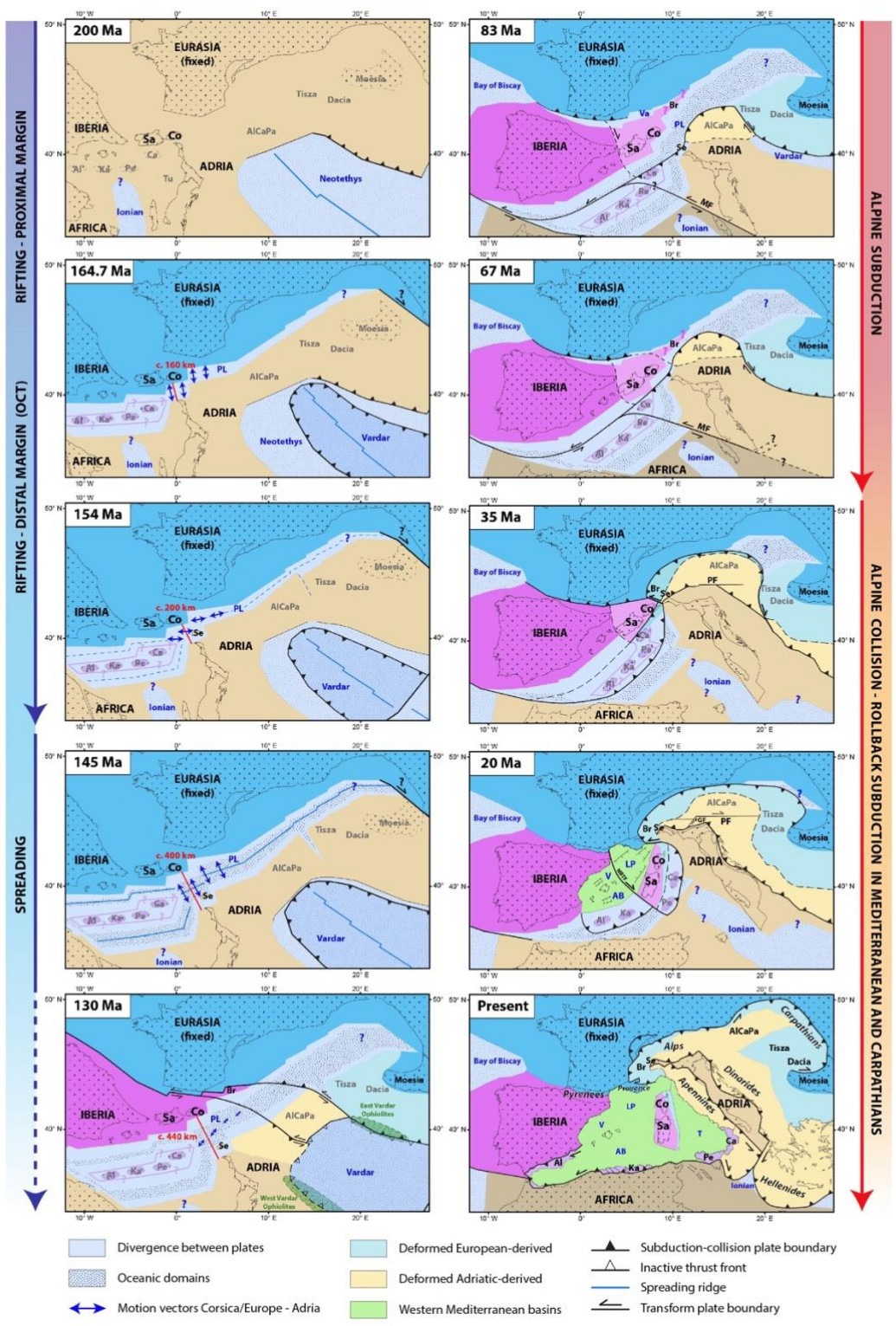

## 4.2 Velocities of rifting and spreading along the PL Basin

We estimate the velocities of plate divergence during the opening of the PL basin (Figure 5) using the motion vectors between Europe (Corsica-Briançonnais) and Adria (blue vectors on Figure 4). Full-rates of opening are first very slow (c. 4 mm/yr from SW to NE) between 200-164.7 Ma. They accelerate first to c. 15 mm/yr (16 to 14 mm/yr from SW to NE) between 164.7 -154 Ma and secondly to c. 22 mm/yr (ranging from 24 to 21 mm/yr from SW to NE) between 154-145 Ma. They decrease significantly at 145 Ma down to c. 5 mm/yr until 130 Ma. We assess an uncertainty of ± 5 km in measuring the vector length, which - once divided by the shortest time period (154-145 Ma) - leads to an error bar up to ± 0.5 mm/yr in the calculated rates. The slight increase in rate from SW (Corsica-Adria) to NE (Briançonnais-Adria) indicates a triangular-shape opening of the PL Basin.

The first slow phase (c. 4 mm/yr) coincides in time with the opening of the Central Atlantic (Kneller et al., 2012) and tectono-stratigraphic record of rifting along the PL Basin (Figure 5; section 2.1.1; e.g. Froitzheim and Eberli, 1990). The first acceleration at c. 165 Ma fits with an increase in spreading rate in the Central Atlantic (from ultra-slow to slow; Kneller et al., 2012) and onset of deep-sea sedimentation, mantle exhumation and MORB-type magmatism in the PL Basin (Figure 5; section 2.1.1; e.g. Bill et al., 2001; Manatschal and Müntener, 2009). The fastest rates of divergence are found on a very short time period of 9 Myr (154-145 Ma) and are in the range of slow to ultra-slow spreading ridge (< 20 mm/yr; Dick et al., 2003). The drop at 145 Ma coincides with a decrease in spreading rates in the Central Atlantic (Kneller et al., 2012) and the opening of the North Atlantic (Barnett-Moore et al., 2018).

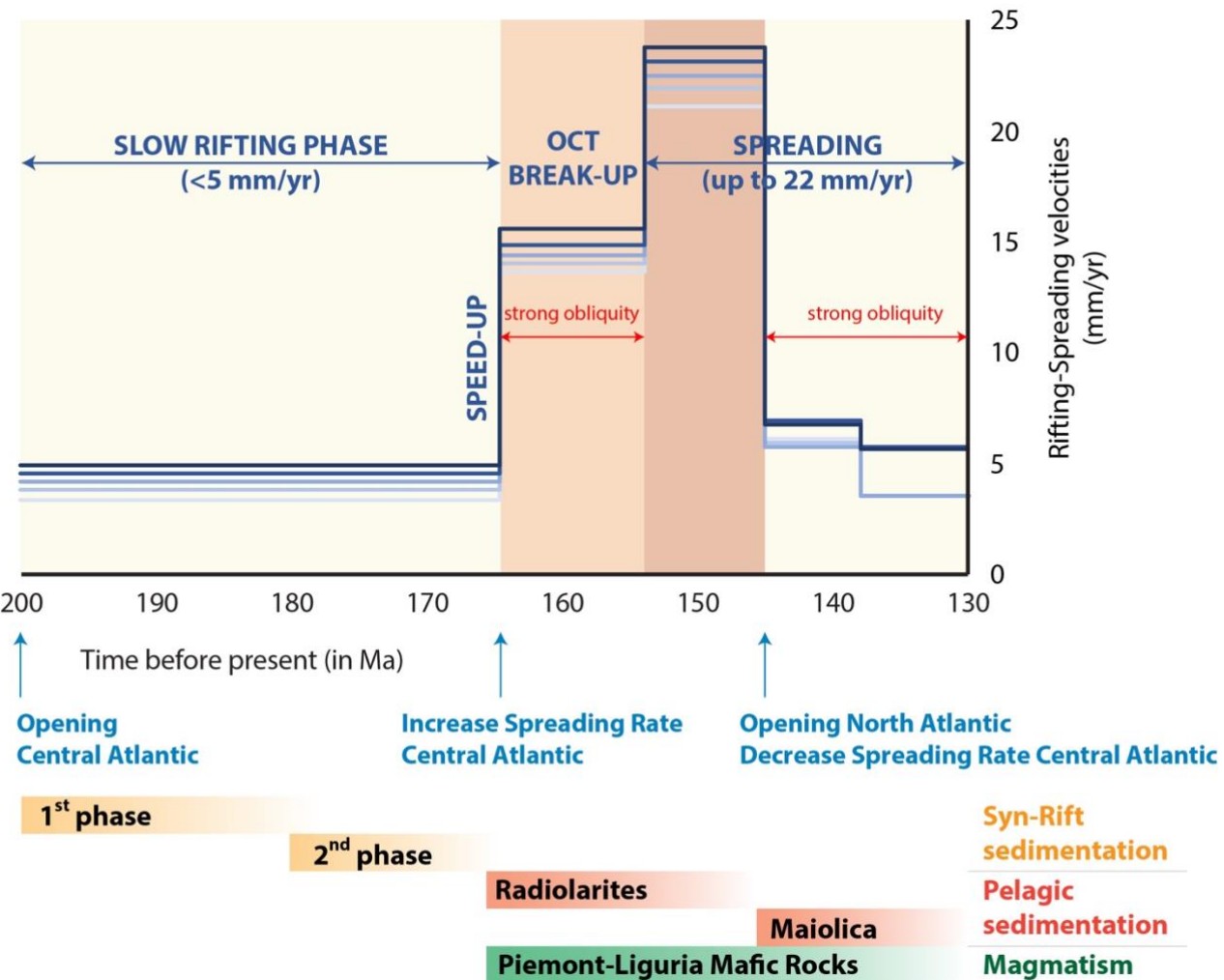

**Figure 5. Velocities (full-rate) of rifting and spreading along the PL basin between Europe (Corsica-Briançonnais) and Adria (blue vectors on Figure 4) in Jurassic-Early Cretaceous time. Timing of syn-rift sedimentation from Froitzheim and Eberli (1990), deep-sea pelagic sedimentation from Bill et al. (2001) and Ferrando et al. (2004), and U/Pb ages on zircons from mafic rocks of the PL ocean from Manatschal and Müntener (2009 and references therein). Note the slight increase in velocities from NE to SW (light to dark blue curves) and the strong acceleration at 164.7 and 154 Ma for all curves.**

### 4.3. Thinning of the continental margins

The total amount of divergence along a NW-SE directed transect between Corsica and Northern Adria (in red on Figure 4) equals c. 440 km and can be subdivided into 160 km between 200-164.7 Ma, 40 km between 164.7-154 Ma, 200 km between 154-145 Ma and 40 km between 145-130 Ma. However, this does not represent the actual width of the basin but the amount of total horizontal divergence (elongation). To estimate the final length of the rifted margins and ocean, we use a simple "pure-shear" (symmetric and uniform) stretching approach (McKenzie, 1978) for which we need to assume the initial length of area

affected by rifting. Here we presume an initial length of c. 300 km, obtained from restoring the length of the base of syn-rift horizons across the present-day Southern Alps and Adriatic Foreland affected by rifting in the Jurassic (Masetti et al., 2012) and facing "stable" (i.e. not affected by rifting) Corsica in our reconstructions back to 200 Ma (Figure 6). This amount should

be considered as an absolute minimum, as this section represents only the preserved proximal part of the Adriatic margin. The present-day W-E orientation of the rift basins becomes NNW-SSE relative to Europe once back-rotated with Adria at 200 Ma (Figure 6), which fits well with the kinematics of opening of the PL Basin obtained from our reconstructions (Figure 4; section 4.1). Assuming an initial crustal thickness of 30 km (Masini et al., 2013) and a localization of the rift after the slow rifting phase described above (200-165 Ma) along a zone of 80 km (typical for Narrow Rift such as the East African Rift today,

Ebinger et al., 1999; Corti, 2009; Brune et al., 2017a), we estimate a total length of 380 km for the proximal margins (> 10 km thick), 120 km for the hyper-extended/OCT zone (< 10 km thick) and 240 km of oceanic crust (Figure 6). This remains a simple approximation as it does not take into account asymmetry during rifting, necking and hyper-extension, but nevertheless provides a first-order estimate.

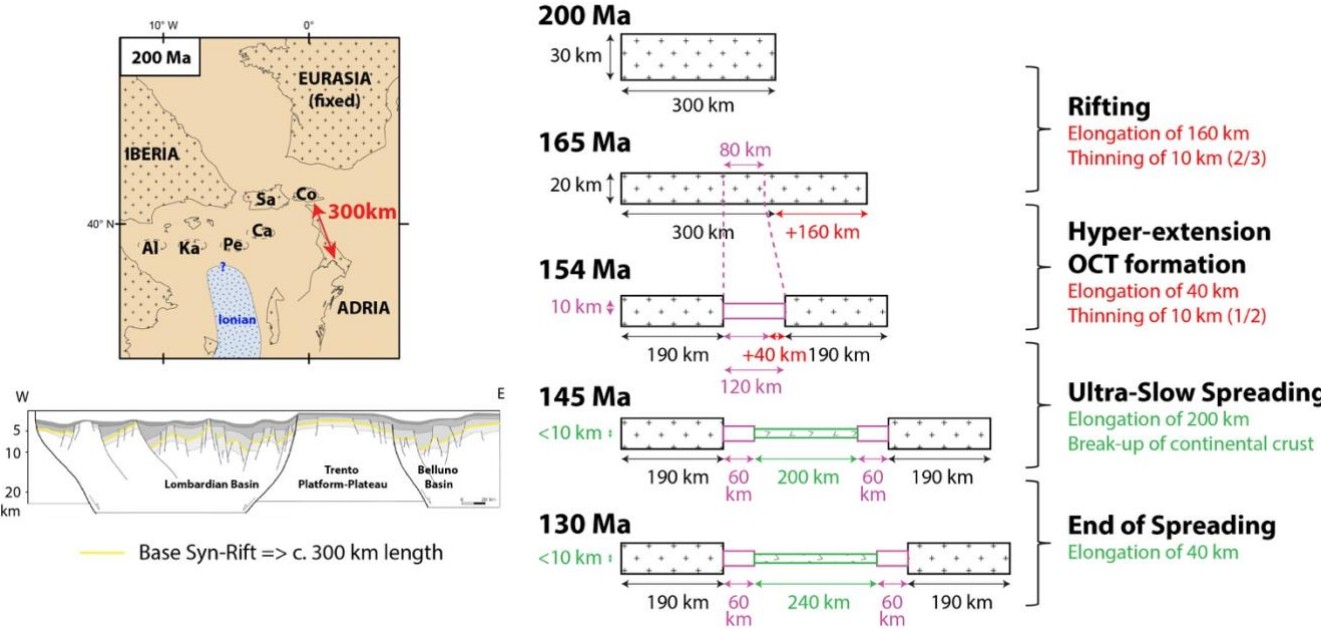

**Figure 6. Thinning of continental crust and extent of rifted margins along a 2D transect from Europe/Corsica and Adria using amount of divergence (elongation) from kinematic reconstructions (red transect on Figure 4) and a pure-shear approach (McKenzie, 1978). The initial length of 300 km is assumed by restoring length of base of syn-rift horizons along Southern Alps and Adriatic Foreland (down left, redrawn from Masetti et al., 2012). Note the present-day W-E orientation becomes NNW-SSE relative to Europe once back-rotated with Adria at 200 Ma as shown on our tectonic map (up left, red arrow).**


## 4.4. Geodynamic consistency

In order to test our kinematic model in the context of rifting and thinning of the PL Basin in more detail, we use a 2D thermo-mechanical modelling approach that is based on the geodynamic finite-element software SLIM3D (Popov and Sobolev, 2008). This model is driven solely by boundary velocities while localization of rift faults, lithospheric necking, continental break-up and subsequent sea-floor spreading evolves in a purely self-consistent manner. The model involves four layers (upper crust, lower crust, strong and weak mantle), which all deform in accordance with corresponding flow laws (wet quartzite, wet anorthite, dry olivine and wet olivine, respectively). The initial Moho depth is set at 33 km and the Lithosphere-Asthenosphere Boundary (LAB, 1300°C) at 120 km. The LAB is slightly elevated (5 km) in the center of the model to avoid rift localization at the model boundaries (Figure 7A). We follow a model setup that has been extensively used in modelling narrow rifting and rifted margin formation (Brune et al., 2012, 2013, 2017a, 2017b; Brune, 2014) and for a more detailed description of the employed boundary conditions, initial conditions, weakening mechanisms and material parameters, we refer the reader to Brune et al. (2014).

It is well-established that the rift velocity exerts key control on the structural evolution of a rift (Huismans and Beaumont, 2003; Pérez-Gussinyé et al., 2006; Tetreault and Buiter, 2018). In order to test the impact of our kinematic plate tectonic model, we derive rift-perpendicular velocities from the 2D transect between Corsica-Adria (red transect in Figure 4) as input for the geodynamic modelling by means of the following steps: (1) 4 mm/yr for the first 46 Myr (200 km divergence between 200-154 Ma); (2) 22 mm/yr for 9 Myr (200 km divergence between 154-145), and (3) 3 mm/yr for 15 Myr (40 km divergence between 145-130 Ma). We point out that the first acceleration shown in Figure 5 (15 mm/yr between c. 165-154 Ma) is obtained from the plate motion vectors (blue vectors on Figure 4) and reflects the strong obliquity of motion during that phase, however, along a 2D transect the net divergence is only 40 km between 164.7-154 Ma (3.6 mm/yr). Thus, this phase is accounted for in the first step of our model (with rates of 4 mm/yr).

The modelling results are available as an animation in supplementary material (movie S1). The results after 1 Myr (model setup), 20 Myr, 35 Myr, 46 Myr, 55 Myr and 70 Myr are shown on Figure 7 and allow us to distinguish different rifting stages (Figure 7A-D, constant velocity of 4 mm/yr). First, the entire domain is affected by normal faults (Figure 7A) until lithospheric necking commences at about 12 Myr into the model runtime. The rift localizes along a zone of about 110 km width and the onset of hyper-extension (< 10 km crustal thickness; Lavier and Manatschal, 2006) starts after 20 Myr (Figure 7B). The onset of mantle exhumation occurs after 35 Myr (Figure 7C) and is accompanied by the detachment of continental allochthons. Finally, the continental lithosphere breaks up rapidly when the rift velocity increases to 22 mm/yr at 46 Myr (Figure 7D). This is the moment when the lithosphere shallows drastically (< 10 km; Figure 7D; movie S1), as observed beneath present-day ultraslow-spreading ridges (e.g. Mohns Ridge; Johansen et al., 2019), which would lead to decompression melting and onset of sea-floor spreading.

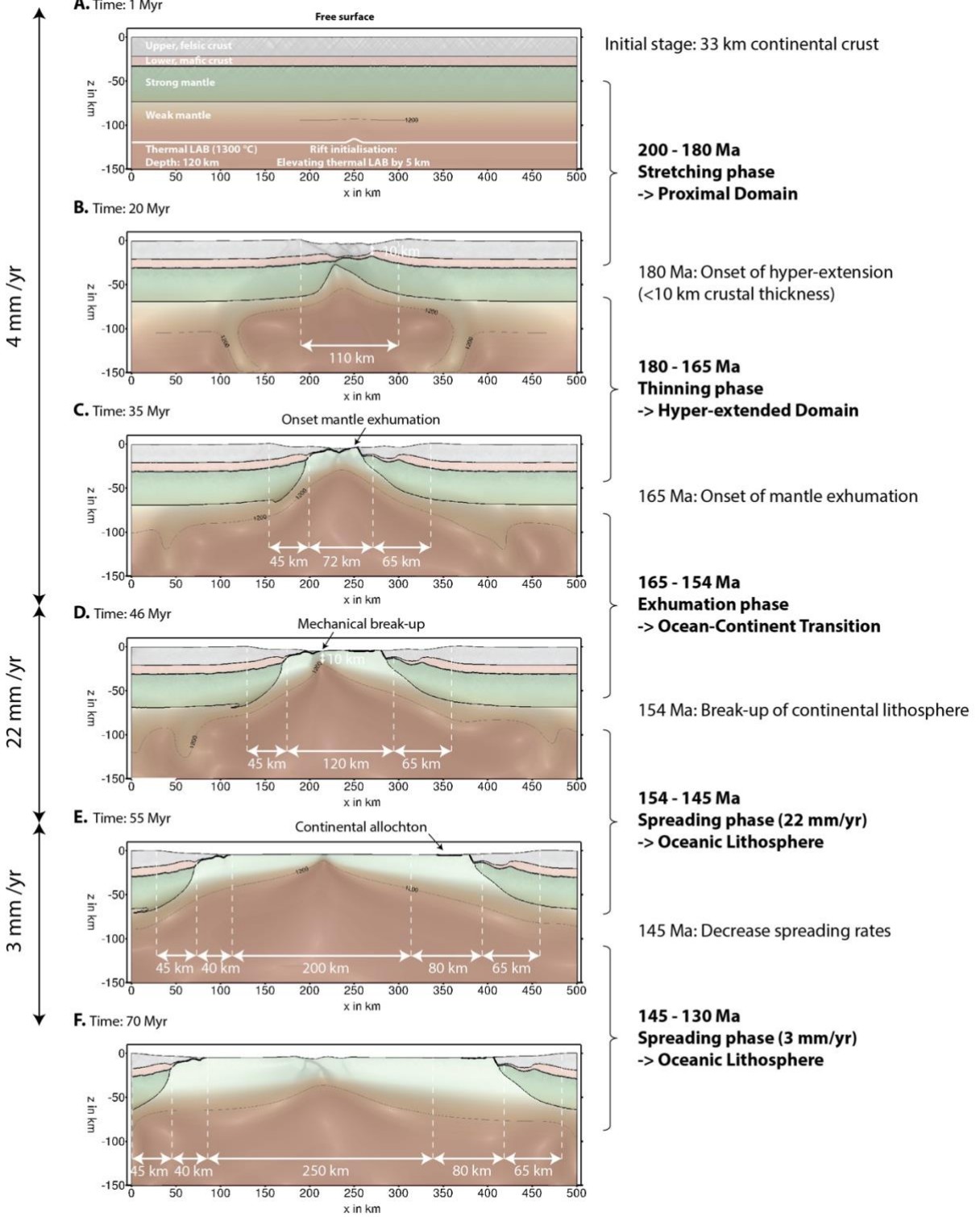

 **Figure 7 (previous page). 2D high-resolution thermo-mechanical modelling of the opening of the PL Basin using velocities from our kinematic reconstructions (red transect between Europe and Adria on Figure 4). Material phases are indicated in Panel A. Strain rate is visualized as black transparent overlay. Deep lithospheric and asthenospheric temperatures are depicted by brownish overlay and 1200°C isotherm (black line). Extent of necking zone (30-10 km thick), hyper-extended and OCT (< 10 km thick) and oceanic domain is indicated in white. See text for further discussion. Animation of model results can be found as supplementary movie (S1).**

## 5 Discussion

### 5.1 Opening and extent of the PL Basin

Using our kinematic reconstructions and geodynamic modelling of the rifting phase, we can identify and date the following four stages of opening of the PL Basin (Figures 5, 6, 7), as defined by Lavier and Manatschal (2006):

    (1) "Stretching Phase": rifting of the proximal continental margin and necking of the lithosphere between 200-180 Ma
600         (first rifting phase);

    (2) "Thinning Phase": hyper-extension of the distal continental margin between 180-165 Ma (second rifting phase);

    (3) "Exhumation Phase": OCT formation with mantle exhumation between 165-154 Ma;

    (4) "Spreading Phase": break-up and slow to ultra-slow oceanic spreading between mostly 154-145 Ma.

Our results fit very well with independent datasets (tectono-sedimentary, geochemical, geochronological) from the Alps
(section 2.1.1; Figure 5) that show two-rifting phases (c. 200-180 Ma, 180-165 Ma; e.g. Froitzheim and Eberli, 1990; Mohn et al., 2010; Ribes et al. 2019), activation of the necking zone around 180 Ma (Mohn et al., 2012; Ribes et al. 2019), onset of mantle exhumation and magmatism along typical magma-poor OCT at about 165 Ma (e.g. Manatschal and Müntener, 2009; Ribes et al. 2019).

Our kinematic reconstructions show an increase in obliquity and rate of plate motion between 165-154 Ma (Figures 4 and 5) and a progressive opening of the PL Basin from SW to NE (Figure 5). Thus the transition between stages 3 and 4, i.e. break-up and onset of spreading, was most likely a propagating process from SW to NE. Interestingly, abrupt plate acceleration such as the one at c. 165 Ma ("Speed-up" on Figure 5) is commonly observed along other continental margins worldwide (Brune et al., 2016). On other margins, this acceleration corresponds to the shaping of the distal part of the margins, pre-dating
continental breakup of about 5-10 Ma, and reflects a strength-velocity feedback mechanism during rifting ("dynamic rift weakening"; Brune et al., 2016). Moreover, Brune et al. (2018) showed a clear correlation between rift obliquity and increase in plate motion velocities, as we observed between 165-154 Ma (Figure 5). However, it is the effective rift-perpendicular velocity that leads to lithospheric thinning and thus oblique segments need more time to reach breakup than orthogonal rifts. This corroborates our interpretation that stage 3 (165-154 Ma) corresponds to the formation of the (hyper-)distal OCT zone,
with the juxtaposition of pelagic sediments, serpentinized mantle, gabbros, basalts (e.g. Epin et al., 2019), prior break-up. The detachment of continental allochthons, such as Sesia (Figure 4; e.g. Babist et al., 2006), occurs also during that stage, which fits with the crystallization age (164-156 Ma) of zircons within syn-magmatic shear zones south of the Sesia zone (U/Pb zircon age; Kaczmarek et al., 2008). We therefore diverge from Li et al. (2013) who interpret this phase as oceanic spreading. We

rather suggest that most of the preserved mafic rocks came from the OCT domain rather than the mature oceanic domain, mostly lost to subduction, which is supported by the sub-continental (refertilized) mantle-type composition of most peridotites of the ophiolites preserved in the Alps (Picazo et al. 2016). A subcontinental affinity of the exhumed mantle has been inferred from petrological analysis in the Western Alps suggesting diffuse porous flow of asthenospheric melts through the continental lithospheric mantle as a key process (Piccardo and Guarnieri, 2010). Similar to other recent modelling efforts (Hart et al., 2017; Andrés-Martínez et al., 2019; Jammes and Lavier, 2019), our numerical models do not capture porous flow processes and are therefore not comparable with this type of observation. However, they show that the asthenospheric mantle resided close to continental mantle lithosphere prior to exhumation, which might be indicative of continental mantle affinity. Furthermore, our interpretation does not exclude the possibility that first immature or embryonic oceanic domains, such as the Chenaillet-Montgenèvre Ophiolite (oceanic mantle-type, Picazo et al. 2016; U/Pb ages ranging from 148-156 Ma to 165 Ma; Costa and Caby, 2001; Li et al., 2013, respectively) have formed already during this phase of very oblique plate motion. Still we propose that formation of mature oceanic crust took place when the plate motion became more orthogonal and rates of opening reached c. 22 mm/yr (Figures 4 and 5). Indeed, our modelling results show that mechanical break-up of the continental lithosphere coincides with the increase of rift-perpendicular velocity (Figure 7D; 1200°C at  10 km depth). We have to caution that the numerical model cannot clearly differentiate between mantle exhumation prior to break-up and potential mantle exhumation during spreading. In either case, mature oceanic spreading was slow to ultra-slow and very short-lived (9 Ma), as the divergence along the PL domain decreased significantly at 145 Ma and completely ceased at 130 Ma.

Our geodynamic modelling, constrained by our plate reconstructions, reproduces the kinematics of the different rifting phases recorded in the Alps, but also the first-order structures of the hyper-extended margins. The final "shape" of the distal rifted margin obtained from our modelling (Figure 7E) agrees with geological reconstructions of the Alpine Tethys margins (Mohn et al., 2012, their Figure 2). Moreover, we provide quantitative constraints on the final width of the different domains: (1) 45-65 km for the necking zone (thinning of the crust from 30 to 10 km; Figure 7); (2) 40-80 km for the distal margin (crust thinner than 10 km, hyper-extended and OCT domains; Figure 7); (3) maximum 250 km for the oceanic domain (Figures 4, 6 and 7). Hence, the total width of the PL Basin, including both the rifted continental margins (thinner than 30 km) and the oceanic crust featured a total extent of 480 km (Figure 7E).

The kinematic evolution of the comparably small PL Basin is closely linked to the much larger neighboring Atlantic Ocean and the variation of rifting/spreading velocities coincides with changes in divergence rates of the Central Atlantic (Figure 5). The PL Ocean opened as a third arm of the Central Atlantic Ocean in Jurassic time (as previously proposed; e.g. Stampfli and Borel, 2002) and "died out" when the North Atlantic took over at c. 145 Ma (Barnett-Moore et al., 2018). It then closed due to the opening of the South Atlantic (e.g. Müller et al., 1999) and ensuing counter-clockwise motion of Africa relative to Europe in Cretaceous-Cenozoic times.

## 5.2 Assessing the model

The kinematics of the opening of the PL Basin is primary governed by the motion of Europe and Adria, as Iberia-Sardinia-Corsica started to move at 145 Ma, i.e. after the opening of the PL Basin. In our model, it is more precisely governed by the motion of Europe and Africa since we assume Adria rigidly connected with Africa after the opening of the Ionian Basin in Triassic time and prior the strike-slip motion along the MATF in Cretaceous-Cenozoic time (see section 3.3). Thus our results are extremely dependent on the kinematic reconstructions used for the opening of the Central Atlantic in the Jurassic, here the model of Kneller et al. (2012). The latter is based on a non-rigid inversion method that restores incrementally crustal stretching allowing a better restoration of the syn-rift to breakup phase and producing realistic subsidence and crustal thinning patterns that are consistent with field observations. Other recently published model for the Central Atlantic, such as Labails et al (2010), show, however, significant gaps in palinspastic restorations of the margins (see further discussion in Kneller et al., 2012). Moreover, when using the rotation poles of Labails et al (2010) for the Central Atlantic, we obtain a phase of convergence rather than divergence between 190 and 170 Ma in the PL domain between Europe-Adria. This is in contradiction with the clear record of syn-rift basins of Early Jurassic age in the Alps (section 2.1.1) while our kinematic model using Kneller et al. (2012) fits the geological record of rifting and spreading along the Alpine Tethys (section 5.1).

We assume that the Ionian Basin opened in the Triassic (Speranza et al., 2012). However, Tugend et al. (2019) recently re-evaluated the age of opening of the Ionian Basin to Late Triassic-Early Jurassic (rifting) and late Early Jurassic-Middle Jurassic (spreading), i.e. overlapping in time with the two first rifting stages of the PL Basin. However, uncertainties remain on the exact amount and direction of extension between Africa and Adria that took place at that time. Indeed, the Ionian Basin probably resulted from several phases of extension as indicated for example by the presence of Permian oceanic basins in Sicily (Catalano et al., 1991) and the exact direction and amount of spreading cannot be clearly reconstructed as most of the oceanic domain is now subducted. We tested nevertheless an alternative kinematic scenario in which the Ionian Basin (using the present-day total width of c. 350 km between the Malta and Apulian escarpments; Tugend et al., 2019, their Figure 13a) opens – and thus Adria (Apulia) moves relative to Africa (Tunisia) – in Early-Middle Jurassic (200-164.7 Ma; Tugend et al., 2019) and in a NW-SE opening direction following Frizon de Lamotte et al. (2011). This would reduce the overlap problem between northern Adria and Sardinia-Corsica mentioned in section 3.2 (Figure 3). However, it would significantly increase the obliquity of motion between Sardinia-Corsica and Adria (and the rates of motion up to 9 mm/yr) and therefore reduce considerably the width of the rifted PL domain. Moreover, if we include a significant sinistral strike-slip motion between Africa and Adria during the opening of the Ionian Basin (following the interpretation that the Malta and Apulian escarpments are transform margins; Frizon de Lamotte et al. 2011), Adria would converge towards Sardinia-Corsica rather than diverge, which would be in conflict with the timing of syn-rift deposits and normal faulting along northern Adria (section 2). Future work is therefore required to test in more details such alternative scenarios. Similarly, as discussed in section 2, a possible Jurassic rifting phase affected the Pyrenees and Valais Basin and would also reduce the width of the rifted PL domain between

Europe-Adria. Therefore, our calculated extent of the PL Basin and subsequent amount of convergence (below) constitutes maximum estimates.

## 5.3 Implications for subduction processes in the Alps

Using the motion path of NW Adria (Ivrea) relative to Europe, we can estimate the magnitude, direction and rate of Adria-
Europe convergence since the onset of "Alpine" south-directed subduction at about 84 Ma (Figure 8; Table 2). Our model indicates a total amount of plate convergence of c. 680 km towards the NW (relative to Europe) since 84 Ma. We consider two time periods of two different modes of orogenic processes in the Alps (section 2.2): (1) subduction of both oceanic and extended continental/transitional crust to high depth (marked by (ultra)high-pressure metamorphism) between c. 84-35 Ma; and (2) continental collision dominated by positive buoyancy of the continental lithosphere after slab break-off at about 35 Ma
(Von Blanckenburg and Davies, 1995). Following this definition, the total amount of plate convergence can be subdivided into c. 420 km during subduction (84-35 Ma; average rate of 8.6 mm/yr) and c. 260 km during collision (35-0 Ma; average rate of 7.4 mm/yr). If we take into account that the PL ocean had a maximum width of 250 km (in the same NW-SE direction), we can furthermore estimate that at least 63% ((680-250)/680 x 100) of the material involved in the Alpine Orogeny was extended continental lithosphere and OCT zones of the PL and Valais basins (Figure 4). This may explain why the Alps are such a
singular mountain belt, e.g. with the absence of well-developed magmatic arc during the alpine subduction (McCarthy et al., 2018, 2020) and widespread high-pressure metamorphosed continent-derived rocks (e.g. Kurz et al., 1999; Berger and Bousquet, 2008; Bousquet et al., 2008; Nagel et al., 2013; Sandmann et al., 2014; Fassmer et al., 2016).

Our estimates of plate convergence are lower than the one obtained from geological reconstructions of the Alps (Handy et al.,
2010, 2015), especially for the subduction phase during 84-35 Ma (c. 200 km difference; Table 2). The 615 km of motion obtained by Handy et al. (2010, 2015) between 84 and 35 Ma is based on a projection (465 km) of the estimated c. 400 km shortening in the eastern Central Alps from (Schmid et al., 1996) between 67-35 Ma, added to the estimated 150 km subduction of Sesia between 84-67 Ma based on the depth of high-pressure nappes of this unit (Babist et al., 2006). A major part of the 400 km shortening in the Central Alps estimated by Schmid et al. (1996) is itself based on an inferred width of the Valais
Basin and Briançonnais that went into subduction between 65-50 Ma and an inferred amount of subduction also based on depth of high-pressure metamorphism of the Adula Nappe (150 km) between 50-40 Ma. Two possible sources of error arise and may explain the discrepancy with our results: (1) the past extent of the Valais/Briançonnais domain inferred by Schmid et al. (1996) was not kinematically constrained and (2) the depth of high-pressure rocks – and thus the amount of subduction – may be overestimated due to possible tectonic overpressure (Ford et al., 2006; Schmalholz and Podladchikov, 2013; Pleuger and
Podladchikov, 2014; Yamato and Brun, 2017).

We note that our estimates of plate convergence are, however, slightly higher than the recent model proposed by van Hinsbergen et al. (2020; Table 2). This is due to the different assumptions we made for the motion of Iberia, Sardinia-Corsica and Adria as previously discussed (section 3). The major difference impacting the amount and rate of plate convergence is that

we consider a sinistral strike-slip motion within Adria of c. 230 km along the MF (Figure 3) between 100-40 Ma (Schettino and Turco, 2011; Hosseinpour et al., 2016). This remains poorly constrained and needs to be further studied in the future. Thus, our amounts and rates of plate convergence should be taken as maximum values. We want to emphasize however, that our assumptions regarding the motion of Iberia (post-145 Ma) and strike-slip motion within Adria (100-40 Ma) do not affect our results and interpretation for the kinematics of opening (200-145 Ma) and former extent of the PL Ocean and its margins.


To solve the remaining problematic areas especially regarding the paleo-magnetic data from Iberia and Sardinia, future work should focus on quantifying, and incorporating into kinematic reconstructions, the tectonic deformation within the Iberian and Adriatic plates, and between Sardinia-Corsica and France. Interestingly, Dannowski et al. (2020) re-evaluated the nature of basement beneath the Liguro-Provencal Basin to be hyper-extended continental crust (only a few km thick) and exhumed

mantle, rather than oceanic crust. Moreover, compressional earthquakes and inverted structures have been observed within the Liguro-Provencal Basin and along the northern African margin, which suggest that those hyper-extended margins and OCT zones may represent incipient subduction zones accommodating the present-day Africa-Europe convergence (Billi et al., 2011; Hamai et al., 2018; Thorwart et al., 2021). This reflects once more the importance of those domains as inherited weakness for potential subduction initiation processes (Beltrando et al., 2010; Tugend et al., 2014; Kiss et al., 2020; Zhou et al., 2020).

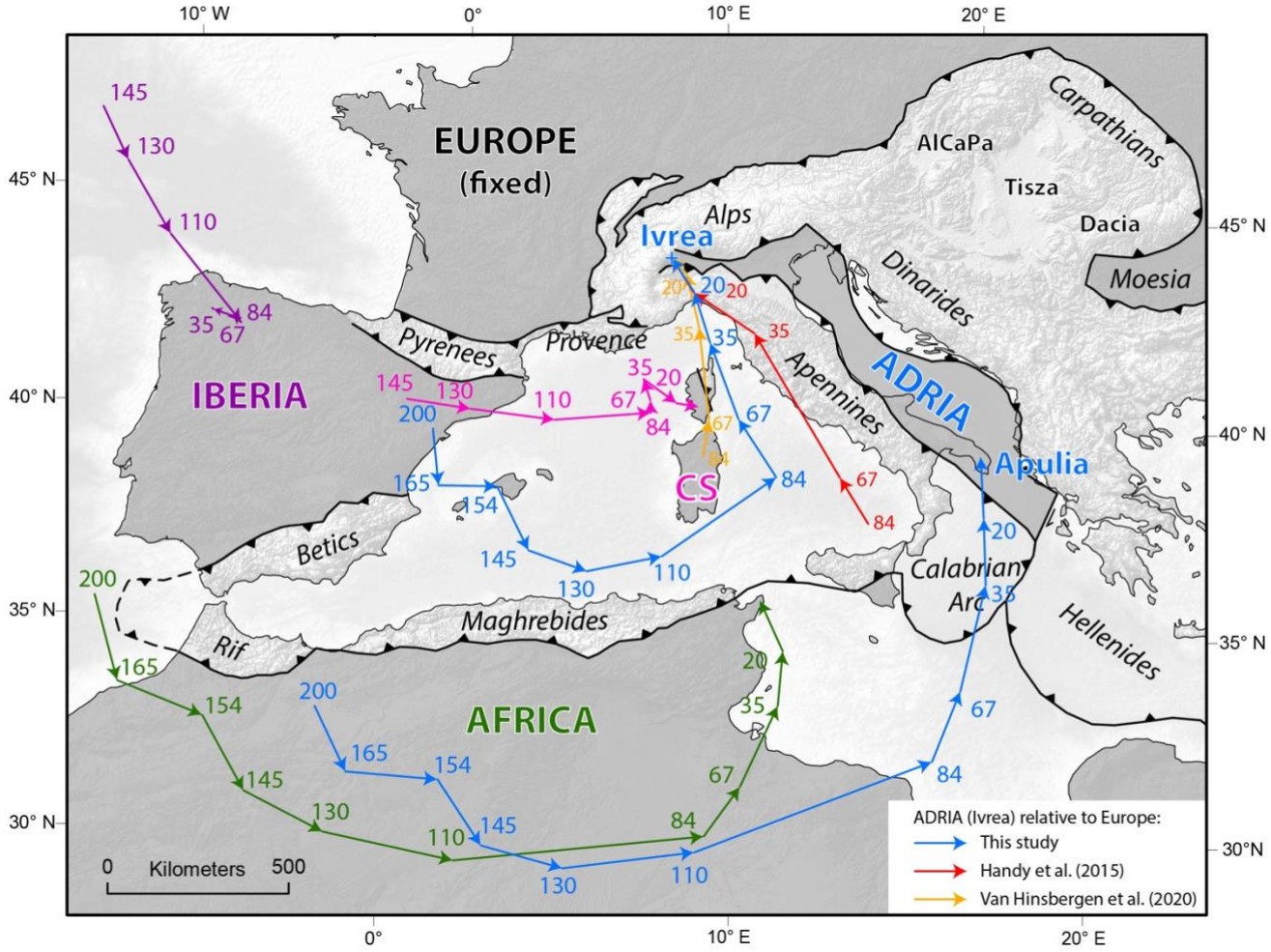


**Figure 8: Motion paths of northern (Ivrea) and southern (Apulia) Adria (blue), Iberia (dark purple), Corsica (light purple) and Africa (green) relative to Europe stationary back to 200 Ma (time step indicated in Ma) using our kinematic reconstructions. Alternative motion paths of northern Adria are also shown for comparison (Handy et al., 2015 in red and van Hinsbergen et al., 2020 in orange). Amount, direction and rate of convergence of Adria (Ivrea) – Europe are given in Table 2.**


**Table 2: Amount, mean direction (azimuth) and mean rate of Adria (Ivrea) - Europe convergence during Alpine collision (35-0 Ma) and subduction (84-35 Ma). Comparison with model of Handy et al. (2010, 2015) and van Hinsbergen et al. (2020).**

| Time Period | Adria (Ivrea) – Europe Convergence | | | Reference |
| --- | --- | --- | --- | --- |
| | Amount | Azimuth | Rate | |
| **35-0 Ma** **Alpine Collision** | 313 km | 315° | 8.9 mm/yr | Handy et al. (2015) |
| | **260 km** | **335°** | **7.4 mm/yr** | **This study** |
| | 210 km | 338° | 6 mm/yr | van Hinsbergen et al. (2020) |
| **84 - 35 Ma** **Alpine Subduction** | 615 km | 330° | 12.6 mm/yr | Handy et al. (2015) |
| | **420 km** | **335°** | **8.6 mm/yr** | **This study** |
| | 350 km | 358° | 7.1 mm/yr | van Hinsbergen et al. (2020) |

**6 Conclusions**

In this study, we focused on the divergent Mesozoic history of the Alpine region, which is crucial to better understand the complex tectonic evolution of this mountain chain. By updating the regional reconstructions of the Western-Central Mediterranean area, implemented into a global plate circuit and complemented by thermo-mechanical modelling, we shed new light on the past extent and opening kinematics of the Piemont-Liguria Ocean. The opening of the Central Atlantic in Jurassic times led to divergence between Europe and Africa and the opening of the Piemont-Liguria Ocean in four stages. Two successive phases of stretching and thinning of the continental crust (> 10 km thick, 200-180 Ma; < 10 km thick, 180-165 Ma) were followed by mantle exhumation and MORB-type magmatism along the hyper-distal (OCT) part of the margin (165-154 Ma). Mature oceanic spreading in a NW-SE direction (relative to Europe) was short-lived (154-145 Ma) and was slow to ultra-slow (up to 22 mm/yr), ceasing progressively in the Early Cretaceous (145-130 Ma). This led to the formation of a small mature oceanic domain of maximum 250 km width. This kinematic evolution is in strong agreement with independent geological records of rifting in the Alps and the age of the Alpine Tethys derived units. The opening of the North Atlantic at 145 Ma led to a period of mostly strike-slip tectonics along this domain. South-directed "Alpine" subduction of the PL Ocean initiated at about 84 Ma, at rates of c. 8.6 mm/yr, and progressed from SE to NW until collision of the Adriatic Plate with the European proximal margin at about 35 Ma. We highlight that at least 63 % of the lithosphere involved in the Alpine Orogeny was highly thinned continental (and transitional) lithosphere and that most of the Alpine Ophiolites were derived from the OCT zones, which emphasizes the importance of the distal domain of continental margins in subduction and exhumation processes. We provide new kinematic constraints (maximum extent of oceanic crust and plate convergence direction and rates) that can be used as set-up for future geodynamic modelling and a better understanding of subduction initiation, magmatism and exhumation processes during the Alpine Orogeny.

**Data availability**

All reconstruction files of the kinematic model presented in this paper, as well as animations and age-grids, are available at https://www.earthbyte.org/webdav/ftp/Data_Collections/Muller_etal_2019_Tectonics/ or by contacting the first author.

**Video supplement**

Results of the 2D thermo-mechanical modelling of the opening of the Piemont-Liguria Basin, using velocities derived from our kinematic reconstructions, are shown in an animation (Movie S1) in supplement to this manuscript.

## Author contribution

ELB developed the research concept, acquired funding, carried out the kinematic reconstructions and their interpretation, designed the figures and wrote the manuscript with input from all authors. SB performed the thermo-mechanical modelling and contributed to the interpretation and discussion of the rifting phase. UK participated in the conceptualization of the initial project and in discussion on regional tectonics reconstructions. SZ, MS and DM were involved in the implementation of the regional kinematic reconstructions into the global deforming plate model using GPlates. SZ and MS generated animations centered on the Mediterranean and age-grids of the PL Ocean. DM welcomed ELB for a research stay at the EarthByte Group, University of Sydney in 2017, which initiated this collaborative work.

## Competing interests

The authors declare that they have no conflict of interest.

## Acknowledgements

ELB and US acknowledge financial support of the German Research Foundation (DFG; BR 4900/2-1 and US 100/4-1). SB was funded by the Helmholtz Association through the Helmholtz Young Investigators Group CRYSTALS (VH-NG-1132). SZ was supported by Australian Research Council grant IH130200012 and a University of Sydney Robinson Fellowship. MS was supported by Australian Research Council grants FT130101564 and DP200100966. GPlates development is funded by the AuScope National Collaborative Research Infrastructure System (NCRIS) program. This work benefited greatly from stimulating conversations with Giancarlo Molli, Jan Pleuger, Mark Handy, members of the AlpArray/4DMB community and participants of the 14th Emile Argand Conference on Alpine Geological Studies (Alpine Workshop) in Sion in 2019. Finally, we thank Andrea Argnani and Douwe Van Hinsbergen for their constructive reviews.

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
