# Peer review of "Kinematics and extent of the Piemont-Liguria Basin – implications for subduction processes in the Alps"

_Solid Earth, 2020_

## Short Comment (SC1) · 15 Oct 2020

Any attempt to reconstruct the motion of plates or microplates in the Mediterranean domain is welcome. As a co-author of Hinsbergen et al. (2020) I am perfectly aware that the reconstruction we presented is just one possibility amongst others and hence alternive scenarios are more than welcome. Some of the potential problems were discussed in that paper.

However the intruduction of a Mid-Adriatic Transorm Fault in order solve compatibility problems regarding Sardinia-Corsica 200 Ma ago as a consequence of the input into their reconstruction is totally unsound and not supported by any geolgical or geo-

physical evidence whatsoever. The authors take this Mid-Adriatic Transform Fault with 230km!!!!! strike slip displacment as a precursor of what they call "Mid Atlantic Ridge". The name ridge is totally inappropriate and the amount of displacement at 200 Ma is, in my view, an invention in order to save a problem rising from their reconstruction.

The authors cite d'Agostino et al. (2008) in this context who stated that "We suggest that the present-day microplate configuration follows a recent fragmentation of the Adriatic promontory that during the Neogene rigidly transferred the Africa motion to the orogenic belts that now surround the Adriatic region." There is indeed seismic evidence for some complex movements with absolutely subordinate strike-slip displacments; clearly the primary role of this only very recently active fault array is to acommodate differential rotations of northern and southern Adria that take place at the present day according to solid and undisputed GPS analyses. There is no evidence available in terms of activity of this fault in the more distant past. In the contrary, the reconstruction of Triassic seaways and platforms (e.g.Bernoulli 2001, and many others) shows a perfect fit of the paleogeographic features for Triassic times across the Adriatic Sea from Albania to the Marche in Italy. The supposed lateral movment by 230km at 200 Ma (end of Triassic) would have severely disrupted this simple paogeographic scheme and would not have been left undetected in the seismic sections across the Adriatic Sea by the Italian (CROP Atlas) and Croatian colleagues, that show undisturbed Mesozoic cover across the Adriatic Sea, except for a complex fault array that is responsible for the present-day neotectonic activity nonitored by GPD-analysis.

I suggest to the authors to be more honest and acknowledge that their reconstruction, like any other one, like for example that of Hinsbergen at al (2020), has severe shortcomings resulting in incompatibities. These shortcomings need to be discussed. Inventing manifestably inexistent "fake"-strike slip movements of this order of magnitude is dishonest in my view.

Fig. 2: scheme of the Periadiatic area for the Triassic of Bernoulli 2001,

Please also note the supplement to this comment:
https://se.copernicus.org/preprints/se-2020-161/se-2020-161-SC1-supplement.pdf

[Figure]

**Figure 18.1** Tectonic units and late Jurassic palaeogeography in the peri-Adriatic area. No palinspastic correction was made (after Bernoulli, 1972; with additions from Cati *et al.*, 1987 and Argnani *et al.*, 1996). (GS, Gran Sasso d'Italia; I, Imerese basin; L, Lagonegro basin; M, Montagna della Maiella, P, Panormic platform; S, Sicani basin; T, Toscanides; the Umbria seamounts are too small to be indicated at this scale).

**Fig. 1.** Figure from Bernoulli 2001

---

## Author Comment (AC1) · 22 Oct 2020

Any comment, as strongly worded as they might be, to improve our manuscript and discussion on reconstructions of the Alpine-Mediterranean belt is welcome. We thank Stefan Schmid for raising an important discussion on potential deformation within the Adriatic plate and would like to reply below to his critical points.

First, as we wrote in section 3.3 (lines 312-314), the idea of splitting Adria into two plates to solve this "compatibility problem" with Iberia-Sardinia-Corsica at 200 Ma, is not an invention from our part but was proposed by Stampfli and Borel (2002) and applied in the kinematic reconstructions of Schettino and Turco (2011). We follow this

model as it solves paleogeographic problems of Sardinia-Calabria relative to Adria in Permian-Triassic times (section 3.3) and the closing time of the Piemont-Liguria (PL) Ocean between Iberia and Europe-Adria (section 2.3, lines 193-209). We used the rotation poles of Schettino and Turco (2011) for the motion of northern Adria for time prior to 20 Ma, as listed in our Table 1.

Second, we refer to d'Agostino et al. (2008) for the present-day subdivision of the Adriatic plate (indeed clearly identified by GPS data) but not for the deformation along the Mid-Adriatic Ridge. For the latter, we quote Scisciani and Calamita (2009), who introduced the term "Mid-Adriatic Ridge" (not us, and not "mid-Atlantic") and who presented interpretation of seismic reflection and boreholes data from the central Adriatic Sea (including the CROP data). They clearly show that the so-called Mid-Adriatic Ridge formed by transpressional reactivation of pre-existing rift-related Mesozoic structures. The inversion occurs clearly in Plio-Pleistocene time and possibly even earlier. A first inversion phase is indeed observed by lateral thickness variations and unconformities above the Albian-Aptian reflector (please see Figure 6 of Scisciani and Calamita, 2009), especially during the Paleogene-Miocene succession (Figures 3 and 8 of Scisciani and Calamita, 2009). We will add this to our text in section 3.3.

That said, we fully agree that the model we present is a major simplification of the reality. As we wrote in lines 316-325, we support the idea that pre-existing structures related to rifting in the Triassic-Jurassic were reactivated in transpression during the Late Cretaceous-Cenozoic convergence of Africa with Europe not only along the Mid-Adriatic Ridge but throughout the entire Adriatic plate. This deformation was most likely diffuse throughout the entire plate and distributed along several tectonic structures rather than localized along one major strike-slip fault. We had to introduce this simplification for kinematic purposes. But it is important to emphasize that this strike-slip motion along the Mid-Adriatic Transform Fault during the Alpine Orogeny (implemented in our model between 100 and 40 Ma, following rotation poles of Schettino and Turco 2011) does not affect the kinematics of rifting and opening of the PL Basin, which

is the main aim of our paper. However, it does affect the paleogeographic position of Adria with respect to Sardinia-Calabria and Iberia and it affects the closure of the PL Ocean, as mentioned above (also see our manuscript in sections 2.3 and 3.3). We discuss further the implications of our model for subduction processes in the Alps in section 5.3, where we provide estimates of plate convergence between northern Adria and Europe not only for our model but also for two other reconstructions, including the recent model of van Hinsbergen et al. (2020), and thus a range of possibilities to the reader.

In order to make that issue clearer in the text, we propose to present our model as a potential "end-member" (simplified) kinematic model that would imply a maximum amount of strike-slip motion within Adria, as well as between France and Iberia-Sardinia-Corsica, and thus a maximum width of the PL Ocean and of plate convergence during the Alpine Orogeny between northern Adria and Europe. In that view, the model of van Hinsbergen et al. (2020) can be considered as the opposite "end-member" model that would imply on the contrary no deformation within Adria, no strike-slip motion between France and Sardinia-Corsica (but convergence between Iberia and Sardinia, and in the Pyrenees in Early Cretaceous time) and a minimal width of PL Ocean and plate convergence between northern Adria and Europe. While both models are geometrically viable based on rigorous plate reconstruction principles (most importantly rigid body rotations around Euler poles), the geological reality lies surely somewhere in between and would require further work on documenting and implementing intraplate deformation within Adria and also within Iberia (as recently proposed by Angrand et al., 2020), into future kinematic models.

References

Angrand, P., Mouthereau, F., Masini, E., and Asti, R.: A reconstruction of Iberia accounting for Western Tethys–North Atlantic kinematics since the late-Permian–Triassic: Solid Earth, 11, 1313–1332, doi:10.5194/se-11-1313-2020, 2020.

D'Agostino, N., Avallone, A., Cheloni, D., D'Anastasio, E., Mantenuto, S., and Selvaggi, G.: Active tectonics of the Adriatic region from GPS and earthquake slip vectors: Journal of Geophysical Research: Solid Earth, 113, 1–19, doi:10.1029/2008JB005860, 2008.

Schettino, A., and Turco, E.: Tectonic history of the Western Tethys since the Late Triassic: Bulletin of the Geological Society of America, 123, 89–105, doi:10.1130/B30064.1, 2011.

Scisciani, V., and Calamita, F.: Active intraplate deformation within Adria: Examples from the Adriatic region: Tectonophysics, 476, 57–72, doi:10.1016/j.tecto.2008.10.030, 2009.

Stampfli, G.M., and Borel, G.D.: A plate tectonic model for the Paleozoic and Mesozoic constrained by dynamic plate boundaries and restored synthetic oceanic isochrons: Earth and Planetary Science Letters, 196, 17–33, doi:10.1016/S0012-821X(01)00588-X, 2002.

Van Hinsbergen, D.J.J., Torsvik, T.H., Schmid, S.M., MaÅ̌cenco, L.C., Maffione, M., Vissers, R.L.M., Gürer, D., and Spakman, W.: Orogenic architecture of the Mediterranean region and kinematic reconstruction of its tectonic evolution since the Triassic: Gondwana Research, 81, 79–229, doi:10.1016/j.gr.2019.07.009, 2020.

---

## Referee Comment (RC1) · Douwe J. J. van Hinsbergen (Referee) · 3 Nov 2020

Dear editor, dear authors,

Hereby I provide my review of the paper by Le Breton et al entitled 'Kinematics and extentofthe Piemont-LiguriaBasin–implications for subduction processes in the Alps'.

Le Breton and colleagues make a kinematic model for the western and northern, and a bit of the northeastern Mediterranean region and display it as GPlates files. The Mediterranean region is complex and therefore an interesting challenge to reconstruct, and many people have attempted this before, and I understand the desire of the authors

to also give it a shot. I have recently done this as well, using a reconstruction protocol to restore paleogeography from kinematic constraints from orogens, and I therefore read this paper with interest to see what this team of authors, with a strong track record in plate reconstructions, would make of it.

I find it difficult, however, to see why this paper would be better than previous attempts to restore the Mediterranean region for the following reasons. 1. The authors provide no reason in their introduction why they found it necessary to make a new reconstruction and how their reconstruction systematically differs from previous attempts, or why they expect fundamentally different conclusions for the evolution of the Alpine Tethys with their approach. 2. The paper cites almost no geological data, but instead hinges almost entirely on interpretations and models. They don't describe the geological record of the Alpine Tethys, but the interpreted paleogeography. In my detailed comments below, I point this out for specific cases. For instance, they describe 'en echelon transtensional basins' in the Pyrenees. That is not what you see in the Pyrenees, you see a faulted, folded, intensely deformed, partly metamorphosed bunch of rocks that were interpreted to have been deposited in such basins. They describe a distal passive margin, or a hyperextended ocean floor, but the underlying observations or even the location where those are made are not specified. The model of Iberian motion is given as fact before alternatives and discussions are provided. This makes it very difficult for a reader to judge the validity of the final model, which to me rather seems a model based on selected models. 3. But my main problem lies in the inconsistency in argumentation. I believe this is best illustrated by the following example. The Valais Basin in the Alps is interpreted from an intensely deformed and metamorphosed series of deep-marine Mesozoic sediments and associated mafic and ultramafic rocks. These mafic rocks have crystallization ages estimated with geochronology at Jurassic to Early Cretaceous consistent with the sedimentary record, which is what the authors interpret as the age of hyperextension. ArAr ages of these same rocks nowhere give Jurassic or Cretaceous ages, but Cenozoic, which they (and everybody else) interpret as related to exhumation following burial. Regional HT metamorphism is then taken

by the authors as evidence for slab break-off following collision. In the Pyrenees are also ultramafic and mafic rocks interpreted to reflect hyperextension. These are associated with Jurassic Sm/Nd ages and such Jurassic extension also follows from rifting recorded in the Bay of Biscay as well as in Pyrenean stratigraphy. Also here, ArAr ages are considerably younger, and follow upon HT metamorphism. In analogy to the Pyrenees, we have argued that the HT metamorphism is related to slab break-off following collision, pointed out where the slab is currently residing, and showed that that model is fully consistent with interpreted magnetic anomalies and paleomagnetic data. But the authors prefer for the Pyrenees to discard the evidence for Jurassic rifting, the magnetic anomalies, and the paleomagnetic data without explaining these data, and instead interpreting the HT metamorphism as hyperextension related. 4. As a result of the models they follow for the Pyrenees, they make a model for Iberia that devoid of any quantitative kinematic constraints (all constraints are discarded), from which it follows that also their reconstructed Piemonte-Ligurian ocean is does not have a quantitative basis.

The model in this paper is mostly a compilation of previous models, and in my view adds little to the discussion. I don't mind if it's published, I disagree with this model as much as with the previous ones for reasons I published before and repeat in my review. The authors do point out, however, that my model predicts shortening between Sardinia and Iberia during Iberian rotation, for which I must have become blind when I was working on my version: there is indeed no evidence for this, and it bothers me, so I'll research it and see how I can solve it. But I believe that solving that should not result in ignoring all paleomagnetic data and invoking a 230 km strike-slip fault through Adria for which there is no evidence (and which, if I may add, is not a simplification as you respond to Stefan, but rather a complication).

The detailed comments below may help the authors to clarify their paper and bring their new message across better. I don't expect the authors to agree with me, but I do ask them to explain the basis for discussions, rather than to just choose models based

on other models.

Douwe van Hinsbergen Utrecht, November 3, 2020

l. 16: Should a kinematic model be geodynamically 'consistent'? Suppose that data show a kinematic pattern, but you're not able to model it because we currently lack the conceptual understanding of the process, should we then adjust the kinematic model? Or learn something about the process? I believe that kinematic models should be boundary conditions for dynamic models, not the other way around, don't you?

l. 22: What is a bit confusing is that you use a paleogeographic rather than a tectonic definition of the Piemonte-Ligurian Ocean. Paleogeographically, it is an oceanic basin that bounds Adria to the west and north. But plate tectonically, it contains three plate boundaries: Adria-Iberia, Adria-Europe, and Europe-Iberia. So the spreading has different rates and timing and direction between Iberia and Adria, and between Europe and Adria, and it may be a good idea to treat those separately. Iberia is for instance irrelevant for the opening of the Valais-Magura ocean.

l. 25: How do you subdivide between subduction and collision? There is only subduction, of oceanic and continental lithosphere, and then convergence stops, or flips to Adria subduction/underthrusting. Collision is a rather vague term, since accreted oceanic and continental units behave similar and together form one nappe stack stripped from its lower crustal and mantle lithospheric underpinnings.

l. 25: How did you measure these distances? Net Adria-Europe convergence between 84 and 0 Ma is about 550 km or so, so are your distances measured along the motion path? Or do you invoke extension between Adria and Africa in the Late Cretaceous-Cenozoic? And if you measure along the motion path, then why do you start measuring at 84 Ma and not before? There is extensive evidence in the eastern Alps for nappe stacking and subduction since the Early Cretaceous already.

l. 26: Does extended continental material subduct or collide in your terminology? And

where do you put the difference? In addition: much of the oceanic portion subducted without accretion, so of the accreted material probably much less than 60% is derived from hyperextended crust I presume?

l. 27: What is your definition of an ophiolite? Ophiolites in the sense of Oman (or hyperextended versions with much thinner crust like in the Alps or southern Tibet) are upper plate-derived, and hence do not have to be exhumed. They're the uppermost structural unit. Accreted ocean plate stratigraphy, like the Valais sediments or the HP-LT metamorphic units derived from the PL Ocean, may exhume but are offscrapings from subducted crust. Do you call all these units ophiolites, or do you make a subdivision?

l. 28: Can you indicate what this importance is? Those margins just subduct and accrete their sediments to orogens right?

l. 34: If it subducted, there's no record of it. If there's a record, it accreted.

l. 39: I think the work of Geoff Mohn is also relevant here.

l. 40: But it didn't, did it? The highest structural unit of the western Alps are ophiolites, and the highest accreted units are PL ocean-derived, so subduction in the western Alps formed within the PL ocean, perhaps at the previous slow-spreading ridge or so, not on its hyperextended margins. In addition, plate motion went from extension to transform to subduction didn't it? So the 2D concepts of rifting and inversion don't really apply here.

(l. 44: I'd drop 'very' twice, text becomes more convincing if you don't use such emphasis.)

l. 45: Are there multiple episodes of slab break-off? I think there's only one, with one slab right? Davies and von Blanckenburg are relevant here.

l. 46: Why is it difficult (don't use the emphasis) to interpret? I find it very easy to interpret (although that interpretation may be wrong). And why is the curvature or the lack of a magmatic arc important to interpret tomography? Without that interpretation,

how do you know how many phases of slab break-off there were?

l. 47: Is it? Numerical modeling is a key tool in understanding the drivers of processes that of which we documented the result. Numerical models in no way constrain the structure or evolution of the Alps, nor how we should interpret seismic tomography. Numerical models are only helpful in interpreting what may have caused features that we documented. The papers you cite are not about the Alps, but about processes in which some of these papers use a highly simplified setup that they interpret to be somewhat relevant for the Alps.

l. 50: Which models need that, and for what purpose? If the direction of plate motion should be taken into account, then all numerical models for the Alps or Pyrenees that do 2D experiments go out the window since neither orogen resulted from 2D motion. The input parameters required for a numerical experiment depend highly on what that experiment tries to resolve.

l. 55: This introduction is a bit unfocused: how do overpressures change kinematic models? I have never used the inferred depth of burial in a kinematic reconstruction, but only structural overlap between nappes, and the stratigraphically constrained time interval of its subduction combined with constraints on plate convergence. Those are always enough to explain documented pressures, with or without overpressure. Do you use these pressures for your reconstructions, and if so, how do you date climax pressure and how do you correct for slab dip, etc?

l. 60: So is this paper about hyperextension, or the role of thinned crust in subduction, or overpressure? Or just kinematic reconstructions?

l. 64: During the Alpine orogeny, or during orogeny in the Alps? The Alpine orogeny is much larger than just the western Alps (you seem not to include the eastern Alps?)

l. 66: See earlier comment: I think you should check whether interpretations of geodynamic models are consistent with kinematics, not the other way around.

l. 76: This is not really a geological setting, but rather an overview of current interpretations on plate tectonic history. In addition, it is important here to identify the difference between Iberia-Adria and Europe-Adria motion. You seem particularly focused on the Alps part (Adria-Europe), but you also mention Alpine Tethys units from the Apennines-Maghrebides-Alboran region (Adria-Iberia).

l. 80: Adria is arguably part of a separate microplate today (but with a diffuse southern plate boundary), but for most of its history it is not a separate plate, also not in your reconstruction. So what do you refer to here, the continental realm of Adria?

l. 81: In the Alps, Adria is presently also downgoing plate, with the Southern Alps as offscraped crustal relics, isn't it? E.g. Ustaszewski et al 2008?

l. 81: it is not entirely surrounded by orogens today, the Adria continent has a passive margin in the south, bordering the ocean basin underlying the Ionian Sea.

l. 87: The upper plate is also Adriatic continental crust, so from this it follows that you consider Early Cretaceous subduction to occur within the Adriatic continent? I think that's inevitable, but it's not often explicitly mentioned this way.

l. 90: The closure of the Neotethys started in the Jurassic (170 Ma metamorphic soles and SSZ ophiolites in the Dinarides and Hellenides), but its closure didn't finish until the late Cretaceous.

l. 101-109: What are the underlying observations of this interpretation, and where were they made?

l. 110: where are these cherts found? What is the observation that shows that the cherts were deposited on a distal margin? What does the 166±1 Ma age represent, cherts in a coherent sedimentary sequence from clastics interpreted as syn-rift to open marine chert sedimentation? Or are these simply the oldest reported cherts?

l. 112: Vissers et al J Geol Soc 2017 reported ArAr ages from shear zones in Cap de Creus that they interpreted as extension-related normal faults in the Iberian margin.

These ages are 175 and 159 Ma, consistent with ages Etheve et al 2016 for extension in the Gulf of Valencia. So the end of rifting may not everywhere by 166±1 Ma.

l. 119: See also the ages in this range in the Betics (e.g., Puga et al 2011 and refs therein).

l. 128: the Jura mountains are not the Alps right? The Brianconnais became separated from Europe by the Valais ocean basin, so is not the European margin.

l. 131: No, this is not commonly accepted, this is commonly speculated. But this can be tested, through paleomagnetism, and those data are grossly inconsistent with that speculation. Why do you present this as a fact, ignoring our arguments for an alternative? It's fine if you disagree, but it's not that we just threw out some nonsensical armwaving that is best shoved under the rug I believe.

l. 136: Sorry, but please provide the geological documentation instead of the armwaving. The Organya basin is bounded to the south by what is now the Montsec thrust, and which originally was a listric normal fault if you restore the stratigraphy. This is not a pull-apart basin. The North Pyrenean fault is a post-84 Ma structure along which oceanic/hyperextended units are exposed, and it is far from certain what its structural evolution is. The 'en echelon pull-apart basins' is very much an interpretation of currently highly deformed relics within the Pyrenean fold-thrust belt. To give the reader an opportunity to make up his or her own mind, I would like to ask you give the geological facts alongside the major interpretations. I also don't understand why you present this model as a fact, and ignore the more recent (data-based) discussions on Iberia as well as its connection (or absence thereof) to Sardinia-Corsica, and hence Brianconnais.

l. 140: But this is not required by that stratigraphic age. The opening of the Valais ocean is easily explained without a connection to Iberia, in a way that is consistent with paleomagnetic data.

l. 165: The Austro-Alpine units are not intra-oceanic. They're entirely intracontinental.

[Figure]

They're continent-derived nappes shoved underneath Adria. The oldest HP metamorphism that I am aware of in PL-derived units are 83 Ma or so on Corsica, and the Sesia fragment (also continental) gives an 85 Ma or so age from HP units.

l. 170: Handy et al 2010 is relevant here.

l. 176: Why is the underthrusting of the European margin 'collision' and of the Brianconnais continent 'subduction' if both are thrusted below an upper plate, make nappes, and cause metamorphism?

l. 178: Is collision depending on whether a basin is under- or overfilled? When is a basin overfilled? Because there is Paratethys waters in the Alpine foreland into the Miocene, is that basin that underfilled? Why does HT metamorphism signal 'collision'? There is HT metamorphism in Japan, so is there collision there? Do you need slab break-off for a collision? There was slab break-off in California, but there is no collision. Sorry to fuss about details, but there is a large basket of apples and oranges here to call something by a term of which the implication is unclear.

l. 180: Also: where else are you referring to?

l. 181: No, there is upper plate extension in the Miocene. The 'roll-back' subduction started well before, like in the Alps, but was at the same rate as upper plate advance.

l. 181: What is indentation and how does it differ from collision?

l. 184: Can you explain what you mean here? It is kinematically decoupled because of the opening of the Pannonian Basin, or because of a plate boundary in the Dinarides?

l. 23: All of the previous in this paper were kinematic scenarios (without the debates). I suggest you present the kinematic scenarios first, and then provide the facts, or the other way around, but in the previous 6 pages you have presented models and scenarios as geological facts, so what is a reader supposed to get from the rest of the paper? I have a hunch what your conclusions are going to be, regardless of the debates that you are going to outline. You have already described your model above.

l. 191: For which time interval? Adria is at present moving relative to Africa, has moved relative to Africa during Ionian Basin opening. And at other times it did not.

l. 194: This is a strange argumentation. There is little deformation in that basin because data show that there is little deformation in that basin, not because it is oceanic. If it is oceanic and there is still deformation, then there is still deformation.

l. 195: There is a rich debate on the opening of the Alpine Tethys. I followed Speranza, but he is far from the only one who has made an interpretation of this.

l. 197: How limited?

l. 203: 118 Ma according to which timescale?

l. 204: replace 'Spain' by 'Iberia'. They also come from Portugal.

l. 206: Again, indicate what the basis for this interpretation is. The hyper-extension interpretation is first and foremost based on the presence of subcontinental mantle rocks in the NPZ. And it makes a lot of sense to interpret those as hyperextended. Gabbros in those rocks contain Sm/Nd ages of 170 Ma, and there are Jurassic marine sediments that suggest rifting of that time, but also ArAr ages of ∼105-100 Ma HT metamorphism and associated volcanism that affects the mafic rocks and overlying sedimentary rocks. The papers you cite interpret the 105 Ma HT metamorphism as related to hyperextension-related (no such HT records in any other hyperextended margins or ophiolites that I'm aware of), and thus discard the paleomagnetic data (the largest paleomagnetic dataset of any continent, ever) because those are not possible to reconcile with extension in the Pyrenees. None of the authors you cite explain those data. However, those data are explained as HT metamorphism following slab break-off (you know, the argumentation you cite for the Alps as dating collision). I admit that in my reconstruction of Iberia I predict convergence in the PL ocean during rotation – I shamefully acknowledge that I had not realized this and it invites reinterpretation because I am not aware of evidence for this. But this does not mean that the paleomagnetic data are wrong. Data are data. And you seem to throw them overboard without explaining them.

l. 209: Dated how? And how is this relevant for the Alps? The Alps form at a different plate boundary.

l. 213: It doesn't help the debate if you armwave at some possible explanations. Could you explain (because Neres does not) how you can remagnetize rocks across Iberia in such a way that those data give a larger rotation than Iberia ever underwent? And what does 'incompatibility with the GAPWAP' mean? The fact that the Iberian APWP is not the same as the GAPWAP in Eurasia, African, or North American coordinates is the argumentation that it was a separate plate that rotated relative to all three. Nothing of what you write here provides an explanation for the data and does not explain to the reader what the discussion is about. You're just choosing one model over the other without explaining why.

l. 215: The data from Iberia come from all over Iberia. How do you explain a rotation in the Aptian documented in Central Iberia, in SW Portugal, and in the South Pyrenean basins by intraplate deformation possibly along the Ebro Block? How do local rotations in the Ebro block explain that data that are collected all along the 100's of km long Messejena dyke from the CAMP align perfectly with data from dykes in the Moroccan Atlas and from Canada in the Vissers and Meijer fit, but misalign with the Jammes fit (Ruiz-Martinez et al., EPSL 2012). I am entirely open to alternative explanations, but sorry, we're not bringing science forward if we only throw some interpretations on the table and pick a few that we like better.

In addition: if you throw out marine magnetic anomalies from the North Atlantic, as well as paleomagnetic data, you have no quantitative data left to reconstruct Iberia. Jammes etc don't use any quantitative data for Iberia, those reconstrutcions are entirely qualitative. How are you going to make a reconstruction of the Alpine Tethys then, if you have no data to reconstruct Iberia?

l. 220: But how did these authors estimate those widths? I think what you are comparing here are estimates of the amount of extension (Vissers et al) with estimates of the width of the basin (Handy) and those are different things in the first place.

l. 224: Can you indicate what the kinematic constraints were of Vissers and Handy?

l. 225: Nowhere have you indicated that there is a long-lasting controversy on the width of the PL ocean. I don't really think there is one, everyone agrees it's a few hundred km, there's no space for more.

l. 234: Why do you need shortening in the Dinarides to reconstruct the Alps or the PL Ocean?

l. 235: You make it sound like you're the first to use the Atlantic to reconstruct the PL ocean, but everyone has done this so far. Frisch, Stampfli, Vissers, Dercourt, Rosenbaum, Dewey. Why do you need to do this again?

l. 245: Fine, but I believe I have done exactly this in my 2020 paper. And I may be entirely wrong, but it may be informative for a reader to indicate in the introduction what the rationale is to do this again.

l. 270: and violates all constraints from paleomagnetism except for the 5% outliers that Barnett-Moore does find reliable.

l. 273: none of these FT belts constrain Corsica or Sardinia.

l. 277: How does the debate on Iberia affect Adria reconstructions?

l. 287: Well, that entirely depends on your model for Iberia, and since you don't use quantitative constraints for Iberia, you can pretty much choose what you want. Besides, none of this argumentation provides an explanation for the paleomagnetic data from Sardinia.

l. 301: But this creates one hell of a Cretaceous subduction zone between Sardinia/Corsica in the early Cretaceous, as shown by Schettino and Turco. Do you have

evidence for that?

l. 315: In your response to Stefan Schmid's comment you call this 230 km of strike-slip in Adria a 'simplification'. But that is not a simplification. It's a complication. It would be a simplification if there are 5 faults with a cumulative displacement of 230 km that you summarize by one fault. This overlap is an artifact of the speculation that Corsica and Sardinia are Iberia.

l. 268: Why do you reconstruct Tisza-Datca in this paper? It's off-topic, and you have provided no information on this region in the paper so far.

l. 369: How can this possibly be derived from the Adriatic plate? The Adriatic 'plate', which is not a plate in most of the time you reconstruct it, is the upper plate below which all these units accreted. So it was during accretion part of the Eurasian plate. It may have been part of the Adriatic continent, but not of the Adriatic plate.

l. 377: And then in 2020 I did not speculate about it like Schmid and Vissers did and made a reconstruction using structural geological and paleomagnetic data and showed that this interpretation is impossible. I'd say either make an analysis of this region and properly restore it, or leave it out.

The rest of the paper contains a description of the model, and if the model is correct, the discussion is logical. I don't quite buy the reconstruction choices, so I refrain from discussing the implications of the model, I don't object to the first-order conclusions of slowspreading in the PL Ocean.
* * *

---

## Referee Comment (RC2) · Andrea Argnani (Referee) · 11 Nov 2020

General comments The paper of Le Breton and co-authors aims at presenting an updated kinematic reconstructions of the Alpine-Mediterranean area, with a focus on Corsica-Sardinia-Adria, implemented within a recent global plate model. Kinematic scenarios are tested for geodynamic consistency using thermo-mechanical modelling of the rifting phase and compared with geological records from the Alpine region s.l. Some interpretative choices strongly mark the paper. I am not commenting the adopted motion of Iberia, a debated issue with various models present in the literature, but the choice of having the Corsica-Sardina block attached to Iberia is not the most popular. Besides the palaeomagneti evidence which is against it, it seems that this choice causes some inconsistencies in the presented kinematic reconstructions (Fig. 4). Some are described below. but the most relevant is the necessity of displaced northern Adria along a lithospheric fault for which there is no geological evidence. The reconstructions by Le Breton et al. present significant critical points that perhaps should be better discussed and compared with previous reconstructions, but above all they should be better supported by geological evidence. For this, I think the paper requires substantial revision work. Some of the critical issues are in the specific comments below.

Specific comments Section 2. Geological setting line 80: The microplate nature of Adria is debated. It not certain whether, when and for how long it was a microplate. Perhaps using just the term Adria avoids any questions. lines 127-128: It is not clear whether the Brianconnaise is considered a microcontinent (as CSB) or as an extensional allochthon. It should be clarified. lines 131-132: I believe that alternative interpretation are equally possible, without kinematically linking the Bay of Biscay and the Valais. lines 153-156: from the reconstructions in Fig. 4 it seems that the Valais ocean opened at 130 Ma; the Jurassic opening is not obvious. Moreover, the location of the Valais to the north of the CSB looks a bit strange. This domain is still there at 83 and 67 Ma and even at 35 Ma, always between Eu and the CSB. I am not aware of evidence of remains of an ocean in Provence. Moreover in the 35 Ma frame a N-ward subduction is present north of the CSB; that subduction disappears in the subsequent 20 Ma frame (are there remains of it somewhere?), where subduction jumps to the south of the CSB. line 162: why subduction initiation is intra-oceanic? Most authors consider that subduction initiated at the southern margin. lines 196-197: the extension in the Strait of Sicily may not reflect the motion of Adria, but it could be related to the dynamics of the subducted African slab (e.g., Argnani, 2009 SP Geol. Soc. London).

Section 3.3 MATF: There is some confusion when describing the tectonics of the Adriatic Sea. The boundary inferred by DAgostino et al. (ca. E-W trending) is just sketched

and is not related to a specific structure or set of structures. It refers to the present tectonic activity, and that's why a proper boundary is not yet developed... if there will ever be a plate boundary. The Mid-Adriatic Ridge of Scisciani and Calamita is NNW-SSE trending geological feature. It was previously named Central Adriatic Deformation Belt (Argnani and Frugoni, 1997) and is a belt of foreland inversion structures where the Adriatic seismicity tends to concentrate. It has also been considered as the inland continuation of a major transform fault (Argnani, 2009, Bull Soc Geol It). There is no evidence, however, of a major strike-slip displacement, and Scisciani and Calamita describe inversion not strike-slip structures. The age of this deformation ranges from Quaternary to possibly Eocene, whereas the activity of the large strike-slip fault (MATF) used in the reconstruction ranges from Late Cretaceous to Eocene. In addition, the authors use the poles of Northern Adria from Schettino et al. which considered Adria decoupled along the E-W-trending Mattinata Fault, located in the southern part of the Gargano promontory. This fault shows evidence of Paleogene activity (Argnani et al., 2009, GSA) but the amount of displacement is unlikely to be on the order of 100s km.

Section 3.4 From Corsica to the west there is no evidence of an Alpine subduction followed by an Apennine-Maghrebian subduction. The two-subduction model adopted is one of the possible interpretations and other authors, more or less explicitly, prefer two opposite-vergence subductions since the beginning of convergence, handling in different ways the interpretation of Alpine Corsica (e.g., Jolivet et al., 1998, Argnani 2012 Tectonoph.) Critical in the two-subduction model is the presence of the AlCaPeCa (micro) continent, that is represented at 67 and 35 Ma, with a size of about twice the Corsica-Sardinia Block. I found intriguing that the CSB is still almost intact within the Mediterranean, whereas the larger AlCaPeCa micro continent has been completely dismantled. line 380: explaining the Eo-Alpine orogeny using an intra-continental subduction that is part of strike-slip system linking the PL to the Vardar seems an ad hoc solution not based on evidence. (also for lines 407-408)

Section 4.3 The authors take a geological section across the southern Alps to constrain
the initial with of the rift, which is estimated to be ca. 300 km. This rift system has a Beta < 1.5, which is typical of continental rifts that do not evolve to oceanization. As described by many authors, including Froitzheim and Eberli, the rifting leading to oceanization is located further to the west, in present coordinate. Assuming that also this sector was affected by the first stage of rifting, the initial width of rifting was larger than 300 km. Otherwise, the western sector was affected only by the second stage of rifting, but this does not fit the inferred evolution which is at the base of the numerical modelling. This is actually a minor point for the paper, though the mechanical behaviour that controls the location of oceanization is an interesting issue.

Section 4.4 The portions of exhumed mantle that crop out in the Western Alps are considered of subcontinental origin (e.g., Piccardo and Guarnieri 2010, Int. Geol. Rev.), as in type 1 margins of Huismans and Beaumont, 2011 . Does that fit with the result of the numerical modelling? In the model of Fig. 7 it seems that it is a newly formed lithospheric mantle to be exhumed

Section 5.2 lines 600-603: the opening of the Ionian basin in a NW-SE direction, with Malta and Apulia escarpments acting as transform faults, contrasts with using the Malta and Apulia as conjugate margins.

Section 5.3 The max. width of the oceanic domain in PL is taken as 250 km, based on the results of numerical modelling that also sets a length of 120 km for the hyperextended domain (80 + 40 km) and a length of 110 km for the necking domain (65 + 45 km). This gives a width of 480 km for the PL basin. Plate kinematics describe 680 km of convergence, that is subdivided in 420 km subduction and 260 km collision, based on the geologically inferred age of subduction and collision. With subduction initiating at the SE PL margin, and assuming that subduction occurs at the NW tip of the necking zone, after 370 km the conjugate necking zone is entering subduction: would that be considered onset of collision? The numerical modelling is reproducing the various elements of the PL margins, and continental material was certainly subducted. However, I suspect that mixing geological dates of subduction and collision, amount

of convergence from plate kinematics and time frames from numerical modelling may lead to quantitative results that are not really representative. How representative is the estimate of the amount of subducted continental material?

Figure 4. Some of the reconstructions are puzzling, and many features are not really supported or described in Section 3. Sudden shifts in plate boundaries among frames seem not always justified. The frames from 130 to 35 Ma, in particular, present some intriguing aspects. From 130 to 83 Ma a major plate boundary rearrangement is depicted, with a system of large strike-slip lithospheric faults in the Adria region. I don't see much geological evidence for these features, and the connection between Alpine Tethys and Vardar looks a bit forced. The Valais ocean is positioned between CSB and Europe; such a reconstruction is difficult to support as described above. The MATF that is represented in the frames 83 and 67 Ma has also some problems. as commented above.

Minor corrections Strike slip motion along various fault systems is often mentioned throughout the text; the sense of motion, however, is almost always not indicated. Where possible this is a useful indication. line 23: does the 250 km width refer to the truely oceanic part or does it include the exhumed mantle and hyperextended margin sectors too? The term ocean can be ambiguous. It looks it refers to the truly oceanic part, but this point should be clarified. line 86: Gawlick and Missoni is not in the References list. line 90: Channell and Kozur 1997 should be cited as an early paper describing the oceanic branches. line 371: Handy et al is 2015 line 382: Handy et al is 2015 line 460: "154-145 Ma and... 145-130 Ma". Ma and not km line 470: indicate which is the Brune et al 2017 cited line 878: Handy et al is 2015 Fig. 5: slight increase in velocity: "light to dark blue" instead of "dark to light blue"

---

## Author Comment (AC2) · 4 Jan 2021

Response to Review #1 by D. J. J. van Hinsbergen

We thank Douwe J.J. van Hinsbergen for his review, which highlights the active debate on this topic, especially for the motion of Iberia. Their reconstruction published in 2020 is an amazing compilation of geological and kinematical data for the entire Mediterranean realm. It is not our aim to propose such a reconstruction again. Our aim is to study the detailed kinematics of opening of the Piemont-Liguria Ocean and to quantify precisely the former extent of the PL Ocean and its rifted margins between Europe and Northern Adria. We however disagree with their model for the motion of Iberia

and Sardinia as it implies more than 500 km of convergence between Iberia and Sardinia. Instead, we present here an alternative scenario and show that this scenario is in good agreement with records of rifting and subduction in the Alps and that it is thermo-mechanically viable. We do believe that our work provides new quantitative estimates and alternative view that are useful for the scientific debate to better understand the geodynamic evolution of this area.

Please find attached a detailed response to all comments.

Sincerely

Eline Le Breton, on behalf of all co-authors

Please also note the supplement to this comment:
https://se.copernicus.org/preprints/se-2020-161/se-2020-161-AC2-supplement.pdf

---

## Author Response (AR1)

Dear Editor,

We would like to hereby submit the corrections of our manuscript entitled "Kinematics and extent of the Piemont-Liguria Basin – implications for subduction processes in the Alps" (#se-2020-161).

We thank Andrea Argnani and Douwe van Hinsbergen for the detailed reviews and constructive comments. The main critics raised by the reviewers are the kinematic assumptions we choose for Iberia and Sardinia-Corsica. **This is heavily debated among the scientific community and we clearly disagree with Douwe van Hinsbergen on this topic. We have made significant changes to the manuscript to address this issue. Section 2 now focuses only on geological facts and section 3 on kinematic scenarios and assumptions of our model.** In this section, we discuss current debates for Iberia-Sardinia along with arguments to support our kinematic model. We also changed Figure 3 to visualize the arguments on which we base our reconstruction.

The main argument is that models based on paleo-magnetic data from Iberia and Sardinia, such as the recent one of van Hinsbergen et al. (2020), would imply more than 500 km convergence between Iberia and Sardinia, of which there is absolutely no geological evidence. To avoid this, we rather overestimate the amount of motion along strike-slip faults that do exist and are supported by geological evidence.

**Regarding the debated strike-slip motion along the Mid-Adriatic Ridge, we followed Andrea Argnani's comment and changed the location of the fault to the Mattinata Fault along the Gargano Promontory, and thus follow strictly the kinematic model proposed by Schettino & Turco (2011).** We agree with the reviewers and Stefan Schmid's comment that the amount of displacement (230 km) implied by this kinematic scenario along this strike-slip fault is most likely an overestimation. We believe the deformation is much more diffuse within the Adriatic Plate as other works show Cretaceous reactivations of pre-existing rift-related structures at various places from the Central Apennines to the Southern Alps. However, it is very challenging to constrain the exact amount and timing of this deformation phase and incorporate it into the kinematic model, therefore we have to make some simplification. Future work should look at more diffuse intraplate deformation within Adria, as well as Iberia, which may help to solve the debate on the paleo-magnetic data. Our kinematic model can be viewed as an end-member model that implies a maximum amount of strike-slip displacement along the MF Fault within Adria and between Europe and Iberia-CSB, but fits best the paleo-geography of Adria-Sardinia-Calabria (Molli et al. 2020) and subduction record of the PL Basin. These arguments are carefully formulated and discussed in section 3.4 of the revised manuscript.

**Despite ongoing debates within the community, we would like to stress that the motions of Iberia (post-145 Ma, opening of Bay of Biscay; Tugend et al. 2014) and within Adria (100-40 Ma; Schettino and Turco, 2011) do not influence the opening of the Piemont-Liguria Basin, which is earlier (200-145 Ma) and therefore do not influence our main results and interpretations.** Moreover, our kinematic model is implemented into a global plate model that includes evolving/deforming plate boundaries through time (Müller et al. 2019). All reconstructions and rotation files, the plate boundaries, oceanic age grid etc. are publicly available. We thus believe that our work provides new quantitative constraints on the

opening of the PL basin, as well as amount, rate and direction of plate convergence between Europe and Adria, that are crucial for geodynamic modelling and more generally for better understanding of orogenic processes in the Alps. We are therefore convinced that this paper fits very well into this Special Issue of Solid Earth.

Sincerely
Eline Le Breton, on behalf of all co-authors

**Responses to reviewers' comments below:**

**Response to Review #1 by D. J. J. van Hinsbergen: p. 3-17**

**Response to Review #2 by Andrea Argnani: p. 18-24**

**Response to Review #1 by D. J. J. van Hinsbergen:**

We thank Douwe J.J. van Hinsbergen for his review, which highlights the active debate on this topic, especially for the motion of Iberia. Their reconstruction published in 2020 is an amazing compilation of geological and kinematical data for the entire Mediterranean realm. It is not our aim to propose such a reconstruction again. Our aim is to study the detailed kinematics of opening of the Piemont-Liguria Ocean and to quantify precisely the former extent of the PL Ocean and its rifted margins between Europe and Northern Adria. We however disagree with their model for the motion of Iberia and Sardinia as it implies more than 500 km of convergence between Iberia and Sardinia. Instead, we present here an alternative scenario and show that this scenario is in good agreement with records of rifting and subduction in the Alps and that it is thermo-mechanically viable. We do believe that our work provides new quantitative estimates and alternative view that are useful for the scientific debate to better understand the geodynamic evolution of this area.

I find it difficult, however, to see why this paper would be better than previous attempts to restore the Mediterranean region for the following reasons.
1. The authors provide no reason in their introduction why they found it necessary to make a new reconstruction and how their reconstruction systematically differs from previous attempts, or why they expect fundamentally different conclusions for the evolution of the Alpine Tethys with their approach.

Our model brings new quantitative estimates on the kinematic of opening of the PL Ocean, its former spatial extent including the width of the hyper-extended margins, which is new and crucial to understand the geological and geodynamic processes of subduction and exhumation in the Alps, as well as for rifting processes along magma-poor rifted margins. We clarified our motivation and goal in the introduction (lines 62-79).

2. The paper cites almost no geological data, but instead hinges almost entirely on interpretations and models. They don't describe the geological record of the Alpine Tethys, but the interpreted paleogeography. In my detailed comments below, I point this out for specific cases. For instance, they describe 'en echelon transtensional basins' in the Pyrenees. That is not what you see in the Pyrenees, you see a faulted, folded, intensely deformed, partly metamorphosed bunch of rocks that were interpreted to have been deposited in such basins. They describe a distal passive margin, or a hyperextended ocean floor, but the underlying observations or even the location where those are made are not specified. The model of Iberian motion is given as fact before alternatives and discussions are provided. This makes it very difficult for a reader to judge the validity of the final model, which to me rather seems a model based on selected models.

Everyone's new work has to rely to some degree on previous studies. It is not the aim of this paper to present entirely new kinematic reconstructions. But we do indeed present a compilation/selection of existing models, with some update (for Sardinia-Corsica), which in our opinion fit best the geological record of extension and subduction in the Alps. We now separated more clearly the geological facts, with added geological descriptions and locations (section 2), from kinematic scenarios/interpretation (section 3). In the latter, we discuss in more details possible alternative scenarios, limitation of our own work and arguments that we believe support our kinematic choices.

3. But my main problem lies in the inconsistency in argumentation. I believe this is best illustrated by the following example:

The Valais Basin in the Alps is interpreted from an intensely deformed and metamorphosed series of deep-marine Mesozoic sediments and associated mafic and ultramafic rocks. These mafic rocks have crystallization ages estimated with geochronology at Jurassic to Early Cretaceous consistent with the sedimentary record, which is what the authors interpret as the age of hyperextension. ArAr ages of these same rocks nowhere give Jurassic or Cretaceous ages, but Cenozoic, which they (and everybody else) interpret as related to exhumation following burial. Regional HT metamorphism is then taken by the authors as evidence for slab break-off following collision.

In the Pyrenees are also ultramafic and mafic rocks interpreted to reflect hyperextension. These are associated with Jurassic Sm/Nd ages and such Jurassic extension also follows from rifting recorded in the Bay of Biscay as well as in Pyrenean stratigraphy. Also here, ArAr ages are considerably younger, and follow upon HT metamorphism. In analogy to the Pyrenees, we have argued that the HT metamorphism is related to slab break-off following collision, pointed out where the slab is currently residing, and showed that that model is fully consistent with interpreted magnetic anomalies and paleomagnetic data. But the authors prefer for the Pyrenees to discard the evidence for Jurassic rifting, the magnetic anomalies, and the paleomagnetic data without explaining these data, and instead interpreting the HT metamorphism as hyperextension related.

There no inconsistency in our argumentation. The HT metamorphism and related tectonic structures in the Pyrenees are not the same as in the one mentioned in the Alps. In the Alps, it is a Barrovian-type metamorphism related to collision (Tauern Window, Lepontine Dome), with temperatures about 500-535°C and peak pressure of 8 kbar in the Tauern Window (upper greenschist to amphibolitic facies, the so-called "Tauernkristallisation") and above all associated with thickening of the European crust (Venediger Duplex formation), see for instance Scharf et al. (2013, Journal of Metamorphic Geology). It is in fact a medium T – medium P metamorphism (we corrected this in the text, lines 389-390). In the Pyrenees, the HT metamorphism is of higher temperature (above 600°C) and lower pressure (below 4 kbar) than in the Alps (Clerc et al. 2015). This HT metamorphism is associated with extensional structures, significant basin subsidence and syn-rift sedimentation, exhumation of (ultra)mafic rocks and alkali magmatism, which all tend to extensional tectonics, not collision. The HT-LP metamorphism (and associated structures/rocks) in the Pyrenees is thus not comparable with the metamorphic event in the Alps, and rather not consistent with a slab break-off model.

There is an existing debate on the paleomagnetic dataset of Iberia and the magnetic anomaly M0 and it is not the aim of our paper to solve that debate. Your model (van Hinsbergen et al. 2020) implies more than 500 km convergence between Iberia and Sardinia, for which there is no geological evidence. We therefore make a choice of an alternative kinematic scenario. We do not pretend to have the absolute answer, we agree that our model has limitations (i.e. the amount of strike-slip within Adria and between Europe and Sardinia-Adria is most likely over-estimated) and future work is needed to solve this debated issue. However, in our opinion, our alternative scenario fits better the geological record of rifting and subduction in the Alps.

4. As a result of the models they follow for the Pyrenees, they make a model for Iberia that devoid of any quantitative kinematic constraints (all constraints are discarded), from which it follows that also their reconstructed Piemonte-Ligurian ocean is does not have a quantitative basis.

Of course, our model for Iberia is kinematically constrained (Barnet-Moore et al. 2018), as our kinematic study of the opening of the PL Basin. The different models for Iberia-Sar/Cor-Adria

influence their relative paleo-geographic positions in Mesozoic time and the kinematics of closure of the PL Basin but not its opening. The opening of the PL Basin is not kinematically dependent on the motion of Iberia but depends on the motion of Europe and Africa, i.e. the opening of the Central Atlantic in the Jurassic. We discuss in section 5.2 alternative scenarios for the Central Atlantic and Ionian Basin. Thus, the heart of our paper, which focuses on the kinematics of rifting and former spatial extent of the PL ocean and its margins is constrained kinematically and is not affected by the debated motion of Iberia, and strike-slip motion within Adria.

The model in this paper is mostly a compilation of previous models, and in my view adds little to the discussion. I don't mind if it's published, I disagree with this model as much as with the previous ones for reasons I published before and repeat in my review. The authors do point out, however, that my model predicts shortening between Sardinia and Iberia during Iberian rotation, for which I must have become blind when I was working on my version: there is indeed no evidence for this, and it bothers me, so I'll research it and see how I can solve it. But I believe that solving that should not result in ignoring all paleomagnetic data and invoking a 230 km strike-slip fault through Adria for which there is no evidence (and which, if I may add, is not a simplification as you respond to Stefan, but rather a complication).

Aren't all model to a large degree based on previous model, observations and interpretations? We mention from the beginning that we focus on Sardinia-Corsica-Adria only, the rest is indeed a selection and compilation of existing models. We rephrased this part of the introduction to make that clear (lines 75-77). We modified the text (sections 2 and 3) to present clearly the debated and alternative scenarios, arguments on which we base our selection. We furthermore discuss alternative scenarios in the discussion (sections 5.2 and 5.3).

There is geological evidence of strike-slip faulting within Adria, along the Mattinata Fault along the Gargano Promontory (we corrected the location, slightly more south than the Mid-Adriatic Ridge, following Schettino & Turco 2011 and Argnani et al. 2009). We added geological descriptions of these deformation (lines 850-859). We agree that the amount of slip (230 km) is most likely an overestimation. The deformation was surely more diffuse throughout the entire plate (as geological observations in the Apennines and Southern Alps show Cretaceous reactivations of pre-existing, Jurassic rift-related structures). Future work would need to focus on this intraplate deformation. For now, it is a kinematic simplification (and not a complication) to have the motion along only one plate boundary. We built deforming plate polygons and plate boundaries with GPlates (evolving topologies) back to 250 Ma (available in Müller et al. 2019 database), thus it would be a complication to have 5 or more plates and plate boundaries between southern and northern Adria to reconstruct through time, rather than just one plate boundary. Future work should quantify these intraplate deformation and implement them into kinematic models using deforming network/topologies.

**Response to detailed comments:**

The detailed comments below may help the authors to clarify their paper and bring their new message across better. I don't expect the authors to agree with me, but I do ask them to explain the basis for discussions, rather than to just choose models based on other models.

As soon as we received the second review, we will implement all the changes to reply specifically

**1. l. 16:** Should a kinematic model be geodynamically 'consistent'? Suppose that data show a kinematic pattern, but you're not able to model it because we currently lack the conceptual understanding of the process, should we then adjust the kinematic model? Or learn something about the process? I believe that kinematic models should be boundary conditions for dynamic models, not the other way around, don't you?

We can learn from geodynamic modelling in terms of first-order processes and their driving forces. In that sense, both tools are complementary.

**2. l. 22:** What is a bit confusing is that you use a paleogeographic rather than a tectonic definition of the Piemonte-Ligurian Ocean. Paleogeographically, it is an oceanic basin that bounds Adria to the west and north. But plate tectonically, it contains three plate boundaries: Adria-Iberia, Adria-Europe, and Europe-Iberia. So the spreading has different rates and timing and direction between Iberia and Adria, and between Europe and Adria, and it may be a good idea to treat those separately. Iberia is for instance irrelevant for the opening of the Valais-Magura ocean.

We specified "between Europe and Adria" (line 19).

**3. l. 25:** How do you subdivide between subduction and collision? There is only subduction, of oceanic and continental lithosphere, and then convergence stops, or flips to Adria subduction/underthrusting. Collision is a rather vague term, since accreted oceanic and continental units behave similar and together form one nappe stack stripped from its lower crustal and mantle lithospheric underpinnings.

We have now defined more clearly what we mean by subduction and collision (lines 381-392) and rephrased this sentence in the abstract (lines 25-26).

**4. l. 25:** How did you measure these distances? Net Adria-Europe convergence between 84 and 0 Ma is about 550 km or so, so are your distances measured along the motion path? Or do you invoke extension between Adria and Africa in the Late Cretaceous- Cenozoic? And if you measure along the motion path, then why do you start measuring at 84 Ma and not before? There is extensive evidence in the eastern Alps for nappe stacking and subduction since the Early Cretaceous already.

We measure along the motion path (Ivrea relative to Europe) as shown in section 5.3, which provides the amount of convergence of Adria versus Europe. We discuss the discrepancy with your model (indeed showing ca. 550 km convergence) in section 5.3. The motion before 84 Ma is much more oblique, we focus here on the amount of North-South convergence that initiated at about 84 Ma and was responsible for the formation of the (western) Alps.

**5. l. 26:** Does extended continental material subduct or collide in your terminology? And where do you put the difference? In addition: much of the oceanic portion subducted without accretion, so of the accreted material probably much less than 60% is derived from hyperextended crust I presume?

We define what we mean by subduction and collision in section 2 (lines 381-392). What we mean here is the proportion of the "true" oceanic width of the PL Basin (250 km) represents only 35% of the total amount of plate convergence along the same transect (680 km). So overall, at least 65 % of what went into subduction/collision during plate convergence was extended continental material. We rephrased this sentence in the abstract.

**6. l. 27:** What is your definition of an ophiolite? Ophiolites in the sense of Oman (or hyper-extended versions with much thinner crust like in the Alps or southern Tibet) are upper plate-derived, and hence do not have to be exhumed. They're the uppermost structural unit. Accreted ocean plate stratigraphy, like the Valais sediments or the HP-LT metamorphic units derived from the PL Ocean, may exhume but are offscrapings from subducted crust. Do you call all these units ophiolites, or do you make a subdivision?

We changed "Ophiolites" by "units".

**7. l. 28:** Can you indicate what this importance is? Those margins just subduct and accrete their sediments to orogens right?

This shows quantitatively the significant proportion of continental material involved in subduction in the Alps. This has important implications to better understand subduction and exhumation processes.

**8. l. 34:** If it subducted, there's no record of it. If there's a record, it accreted.

We changed this sentence accordingly ("subducted to ultra-high pressure and later exhumed", line 34).

**9. l. 39:** I think the work of Geoff Mohn is also relevant here.

We do quote Geoffroy Mohn's work (Mohn et al. 2010, Beltrando et al. 2014 and Masini et al., 2014).

**10. l. 40:** But it didn't, did it? The highest structural unit of the western Alps are ophiolites, and the highest accreted units are PL ocean-derived, so subduction in the western Alps formed within the PL ocean, perhaps at the previous slow-spreading ridge or so, not on its hyperextended margins. In addition, plate motion went from extension to transform to subduction didn't it? So the 2D concepts of rifting and inversion don't really apply here.

The first unit entering subduction is the Sesia Zone along the Adriatic continental margin (e.g. Rubatto et al. 1998; Manzotti et al., 2014, 2018).

**11.** (**l. 44:** I'd drop 'very' twice, text becomes more convincing if you don't use such emphasis.)

We removed "very".

**12. l. 45**: Are there multiple episodes of slab break-off? I think there's only one, with one slab right? Davies and von Blanckenburg are relevant here.

Here we mention the entire Alpine Chain, thus this covers slab break-off events beneath the different orogens (Alps, Carpathians, Dinarides, Apennines).

**13. l. 46**: Why is it difficult (don't use the emphasis) to interpret? I find it very easy to interpret (although that interpretation may be wrong). And why is the curvature or the lack of a magmatic arc important to interpret tomography? Without that interpretation, how do you know how many phases of slab break-off there were?

We switch "lack of magmatic arc" and "episodes of slab-breakoff" in the previous sentence so it's clearer that the interpretation of seismic tomography model is difficult due to the complex slab evolution beneath the Alpine Chain. Several new models are being produced in the light of the new high resolution seismic network of the AlpArray. Their interpretation is complex in some cases, as shown for example during GeoUtrecht 2020 (see for example the presentation of Paffrath et al.):

https://www.conftool.pro/geoutrecht2020/index.php?page=browseSessions&form_session=164

**14. l. 47**: Is it? Numerical modeling is a key tool in understanding the drivers of processes that of which we documented the result. Numerical models in no way constrain the structure or evolution of the Alps, nor how we should interpret seismic tomography. Numerical models are only helpful in interpreting what may have caused features that we documented. The papers you cite are not about the Alps, but about processes in which some of these papers use a highly simplified setup that they interpret to be somewhat relevant for the Alps.

The papers quoted here are showing what we want to highlight: numerical models need quantitative constraints, such as plate convergence, width / thickness of oceanic crust and continental crust, that we aim to provide here. But on the other hand geodynamic modeling may help us understand the behavior of slabs at depth and thus, potentially, help us interpret seismic tomography models.

**15. l. 50**: Which models need that, and for what purpose? If the direction of plate motion should be taken into account, then all numerical models for the Alps or Pyrenees that do 2D experiments go out the window since neither orogen resulted from 2D motion. The input parameters required for a numerical experiment depend highly on what that experiment tries to resolve.

The models quoted before. The obliquity of the plate margin is often an important parameter in model setup. We agree that for the Alps modelling in 2D is a major limitation but still provide a better understanding of subduction/orogenic processes.

**16. l. 55**: This introduction is a bit unfocused: how do overpressures change kinematic models? I have never used the inferred depth of burial in a kinematic reconstruction, but only structural overlap between nappes, and the stratigraphically constrained time interval of its subduction combined with constraints on plate convergence. Those are always enough to explain documented pressures, with or without overpressure. Do you use these pressures for your reconstructions, and if so, how do you date climax pressure and how do you correct for slab dip, etc?

Previous studies did use inferred depth of high-pressure metamorphism to estimate amount of plate convergence (Schmid et al. 1996; Handy et al. 2010, 2015; references added line 59). We don't use it here.

**17. l. 60:** So is this paper about hyperextension, or the role of thinned crust in subduction, or overpressure? Or just kinematic reconstructions?

We modified the last paragraph of the introduction to specify more clearly our goal (lines 61-79).

**18. l. 64:** During the Alpine orogeny, or during orogeny in the Alps? The Alpine orogeny is much larger than just the western Alps (you seem not to include the eastern Alps?)

Yes, we focus here on the Western/Central Alps. We rephrased this sentence (line 75).

**19. l. 66**: See earlier comment: I think you should check whether interpretations of geodynamic models are consistent with kinematics, not the other way around.

See previous response.

**20. l. 76**: This is not really a geological setting, but rather an overview of current interpretations on plate tectonic history. In addition, it is important here to identify the difference between Iberia-Adria and Europe-Adria motion. You seem particularly focused on the Alps part (Adria-Europe), but you also mention Alpine Tethys units from the Apennines- Maghrebides-Alboran region (Adria-Iberia).

We changed sections 2 and 3. The part on plate kinematic scenarios / interpretations and their debate is now moved to section 3. We find it necessary to introduce and mention briefly the Apennines-Maghrebides-Alboran region, as well as AlCaPa-Tisza-Dacia units, as these units appear on the figures and their tectonic evolution is kinematically linked with our study area. We indeed focus on the evolution of Europe-Adria (and clarified that in the text) and provide references if the reader wants to find out more about those regions.

**21. l. 80**: Adria is arguably part of a separate microplate today (but with a diffuse southern plate boundary), but for most of its history it is not a separate plate, also not in your reconstruction. So what do you refer to here, the continental realm of Adria?

We refer to the present-day Adriatic plate as defined by present-day location of plate boundaries. We consider the Calabrian Arc and Kefalonia Fault to border the Adriatic Plate to the south, as drawn on Figure 1. We specified this in the text (line 93).

**22. l. 81**: In the Alps, Adria is presently also downgoing plate, with the Southern Alps as offscraped crustal relics, isn't it? E.g. Ustaszewski et al 2008?

This is still debated, most recent models from the AlpArray Working Group suggest otherwise, showing a European Moho dipping Adria and a clear detached slab at depth (see abstracts at AlpArray session of EGU2020 or GeoUtrecht 2020)

**23. l. 81**: it is not entirely surrounded by orogens today, the Adria continent has a passive margin in the south, bordering the ocean basin underlying the Ionian Sea.

We agree that the plate boundary to the south is unclear as it is covered by the accretionary prism of the Calabrian Arc, we removed "entirely" to avoid confusion.

**24. l. 87**: The upper plate is also Adriatic continental crust, so from this it follows that you consider Early Cretaceous subduction to occur within the Adriatic continent? I think that's inevitable, but it's not often explicitly mentioned this way.

Yes, we consider the Eo-Alpine Orogeny as intracontinental subduction as proposed by Stüwe and Schuster (2012) and wrote that clearly in section 2.

**25. l. 90**: The closure of the Neotethys started in the Jurassic (170 Ma metamorphic soles and SSZ ophiolites in the Dinarides and Hellenides), but its closure didn't finish until the late Cretaceous.

Indeed, we modified the sentence accordingly and added a reference for the age of metamorphic soles (Maffione & Van Hinsbergen, 2018; lines 102-103).

**26. l. 101-109**: What are the underlying observations of this interpretation, and where were they made?

We added more geological descriptions, locations and names of tectonic units of these different domains along the rifted margin (proximal/necking/hyper-extended, Central Alps; lines 116, 118, 119-126 and new reference Ribes et al. 2019).

**27. l. 110**: where are these cherts found? What is the observation that shows that the cherts were deposited on a distal margin? What does the 166±1 Ma age represent, cherts in a coherent sedimentary sequence from clastics interpreted as syn-rift to open marine chert sedimentation? Or are these simply the oldest reported cherts?

This was the oldest reported chert. We changed this paragraph to clarify what we meant (see next comment/response).

**28. l. 112**: Vissers et al J Geol Soc 2017 reported ArAr ages from shear zones in Cap de Creus that they interpreted as extension-related normal faults in the Iberian margin. These ages are 175 and 159 Ma, consistent with ages Etheve et al 2016 for extension in the Gulf of Valencia. So the end of rifting may not everywhere by 166±1 Ma.

We agree and that's what we meant with the following description of the OCT zones. We meant that deep-sea sedimentation (radiolarites) was usually interpreted as post-rift sedimentation, thus post break-up and onset of spreading. However, the recent advances of knowledge on rifting processes along distal part of rifted margins, and the formation of complex OCT zones, show that radiolarites may form during that stage and thus be interpreted as syn-rift along the distal part (OCT) zones. We clarify this (lines 129-131) and added a new reference (Ribes et al., 2019) that clearly show this diachroneity in post-rift sedimentation along the margin.

**29. l. 119:** See also the ages in this range in the Betics (e.g., Puga et al 2011 and refs therein).

We focus on the ages from Corsica, the Northern Apennines and the Alps. Puga et al. 2011 provide new ages for the Betics only. We added the reference on which their compilation is based (Bortolotti and Principi, 2005; line 139).

**30. l. 128:** the Jura mountains are not the Alps right? The Brianconnais became separated from Europe by the Valais ocean basin, so is not the European margin.

What we meant is that the Jura mountains (which are indeed separated from the Alpine chain by the Molasse Basin but are still related to the Alpine orogeny), preserve Mesozoic deposition but it's true it does not show the rifted-related structures that we describe for the Adriatic margin. We removed "Jura Mountains" from this sentence and focus on the Helvetic and Penninic nappes. We also reformulated for the Briançonnais as follow: "Another extended continental unit, the Briançonnais, was detached from Europe during the opening of another basin: the Valais Basin." (lines 145-146).

**31. l. 131:** No, this is not commonly accepted, this is commonly speculated. But this can be tested, through paleomagnetism, and those data are grossly inconsistent with that speculation. Why do you present this as a fact, ignoring our arguments for an alternative? It's fine if you disagree, but it's not that we just threw out some nonsensical armwaving that is best shoved under the rug I believe.

Response to 31.-32.-33: We added geological descriptions from the Pyrenees and Provence (lines 165-168 and 187-191) and removed the part on the potential kinematic link between the opening of the Valais and the Pyrenean rift system in Section 3, where we present debates and possible alternative scenarios and the arguments on which we based our choice.

**32. l. 136:** Sorry, but please provide the geological documentation instead of the armwaving. The Organya basin is bounded to the south by what is now the Montsec thrust, and which originally was a listric normal fault if you restore the stratigraphy. This is not a pull-apart basin.

The North Pyrenean fault is a post-84 Ma structure along which oceanic/hyperextended units are exposed, and it is far from certain what its structural evolution is. The 'en echelon pull-apart basins' is very much an interpretation of currently highly deformed relics within the Pyrenean fold-thrust belt.

To give the reader an opportunity to make up his or her own mind, I would like to ask you give the geological facts alongside the major interpretations. I also don't understand why you present this model as a fact, and ignore the more recent (data-based) discussions on Iberia as well as its connection (or absence thereof) to Sardinia-Corsica, and hence Brianconnais.

See response to point 31.

**33. l. 140:** But this is not required by that stratigraphic age. The opening of the Valais ocean is easily explained without a connection to Iberia, in a way that is consistent with paleomagnetic data.

See response to point 31.

**34. l. 165:** The Austro-Alpine units are not intra-oceanic. They're entirely intracontinental. They're continent-derived nappes shoved underneath Adria. The oldest HP metamorphism that I am aware of in PL-derived units are 83 Ma or so on Corsica, and the Sesia fragment (also continental) gives an 85 Ma or so age from HP units.

We rephrased this paragraph (lines 172-177) and removed "intra-oceanic".

**35. l. 170:** Handy et al 2010 is relevant here.

Reference added (line 183).

**36. l. 176:** Why is the underthrusting of the European margin 'collision' and of the Brianconnais continent 'subduction' if both are thrusted below an upper plate, make nappes, and cause metamorphism?

See following response.

**37. l. 178:** Is collision depending on whether a basin is under- or overfilled? When is a basin overfilled? Because there is Paratethys waters in the Alpine foreland into the Miocene, is that basin that underfilled? Why does HT metamorphism signal 'collision'? There is HT metamorphism in Japan, so is there collision there? Do you need slab break-off for a collision? There was slab break-off in California, but there is no collision. Sorry to fuss about details, but there is a large basket of apples and oranges here to call something by a term of which the implication is unclear.

These events when happening separately may indeed not indicate collision at all, but here they happen all contemporaneously (since ca. 35 Ma) indicating a major change in the orogenic process. Yes, the eastern part of the Molasse Basin was not overfilled, but in the western sector, there is a clear shift in the sedimentation that coincides with time with slab break-off, inferred from magmatism along the Periadriatic line. By HT metamorphism, we mean Barrovian type (Tauern Window) which is in fact medium T – medium P (we corrected the text) and is typical for collisional setting. We defined better this subduction/collision distinction in section 2.2, lines 191-204)

**38. l. 180**: Also: where else are you referring to?

Removed

**39. l. 181**: No, there is upper plate extension in the Miocene. The 'roll-back' subduction started well before, like in the Alps, but was at the same rate as upper plate advance.

We changed to "eastward retreat" (line 206).

**40. l. 181**: What is indentation and how does it differ from collision?

We meant with "indentation" extrusion tectonics accompanied with the development of large strike slip faults, as modelled by the indentation model of Tapponier et al. 1982 (see Ratschbacher et al. 1991). However, we don't want to enter in the definition of indentation in the text, as this is out of topic for our study. We thus changed to "collision" (line 207).

**44. l. 184**: Can you explain what you mean here? It is kinematically decoupled because of the opening of the Pannonian Basin, or because of a plate boundary in the Dinarides?

We removed this sentence.

**45. l. 23**: All of the previous in this paper were kinematic scenarios (without the debates). I suggest you present the kinematic scenarios first, and then provide the facts, or the other way around, but in the previous 6 pages you have presented models and scenarios as geological facts, so what is a reader supposed to get from the rest of the paper? I have a hunch what your conclusions are going to be, regardless of the debates that you are going to outline. You have already described your model above.

We followed your suggestion and present only geological facts in section 2 and discussed kinematic scenarios in section 3.

**46. l. 191**: For which time interval? Adria is at present moving relative to Africa, has moved relative to Africa during Ionian Basin opening. And at other times it did not.

We added "since the end of opening of the Ionian Basin" (line 332).

**47. l. 194**: This is a strange argumentation. There is little deformation in that basin because data show that there is little deformation in that basin, not because it is oceanic. If it is oceanic and there is still deformation, then there is still deformation.

This paragraph was clarified and is now moved to section 3.4 (Reconstructions of Adria, Sardinia-Corsica; lines 330-341).

**48. l. 195:** There is a rich debate on the opening of the Alpine Tethys. I followed Speranza, but he is far from the only one who has made an interpretation of this.

Alternative scenario for the Ionian Basin are discussed in section 5.2. We added a sentence here "We note that Tugend et al. (2019) recently suggested a Jurassic phase of opening for the Ionian Basin, which will be discussed in section 5.2." (line 351-352).

**49. l. 197**: How limited?

We added "30 km minimum across Pantelleria rift (CROP M25)" (line 339-340).

**50. l. 203:** 118 Ma according to which timescale?

Removed.

**51. l. 204**: replace 'Spain' by 'Iberia'. They also come from Portugal.

Changed.

**52. l. 206**: Again, indicate what the basis for this interpretation is. The hyper-extension interpretation is first and foremost based on the presence of subcontinental mantle rocks in the NPZ. And it makes a lot of sense to interpret those as hyperextended. Gabbros in those rocks contain Sm/Nd ages of 170 Ma, and there are Jurassic marine sediments that suggest rifting of that time, but also ArAr ages of ~105-100 Ma HT metamorphism and associated volcanism that affects the mafic rocks and overlying sedimentary rocks. The papers you cite interpret the 105 Ma HT metamorphism as related to hyperextension-related (no such HT records in any other hyperextended margins or ophiolites that I'm aware of), and thus discard the paleomagnetic data (the largest paleomagnetic dataset of any continent, ever) because those are not possible to reconcile with extension in the Pyrenees. None of the authors you cite explain those data. However, those data are explained as HT metamorphism following slab break-off (you know, the argumentation you cite for the Alps as dating collision).

The HT metamorphism and related tectonic structures in the Pyrenees are not the same as in the one mentioned in the Alps. There, it is a Barrovian-type metamorphism related to collision (Tauern Window, Lepontine Dome), with temperatures about 500-535°C and peak pressure of 8 kbar in the Tauern Window (upper greenschist to amphibolitic facies, the so-called "Tauernkristallisation") and above all associated with thickening of the European crust (Venediger Duplex formation), see for instance Scharf et al. (2013, Journal of Metamorphic Geology). It is in fact a medium T - medium P metamorphism (we corrected the text accordingly). In the Pyrenees, the HT metamorphism is of higher temperature (above 600°C) and lower pressure (below 4 kbar) than in the Alps (Clerc et al. 2015). This HT metamorphism

is associated with extensional structures, significant basin subsidence and syn-rift sedimentation, exhumation of (ultra)mafic rocks and alkali magmatism, which all tend to extensional tectonics, not collision.

I admit that in my reconstruction of Iberia I predict convergence in the PL ocean during rotation – I shamefully acknowledge that I had not realized this and it invites reinterpretation because I am not aware of evidence for this. But this does not mean that the paleomagnetic data are wrong. Data are data. And you seem to throw them overboard without explaining them.

We don't mean that the paleomagnetic data are wrong but that they are maybe alternative scenarios to explain them that need to be studied in the future. The convergence implied between Iberia and Sardinia in your model is too problematic thus we didn't follow it. We followed reconstructions of Europe, Africa and Iberia (Barnett-Moore et al. 2018) of the global plate model (Müller et al. 2019), which are of course kinematically constrained. We do not provide a final answer but test an alternative scenario. We show that this alternative fits geological record of kinematics and timing of rifting of the PL basin and is thermo-mechanically consistent. We furthermore provide quantitative constraints on the former size/extent of the rifted margin and "true" ocean (200-145 Ma), which are completely independent from the motion of Iberia and Sardinia (post 145 Ma).

**53. l. 209**: Dated how? And how is this relevant for the Alps? The Alps form at a different plate boundary.

It is relevant, because it shows that convergence in the Alps (south-directed "Alpine" subduction) and Pyrenees-Provence started at the same time (ca. 84 Ma).

**54. l. 213**: It doesn't help the debate if you armwave at some possible explanations. Could you explain (because Neres does not) how you can remagnetize rocks across Iberia in such a way that those data give a larger rotation than Iberia ever underwent? And what does 'incompatibility with the GAPWAP' mean? The fact that the Iberian APWP is not the same as the GAPWAP in Eurasia, African, or North American coordinates is the argumentation that it was a separate plate that rotated relative to all three. Nothing of what you write here provides an explanation for the data and does not explain to the reader what the discussion is about. You're just choosing one model over the other without explaining why.

Response to comments 54.-55.:

We admit that our model do not reconcile with the paleomagnetic data, which don't mean the data are wrong, but that they may be other way to explain them, i.e. with intraplate deformation within Iberia. We do not propose an absolute answer but present an alternative scenario that avoid convergence between Iberia and Sardinia. We show that this alternative scenario is in good agreement with paleo-geography of Sardinia-Calabria-Adria and record of subduction in the Alps. Moreover, in our model, the opening of the Bay of Biscay starts at about 145 Ma (Tugend et al. 2014) and thus does not affect the kinematic of opening of the PL Basin (200-145 Ma), which is the focus of our paper.

**55. l. 215**: The data from Iberia come from all over Iberia. How do you explain a rotation in the Aptian documented in Central Iberia, in SW Portugal, and in the South Pyrenean basins by intraplate deformation possibly along the Ebro Block? How do local rotations in the Ebro block explain that data that are collected all along the 100's of km long Messejena dyke from the CAMP align perfectly with data from dykes in the Moroccan Atlas and from Canada in the

Vissers and Meijer fit, but misalign with the Jammes fit (Ruiz-Martinez et al., EPSL 2012). I am entirely open to alternative explanations, but sorry, we're not bringing science forward if we only throw some interpretations on the table and pick a few that we like better.

See response to comments 54.

In addition: if you throw out marine magnetic anomalies from the North Atlantic, as well as paleomagnetic data, you have no quantitative data left to reconstruct Iberia. Jammes etc don't use any quantitative data for Iberia, those reconstrutcions are entirely qualitative. How are you going to make a reconstruction of the Alpine Tethys then, if you have no data to reconstruct Iberia?

See previous comments, we do not aim at providing an entirely new reconstruction. We used reconstructions – that are of course kinematically constrained – for Europe, Africa and Iberia (from Barnett-Moore et al., 2018), implemented in the global plate model of Müller et al. (2019). We updated the motion of Sardinia-Corsica, as well as of Adria (post 20 Ma), and the location of plate boundaries (evolving topologies) back to 250 Ma.

**56. l. 220:** But how did these authors estimate those widths? I think what you are comparing here are estimates of the amount of extension (Vissers et al) with estimates of the width of the basin (Handy) and those are different things in the first place.

Response to 56-57-58: We modified this paragraph and moved it to the introduction. Until now, there was no study providing clear quantitative estimates of the size of the ("true") ocean and the rifted margin of the PL Basin. We totally agree that kinematically there is no space for a large ocean, and we show that with precise kinematic constraints in this study. But often, in geological representations and geodynamic modelling set up, the ocean is quite large (at least 500 km of "true" ocean), which motivated this work. It has significant implications for understanding slab pull forces, rollback and exhumation of high pressure rock in the Alps.

**57. l. 224:** Can you indicate what the kinematic constraints were of Vissers and Handy?

See response to 56.

**58. l. 225**: Nowhere have you indicated that there is a long-lasting controversy on the width of the PL ocean. I don't really think there is one, everyone agrees it's a few hundred km, there's no space for more.

See response to 56.

 **59. l. 234:** Why do you need shortening in the Dinarides to reconstruct the Alps or the PL Ocean?

The Dinarides are useful to restore the past motion of Adria back to 20 Ma (Le Breton et al., 2017).

**60. l. 235**: You make it sound like you're the first to use the Atlantic to reconstruct the PL ocean, but everyone has done this so far. Frisch, Stampfli, Vissers, Dercourt, Rosen- baum, Dewey. Why do you need to do this again?

We rephrased this sentence (lines 220-221), we didn't re-interpret the magnetic anomalies of the Atlantic. This paper is part of a multi-disciplinary Special Issue on the Alps (AlpArray), covering geophysical to geological studies, thus this paragraph aims at introducing the general approach of kinematic reconstructions, applied to the Alpine-Mediterranean area.

**61. l. 245**: Fine, but I believe I have done exactly this in my 2020 paper. And I may be entirely wrong, but it may be informative for a reader to indicate in the introduction what the rationale is to do this again.

Your model indeed fits the paleo-magnetic data of Iberia and Sardinia, however implies significant convergence (500 km) between Iberia and Sardinia in Lower Cretaceous time, as well as convergence between Iberia and Europe at the time of extensional tectonics and sedimentation in the Pyrenees. For these reasons, we choose a different model for Iberia-Sardinia-Corsica, and thus for Adria (split in two following Schettino and Turco, 2011). We now explain in more detail alternative models, our selection and the limitations/flaws of our model in Section 3.1 (Figure 3 is also changed). Plus, we included our regional model into a global model that includes lithospheric deformation, i.e. with deforming topologies and plate boundaries back to 250 Ma, which is new.

**62. l. 270**: and violates all constraints from paleomagnetism except for the 5% outliers that Barnett-Moore does find reliable.

See previous reply above, we now explained better alternative scenarios, our assumptions/selection and the limitation of our model. Our model does not satisfy the paleo-magnetic data, however it fits the geological record of the kinematics and timing of rifting and subduction in the Alps.

**63. l. 273**: none of these FT belts constrain Corsica or Sardinia.

We disagree. The Provence Fold-and-Thrust Belt most certainly constrains the past motion of Corsica and Sardinia relative to Europe. We modified this sentence to clarify (line 344).

**64. l. 277**: How does the debate on Iberia affect Adria reconstructions?

Removed.

**65. l. 287:** Well, that entirely depends on your model for Iberia, and since you don't use quantitative constraints for Iberia, you can pretty much choose what you want. Besides, none of this argumentation provides an explanation for the paleomagnetic data from Sardinia.

This paragraph regards the post-35 Ma motion of Corsica-Sardinia, which doesn't depend on the model used for Iberia but on the opening of the Liguro-Provencal Basin as explained here. You probably mean for pre-35 Ma motion of Corsica-Sardinia. This is explained in the following paragraph (now lines 361-373).

The model we use for Iberia is of course kinematically/quantitatively constrained (Barnett-Moore et al. 2018).

**66. l. 301**: But this creates one hell of a Cretaceous subduction zone between Sardinia/Corsica in the early Cretaceous, as shown by Schettino and Turco. Do you have evidence for that?

There is indeed no geological record for PL subduction at that time period and our model does not include subduction between Sardinia and Corsica. The convergence due to motion of Iberia-Sardinia-Corsica is accommodated more towards the East in connection with the Eo-Alpine Orogeny, as proposed by Handy et al. (2010) (Figures 3.1 and 4).

**67. l. 315:** In your response to Stefan Schmid's comment you call this 230 km of strike-slip in Adria a 'simplification'. But that is not a simplification. It's a complication. It would be a simplification if there are 5 faults with a cumulative displacement of 230 km that you summarize by one fault. This overlap is an artifact of the speculation that Corsica and Sardinia are Iberia.

We built deforming topologies, i.e. deforming plate boundaries through time in GPlates, back to 250 Ma and thus it is a kinematic simplification to split a plate in two rather than in 6 or more.

**68. l. 268:** Why do you reconstruct Tisza-Datca in this paper? It's off-topic, and you have provided no information on this region in the paper so far.

It is indeed off-topic and that's why this section is very short, it is however visible on Figure 4 and need to be introduced to the reader. We thus kept this paragraph.

**69. l. 369:** How can this possibly be derived from the Adriatic plate? The Adriatic 'plate', which is not a plate in most of the time you reconstruct it, is the upper plate below which all these units accreted. So, it was during accretion part of the Eurasian plate. It may have been part of the Adriatic continent, but not of the Adriatic plate.

We removed "Plate" and kept "Adria".

**70. l. 377**: And then in 2020 I did not speculate about it like Schmid and Vissers did and made a reconstruction using structural geological and paleomagnetic data and showed that this interpretation is impossible. I'd say either make an analysis of this region and properly restore it, or leave it out.

See response to comment above (#68). As mentioned in the text to conclude this paragraph: "This area remains extremely simplified in our tectonic maps as this is out of scope of our study; a more detailed review and reconstruction of this area can be found in Schmid et al. (2020) and van Hinsbergen et al. (2020).

The rest of the paper contains a description of the model, and if the model is correct, the discussion is logical. I don't quite buy the reconstruction choices, so I refrain from discussing the implications of the model, I don't object to the first-order conclusions of slow-spreading in the PL Ocean.

**Response to Review #2 by Andrea Argnani**

**General comments**

The paper of Le Breton and co-authors aims at presenting an up-dated kinematic reconstructions of the Alpine-Mediterranean area, with a focus on Corsica-Sardinia-Adria, implemented within a recent global plate model. Kinematic scenarios are tested for geodynamic consistency using thermo-mechanical modelling of the rifting phase and compared with geological records from the Alpine region s.l. Some interpretative choices strongly mark the paper. I am not commenting the adopted motion of Iberia, a debated issue with various models present in the literature, but the choice of having the Corsica-Sardinia block attached to Iberia is not the most popular. Besides the palaeomagnetic evidence which is against it, it seems that this choice causes some inconsistencies in the presented kinematic reconstructions (Fig. 4). Some are described below. but the most relevant is the necessity of displaced northern Adria along a lithospheric fault for which there is no geological evidence. The reconstructions by Le Breton et al. present significant critical points that perhaps should be better discussed and compared with previous reconstructions, but above all they should be better supported by geological evidence. For this, I think the paper requires substantial revision work. Some of the critical issues are in the specific comments below.

We thank Andrea Argnani for the constructive comments and suggestions. We know that the model we choose for Sardinia-Corsica (and Iberia) is debated. We have made significant changes to the manuscript to address this issue, present alternative scenarios and arguments to support our kinematic model. The main argument is that other models based on paleo-magnetic data such as the recent one of Van Hinsbergen et al. (2020) would predict more than 500 km convergence between Iberia and Sardinia, of which there is absolutely no geological evidence. To avoid this, we rather overestimate the amount of strike-slip along existing strike-slip faults. Following your review, we changed the location of the fault within Adria to the Mattinata Fault, to follow strictly the proposed model of Schettino & Turco (2011; see answer to your specific comment on that point below). We acknowledge that the 230 km of strike-slip fault is very likely an overestimation. Future work should look at more diffuse intraplate deformation within Adria, as well as Iberia, which may help solve the debate on the paleo-magnetic data.

Despite this limitation in our kinematic reconstructions, we would like to stress that this motion of Iberia (post-145 Ma, opening of Bay of Biscay; Tugend et al. 2014) and within Adria (100-40 Ma; Schettino and Turco, 2011) does not influence the opening of the PL Basin which is earlier (200-145 Ma) and therefore does not change our main results and interpretations.

**Specific comments**

**Section 2.** Geological setting **line 80**: The microplate nature of Adria is debated. It not certain whether, when and for how long it was a microplate. Perhaps using just the term Adria avoids any questions.

Changed "microplate" to "plate", when referring to present-day setting (line 90). When going back in time, we only refer to Adria.

**lines 127-128:** It is not clear whether the Brianconnaise is considered a microcontinent (as

CSB) or as an extensional allochthon. It should be clarified.

The exact nature/former thickness of the Brianconnais is poorly constrained and discussed, we would rather keep the following formulation: Briançonnais continental "unit", as recorded today in the Alps (lines 146, 148).

**lines 131-132:** I believe that alternative interpretation are equally possible, without kinematically linking the Bay of Biscay and the Valais.

Alternative scenarios for the opening of the Valais have been added in section 3.1 and illustrated in the new version of Figure 3.

**lines 153-156:** from the reconstructions in Fig. 4 it seems that the Valais ocean opened at 130 Ma; the Jurassic opening is not obvious. Moreover, the location of the Valais to the north of the CSB looks a bit strange. This domain is still there at 83 and 67 Ma and even at 35 Ma, always between Eu and the CSB. I am not aware of evidence of remains of an ocean in Provence. Moreover in the 35 Ma frame a N-ward subduction is present north of the CSB; that subduction disappears in the subsequent 20 Ma frame (are there remains of it somewhere?), where subduction jumps to the south of the CSB.

Extension during rifting is not shown on Figure 4 (see text and figure caption: "Note these maps do not show intraplate deformation and rifting along the continental margins but the divergence between plates when considered rigid"). We don't consider the Valais Basin as a mature ocean but an extended continental/transitional crust (section 2.1.2). We checked and corrected the map to show only divergence between Europe and Iberia-Sardinia-Corsica. However, please note that the exact location of divergence/extension depends on where and how we draw the plate boundaries. This is not the scope of our study and is represented only in a simplified manner on Figure 4.

The thrust zone indicates the Pyrenean compressional deformation that affects also Provence until the Eocene. There are numerous remain of this contractional deformation in Provence with E-W trending thrusts and folds and syn-tectonics foreland deposition (e.g. Andreani et al. 2010; Espurt et al. 2012; Bestani et al., 2015, 2016; references added to the text).

**line 162:** why subduction initiation is intra-oceanic? Most authors consider that subduction initiated at the southern margin.

We rephrased this paragraph and focus on the first record of convergence between Adria and Europe between ca. 130-84 Ma (Eo-Alpine Orogeny), within the Eastern Alps (Austroalpine units) that record and intracontinental subduction zone, with sinistral component linked to subduction of Neo-Tethys to the east (see references in the text, lines 173-179).

We agree, the "Alpine" south-directed subduction initiated later at about 84 Ma on the southern margin (Sesia zone).

**Lines 196-197:** the extension in the Strait of Sicily may not reflect the motion of Adria, but it could be related to the dynamics of the subducted African slab (e.g., Argnani, 2009 SP Geol. Soc. London).

We agree and added a sentence and reference to this paragraph (now in section 3.4, lines 340-341).

**Section 3.3 MATF:** There is some confusion when describing the tectonics of the Adriatic Sea. The boundary inferred by DAgostino et al. (ca. E-W trending) is just sketched and is not related to a specific structure or set of structures. It refers to the present tectonic activity, and that's why a proper boundary is not yet developed... if there will ever be a plate boundary. The Mid-Adriatic Ridge of Scisciani and Calamita is NNW- SSE trending geological feature. It was previously named Central Adriatic Deformation Belt (Argnani and Frugoni, 1997) and is a belt of foreland inversion structures where the Adriatic seismicity tends to concentrate. It has also been considered as the in- land continuation of a major transform fault (Argnani, 2009, Bull Soc Geol It).

There is no evidence, however, of a major strike-slip displacement, and Scisciani and Calamita describe inversion not strike-slip structures. The age of this deformation ranges from Quaternary to possibly Eocene, whereas the activity of the large strike-slip fault (MATF) used in the reconstruction ranges from Late Cretaceous to Eocene. In addition, the authors use the poles of Northern Adria from Schettino et al. which considered Adria decoupled along the E-W-trending Mattinata Fault, located in the southern part of the Gargano promontory. This fault shows evidence of Paleogene activity (Argnani et al., 2009, GSA) but the amount of displacement is unlikely to be on the order of 100s km.

We corrected the location of the strike-slip fault to the Mattinata Fault across the Gargano Promontory as proposed by Schettino and Turco (2011). We follow their reconstructions for the motion of Adria (modified only for the post-20 Ma time period; Le Breton et al., 2017) as this avoids the overlap problem of Adria with Corsica-Sardinia. Indeed, our model for Sardinia is not the one recently proposed by Advokaat et al. (2014) and Van Hinsbergen et al. (2020), which uses paleo-magnetic data on Sardinia. However, their model implies more than 500 km convergence between Iberia and Sardinia (please see new version of Figure 3), for which there is no geological evidence of. We thus prefer to have a significant strike-slip motion within Adria and between Europe and Iberia-Sardinia-Corsica. We agree that the amount of strike-slip motion (230 km) implied by the Schettino & Turco (2011)'s model between 100-40 Ma is most likely over-estimated and was very probably distributed along several fault zones. However, this is poorly constrained and requires future work as mentioned in the text. However, we would like to emphasize that this strike-slip motion does not influence the kinematics of opening of the PL Basin which is much earlier, in the Jurassic, and which is the main scope of this paper. We also discuss this assumption in section 5.3 where we compare our plate convergence estimates with other kinematic models that do not have this split of Adria, and we show that our estimates can be considered as a maximum.

We improved the text accordingly in section 3 (lines 387-409) and 5.3, added geological description and the suggested references to describe the Mattinata Fault and Mid-Adriatic Ridge (or Central Adriatic Deformation Belt).

**Section 3.4** From Corsica to the west there is no evidence of an Alpine subduction followed by an Apennine-Maghrebian subduction. The two-subduction model adopted is one of the possible interpretations and other authors, more or less explicitly, prefer two opposite-vergence subductions since the beginning of convergence, handling in different ways the

interpretation of Alpine Corsica (e.g., Jolivet et al., 1998, Argnani 2012 Tectonoph.) Critical in the two-subduction model is the presence of the AlCaPeCa (micro) continent, that is represented at 67 and 35 Ma, with a size of about twice the Corsica-Sardinia Block. I found intriguing that the CSB is still almost intact within the Mediterranean, whereas the larger AlCaPeCa micro continent has been completely dismantled.

We mentioned this debate in the text and added the suggested references (lines 433-434).

**line 380**: explaining the Eo-Alpine orogeny using an intra-continental subduction that is part of strike-slip system linking the PL to the Vardar seems an ad hoc solution not based on evidence. (also for **lines 407-408**)

This was proposed based on field evidence from the Eastern Alps (Schuster and Frank, 1999; Neubauer et al. 1999; Frank and Schlager, 2006; Schmid et al. 2008; Stüwe and Schuster, 2010; these references are in the manuscript).

**Section 4.3** The authors take a geological section across the southern Alps to constrain the initial with of the rift, which is estimated to be ca. 300 km. This rift system has a Beta < 1.5, which is typical of continental rifts that do not evolve to oceanization. As described by many authors, including Froitzheim and Eberli, the rifting leading to oceanization is located further to the west, in present coordinate. Assuming that also this sector was affected by the first stage of rifting, the initial width of rifting was larger than 300 km. Otherwise, the western sector was affected only by the second stage of rifting, but this does not fit the inferred evolution which is at the base of the numerical modelling. This is actually a minor point for the paper, though the mechanical behaviour that controls the location of oceanization is an interesting issue.

We agree, the rift localized more to the west as shown on figure 6 and later in our modelling (section 4.4, figure 7). We used this section through the Southern Alps to estimate approximately the area affected by extension between Corsica and northern Adria. We agree that this is an absolute minimum, as this section represents only the preserved proximal part of (one side) the rift. We added this statement to the text (lines 535-536). We note that this minimalistic approach is completely independent from the geodynamic modelling done in Section 4.4. The geodynamic modelling is based on the amount of net divergence between Corsica and northern Adria in our kinematic reconstructions.

**Section 4.4** The portions of exhumed mantle that crop out in the Western Alps are considered of subcontinental origin (e.g., Piccardo and Guarnieri 2010, Int. Geol. Rev.), as in type 1 margins of Huismans and Beaumont, 2011. Does that fit with the result of the numerical modelling? In the model of Fig. 7 it seems that it is a newly formed lithospheric mantle to be exhumed.

Modelling subcontinental mantle exhumation is still an open challenge for the modelling community. We added a paragraph and references to address this point (lines 622-628):
"A subcontinental affinity of the exhumed mantle has been inferred from petrological analysis in the Western Alps suggesting diffuse porous flow of asthenospheric melts through the continental lithospheric mantle as a key process (Piccardo and Guarnieri 2010, Int. Geol. Rev.). Similar to other recent modelling efforts (Hart et al., 2017; Jammes & Lavier, 2019; Andrés-Martínez et al. 2019), our numerical models do not capture porous flow processes and are therefore not comparable with this type of observation. However they show that the

asthenospheric mantle resided close to continental mantle lithosphere prior to exhumation, which might be indicative of continental mantle affinity."

**Section 5.2**

**lines 600-603**: the opening of the Ionian basin in a NW-SE direction, with Malta and Apulia escarpments acting as transform faults, contrasts with using the Malta and Apulia as conjugate margins.

Yes, we agree that the Malta and Apulian escarpments were not conjugate (e.g. Frizon de Lamotte et al. 2011 and Tugend et al. 2019), but those escarpments were nevertheless formed during transtensional opening of the Ionian Basin. What we meant is that we used the width of the basin between these two escarpments to reconstruct the opening of the Ionian Basin. We changed this sentence accordingly (line 674).

**Section 5.3** The max. width of the oceanic domain in PL is taken as 250 km, based on the results of numerical modelling that also sets a length of 120 km for the hyperextended domain (80 + 40 km) and a length of 110 km for the necking domain (65 + 45 km). This gives a width of 480 km for the PL basin. Plate kinematics describe 680 km of convergence, that is subdivided in 420 km subduction and 260 km collision, based on the geologically inferred age of subduction and collision. With subduction initiating at the SE PL margin, and assuming that subduction occurs at the NW tip of the necking zone, after 370 km the conjugate necking zone is entering subduction: would that be considered onset of collision?

That's indeed a good point. Here, we distinguish between continental subduction and continental collision and define this clearer in section 2 (lines 191-204):
"The exact timing of onset of collision in the Alps differs depending on the criteria used. For instance, continental units, such as the above mentioned Briançonnais, entered the trench and were subducted prior to 35 Ma. However, here we distinguish continental subduction, in which rifted and thinned continental lithosphere behaves similarly to oceanic lithosphere, from continental collision, where slab pull is out-weighted by the positive buoyancy of the (less rifted) continental lithosphere following slab break-off and detachment of the subducted lithosphere. Timing of slab break-off in the Alps is inferred from timing of magmatism along the Periadriatic Line, mainly between 34-28 Ma (Rosenberg, 2004). Indeed, the geochemistry of these magmatic rocks indicates melting of lithospheric mantle, best explained by a slab break-off event (von Blanckenburg and Davies, 1995). Moreover, this time period (35-30 Ma) is also marked by a change in sedimentation from "Flysch" to "Molasse" (Rupelian Lower Marine Molasse, 33.9-28.1 Ma) in the Alpine Foreland Basin (Matter et al., 1980; Sinclair, 1997), and onset of medium temperature – medium pressure Barrovian-type metamorphism within the orogen (Lepontine dome, Tauern Dome; e.g. Bousquet et al., 2008) attributed to thickening of the European crust (Venediger Duplex formation in the Tauern Window; Scharf et al., 2013b). Thus, 35 Ma appears to be a reasonable time for onset of collision in the Alps, as defined above."

The numerical modelling is reproducing the various elements of the PL margins, and continental material was certainly subducted. However, I suspect that mixing geological dates of subduction and collision, amount of convergence from plate kinematics and time frames from numerical modelling may lead to quantitative results that are not really representative. How representative is the estimate of the amount of subducted continental material?

The amount of plate convergence we provide is based on the kinematic reconstruction and thus is quantitatively constrained. We only split the amount of plate convergence and discuss its variation in two time periods that we think represent different modes of convergence: 1) subduction with slab pull and subduction to high-pressure metamorphism which mostly occurred between 84-35 Ma in the Alps, to 2) collision after slab break-off event at about 35 Ma (see previous response above).

We clarified this in the text (section 5.3, lines 685-689).

**Figure 4.** Some of the reconstructions are puzzling, and many features are not really supported or described in Section 3. Sudden shifts in plate boundaries among frames seem not always justified. The frames from 130 to 35 Ma, in particular, present some intriguing aspects. From 130 to 83 Ma a major plate boundary rearrangement is depicted, with a system of large strike-slip lithospheric faults in the Adria region. I don't see much geological evidence for these features, and the connection between Alpine Tethys and Vardar looks a bit forced.

130—84 Ma is the most difficult period to reconstruct, the motion of Iberia and the onset of convergence between Africa and Europe must be accommodated somewhere. There is evidence for intracontinental subduction and sinistral strike-slip deformation within Austroalpine units (please see references above and in our revised manuscript). To link it with sinistral motion along the NPF makes kinematically sense, as previously proposed (e.g. Handy et al. 2010, and other references quoted in the manuscript), and avoid major problem with convergence between Iberia and Sardinia (as implied in model based on paleo-magnetic data; see new version of Figure 3).

The Valais ocean is positioned between CSB and Europe; such a reconstruction is difficult to support as described above.

The MATF that is represented in the frames 83 and 67 Ma has also some problems. as commented above.

Please see responses above, corrections to Figure 4 have been made accordingly.

**Minor corrections**

Strike slip motion along various fault systems is often mentioned throughout the text; the sense of motion, however, is almost always not indicated. Where possible this is a useful indication.

The two main strike-slip fault systems mentioned in the text are between Europe and Iberia-Sardinia-Corsica during the Cretaceous and within Adria during Cretaceous and Paleogene times, both are sinistral, which is now clearly indicated in the text and figure (lines 62, 402-405, Figure 3.1, Figure 4).

**line 23:** does the 250 km width refer to the truely oceanic part or does it include the exhumed mantle and hyperextended margin sectors too? The term ocean can be ambiguous. It looks it refers to the truly oceanic part, but this point should be clarified.

Yes, the 250 km represents the width of the mature ("true") ocean (not transitional), we clarified the text accordingly (line 24; lines 71-72; line 746).

**line 86:** Gawlick and Missoni is not in the References list.

We added the reference to the list

**line 90:** Channell and Kozur 1997 should be cited as an early paper describing the oceanic branches.

We added this reference.

**line 371:** Handy et al is 2015

**line 382:** Handy et al is 2015

**line 878:** Handy et al is 2015

This paper of Handy et al. was first published online in 2014, then in the Issue of January 2015, which brought confusion in our referencing system. We updated the citation as indicated on the website of Springer (2015).

**line 460:** "154-145 Ma and... 145-130 Ma". Ma and not km

corrected

**line 470**: indicate which is the Brune et al 2017 cited

2017a, added

**Fig. 5:** slight increase in velocity: "light to dark blue" instead of "dark to light blue"

We corrected this sentence (increase from NE to SW, light to dark)

---

## Author Response (AR2)

Dear Editor,

We would like to thank you for your decision and hereby submit the minor corrections of our manuscript in response to the two comments raised by Referee #2. All new modifications to the text are indicated in green.
We hope to have satisfactorily replied to the remaining questions (see detailed responses below) and remain at your disposal should there be any further questions/comments.

Best regards,
Eline Le Breton, on behalf of all co-authors

**Response to comment (1): Strike-slip connection between the North Pyrenean Fault and Eoalpine Orogeny**

We rephrased and added a few more references to clarify this paragraph (lines 176-183, also below). The mentioned references indeed suggested a late Jurassic age for the initiation of the intra-continental subduction within Adria (Alcapa/Austroalpine) and its potential link with the western branch of the Neo-Tethys to the east (Schuster and Frank, 1999; Frank and Schlager, 2006; Stüwe and Schuster, 2010). The first syn-orogenic sediments recording the onset of the Eo-Alpine compressional phase are however Early Cretaceous in age (c. 130 Ma; Faupl and Wagreich, 1999; Faupl and Tollmann 1979). Regarding, the potential connection of the sinistral strike-slip faulting within the Eo-Alpine units to the North Pyrrenean Fault towards the west during the opening of the North Atlantic in Cretaceous time, we now refer to two studies describing sinistral strike-slip faulting within the Austroalpine units in Cretaceous time: Neubauer et al. (1995) and Sieberer and Otner 2020 (note the latter is not yet published but was presented at the GeoUtrecht conferenece 2020 and is available on the GeoUtrectht website:

https://www.conftool.pro/geoutrecht2020/index.php?page=browseSessions&form_session=159).

**Text lines 176-183:** (1) Nappe stacking of continental units and high-pressure metamorphism was first recorded in the Eastern Alps, indicating an intracontinental subduction zone within Adria (Austroalpine unit, part of "AlCaPa"; Stüwe and Schuster, 2010), which developed possibly along late Jurassic strike-slip faults connected to the western termination of the Neo-Tethys Ocean (Schuster and Frank, 1999; Frank and Schlager, 2006). This phase corresponds to the "Eo-Alpine" Orogeny and lasted between c. 130- 84 Ma (Faupl and Wagreich, 1999), as indicated by both synorogenic clastics (Rossfeld Formation; Faupl and Tollmann, 1979) and geochronological data on high-pressure metamorphic rocks within the Austroalpine units of the Eastern Alps (e.g. Thöni, 2006; Manzotti et al., 2014, their Figure 5 and references therein). Regional scale sinistral strike-slip faults offsetting Austroalpine units were also active during Cretaceous time and were potentially related to the opening of the North Atlantic and subsequent motion of Iberia relative to Europe (Neubauer et al., 1995; Sieberer & Ortner, 2020).

**Response to comment (2): Test of the Jurassic opening of the Ionian Basin**

We rephrased to explain – hopefully more clearly – what we tested (lines 678-688, also below). The opening of the Ionian Basin influences indeed the position of Adria and if it opens in Jurassic time, it may solve (at least in part) the overlap problem with Corsica back to 200 Ma that we discuss the text and in Figure 3. However, it may leads to convergence rather than divergence between northern Adria and Sardinia-Corsica in the Jurassic, which does not fit with the geological record of rifting in northern Adria during that time. The area between

Africa and Adria was subject to several phases of extension (also earlier in the Permian) and further work is needed to constrain the exact amount and direction of extension/transtension contemporaneous to the opening of the PL Basin in the Jurassic before we can implement it properly in the kinematic reconstructions.

**Text lines 678-688:** We tested nevertheless an alternative kinematic scenario in which the Ionian Basin (using the present-day total width of c. 350 km between the Malta and Apulian escarpments; Tugend et al., 2019, their Figure 13a) opens – and thus Adria (Apulia) moves relative to Africa (Tunisia) – in Early-Middle Jurassic (200-164.7 Ma; Tugend et al., 2019) and in a NW-SE opening direction following Frizon de Lamotte et al. (2011). This would reduce the overlap problem between northern Adria and Sardinia-Corsica mentioned in section 3.2 (Figure 3). However, it would significantly increase the obliquity of motion between Sardinia-Corsica and Adria (and the rates of motion up to 9 mm/yr) and therefore reduce considerably the width of the rifted PL domain. Moreover, if we include a significant sinistral strike-slip motion between Africa and Adria during the opening of the Ionian Basin (following the interpretation that the Malta and Apulian escarpments are transform margins; Frizon de Lamotte et al. 2011), Adria would converge towards Sardinia-Corsica rather than diverge, which would be in conflict with the timing of syn-rift deposits and normal faulting along northern Adria (section 2). Future work is therefore required to test in more details such alternative scenarios.